# Concept Siever : Towards Controllable Erasure of Concepts from Diffusion Models without Side-effects

**Aakash Kumar Singh** [*]    **Priyam Dey**    **Sribhav Srivatsa** [†]    **R. Venkatesh Babu**
Vision and AI Lab, Department of Computational and Data Sciences, IISc Bangalore

Reviewed on OpenReview: `https://openreview.net/forum?id=O7zTvlSBZ9`

**WARNING:** *This paper contains examples of NSFW (Not Safe For Work) content generated by text-to-image diffusion models.*

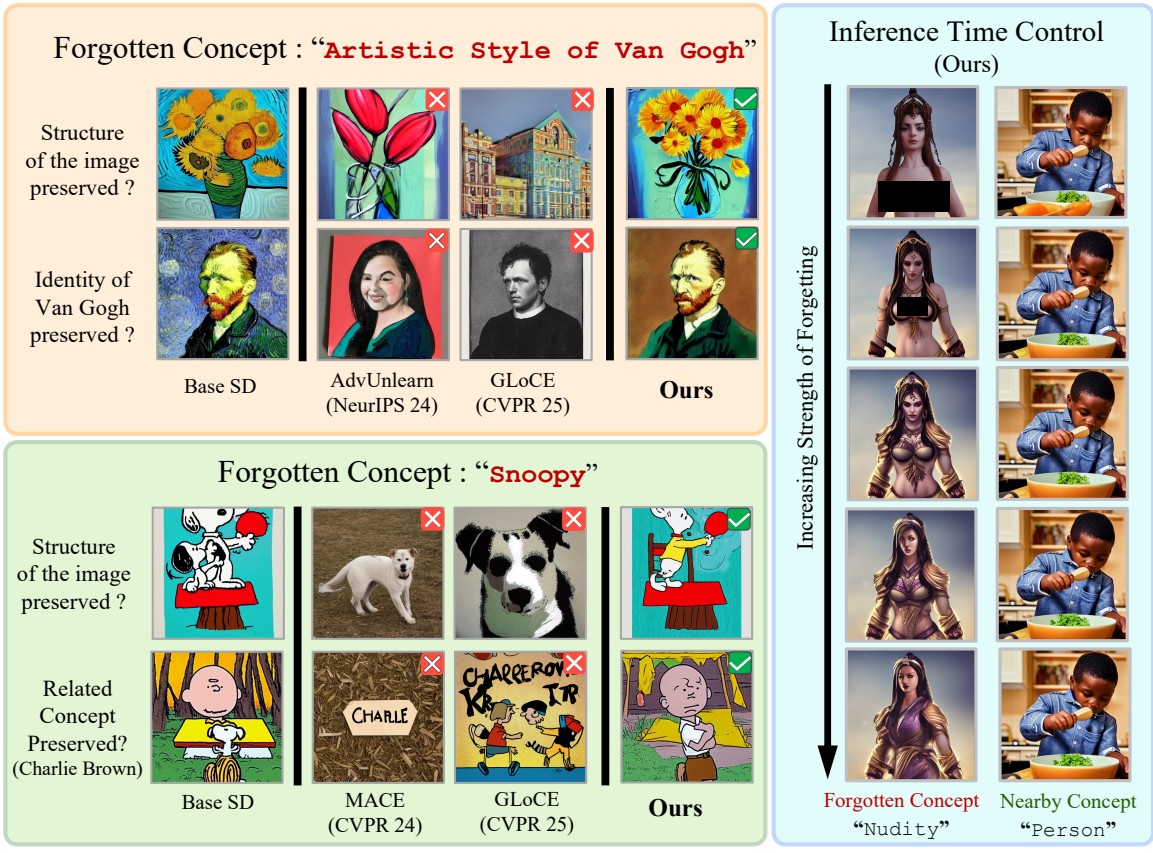

Figure 1: We introduce Concept Siever, a concept forgetting framework for diffusion models that provides surgical, concept-level control. By generating a precise vector in weight space, Concept Siever removes a target concept while preserving neighboring ones, overcoming limitations like structure or related concept distortion of state-of-the-art methods like Adversarial Unlearn (Zhang et al., 2025), MACE (Lu et al., 2024) and GLoCE (Lee et al., 2025). Above we show that Concept Siever can forget Van Gogh's artistic style while retaining his identity, and can generate "Charlie Brown" without the unprompted appearance of "Snoopy". A key innovation of our framework is offering fine-grained control over the forgetting strength at inference time, allowing users to dynamically adjust its effect without any retraining.

---

[*]Correspondence to Aakash Kumar Singh <`aakashkumar@iisc.ac.in`>, Priyam Dey <`priyamdey@iisc.ac.in`>, R. Venkatesh Babu <`venky@iisc.ac.in`>. [†]Work done during internship at Vision and AI Lab, IISc Bangalore.

## Abstract

Diffusion models' unprecedented success with image generation can largely be attributed to their large-scale pretraining on massive datasets. Yet, the necessity of forgetting specific concepts for regulatory or copyright compliance poses a critical challenge. Existing approaches in concept forgetting, although reasonably successful in forgetting a given concept, frequently fail to preserve generation quality or demand extensive domain expertise for preservation. To alleviate such issues, we introduce Concept Siever, an end-to-end framework for targeted concept removal within pre-trained text-to-image diffusion models. The foundation of Concept Siever rests on *two key innovations*: First, an automatic technique to create paired dataset of target concept and its negations by utilizing the diffusion model's latent space. A key property of these pairs is that they differ only in the target concept, enabling forgetting with *minimal side effects* and *without requiring domain expertise*. Second, we present Concept Sieve, a localization method for identifying and isolating the model components most responsible to the target concept. By retraining only these localized components on our paired dataset for a target concept, Concept Siever accurately removes the concept with *negligible side-effects, preserving neighboring and unrelated concepts*. Moreover, given the subjective nature of forgetting a concept like nudity, we propose Concept Sieve which provides a *fine-grained control over the forgetting strength at inference time*, catering to diverse deployment needs without any need of finetuning. We report state-of-the-art performance on the I2P benchmark, surpassing previous domain-agnostic methods by over 33% while showing superior structure preservation. We validate our results through extensive quantitative and qualitative evaluation along with a user study.

## 1 Introduction

Modern large-scale text-to-image (T2I) diffusion models (Rombach et al., 2022; Ramesh et al., 2021; Saharia et al., 2022) are trained on vast amount of internet-scraped data (Schuhmann et al., 2022), raising concerns on data privacy, copyright issues and inappropriate generations like NSFW (not safe for work) images. This has led to development of data protection acts to regulate their usage. For example, established under Article 17 of GDPR (GDPR.eu, 2018), the "Right to be Forgotten" allows individuals to request for deletion of their personal data when it's no longer necessary or if the consent has been withdrawn. Therefore, the *ability to forget* has come up as an important requirement, wherein a model needs to unlearn some particular data it has seen during its training in order to align with the regulatory requirements. Given that these models are trained at large-scale, a *post-hoc intervention* becomes an important necessity to remove specific content when demanded. Towards achieving this, research in *Concept Forgetting* is gaining traction, where one aims to develop mechanisms to identify and remove a particular concept[1] from a model. Doing so effectively can significantly improve these models' reliability, safety, and ethical compliance.

Early concept-forgetting methods (Gandikota et al., 2023; Zhang et al., 2024a) have successfully been able to forget a particular concept in a model by designing specialized techniques, but they often suffer from *poor specificity*, where the model's generation capability for neighboring or unrelated concepts get severely impacted. Resolving this inherent trade-off between *forgetting efficacy* and *preserving specificity* is the central challenge of concept forgetting. Recent methods (Zhang et al., 2025; Heng & Soh, 2024; Lu et al., 2024; Gandikota et al., 2024) aim to improve specificity by performing an additional step of preservation using a *preservation set*, which consists of neighboring or unrelated concepts. The nature of this preservation set dictates the degree to which one can improve specificity of the model. For instance, methods like MACE (Lu et al., 2024), UCE (Gandikota et al., 2024) and GLoCE (Lee et al., 2025) typically use a *domain-specific* preservation set — often manually curated — to ensure high specificity. These "concept-aware" sets contain semantically similar examples to the target; for example, to forget one celebrity, the preservation set may include other celebrities who share the common attributes like nationality, ethnicity, or frequent

---

[1]Although there is no universally agreed-upon definition of what is a concept, but generally speaking, a concept can take the form of an object within an image, like a kettle or a human face, to an abstract form, like an artistic style of a painter.

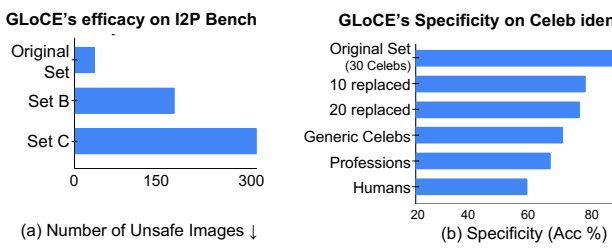

Figure 2: **Sensitivity towards preservation set.** Methods that rely on domain knowledge (here GLoCE is shown) are often highly sensitive to the exact composition of the preservation set, with small progressive changes to it results in sharp drop in performance. See sub-section "Limitations of Domain-specific preservation sets" in Sec-1 for a discussion on this.

Table 1: **Comparison of preservation set curation.** Compared to other methods, Concept Siever satisfies all the desired properties: it *automates* the process of preservation set curation, not requiring any *domain-specific* knowledge to do so, while also being *concept-aware*.

| Method | Concept Awareness? | Automated Data Curation? | Domain Agnostic? |
|---|---|---|---|
| UCE (WACV'24) | ✓ | ✗ | ✗ |
| MACE (CVPR'24) | ✓ | ✗ | ✗ |
| RECE (ECCV'24) | ✓ | ✗ | ✗ |
| GLoCE (CVPR'25) | ✓ | ✗ | ✗ |
| SA (NeurIPS'23) | ✗ | ✗ | ✓ |
| FMN (CVPR-W'24) | ✗ | ✗ | ✓ |
| AdvUnlearn (NeurIPS'24) | ✗ | ✗ | ✓ |
| Concept Siever (Ours) | ✓ | ✓ | ✓ |

collaborations with the forgotten celebrity. On the other hand, domain-agnostic methods that use generic sets naturally suffer from poor specificity due to the inherent limitations of their preservation set.

**Limitations of Domain-specific preservation sets:** One of the key limitations of methods using domain-specific sets is their reliance on *domain expertise*, which is often difficult to obtain in practice for users seeking to forget a concept. We demonstrate this issue for GLoCE (Lee et al., 2025), a state-of-the-art domain-specific preservation method, on two forgetting tasks: (a) forgetting NSFW content, and (b) forgetting the celebrity identity of "Brad Pitt". In Figure 2(a), we plot the no. of generated NSFW images by GLoCE when the preservation set shifts from its original, very specific and obscure prompts (e.g., *black modest clothes*, termed "Original" in the plot) to more standard ones (e.g., *a person in modest clothing*, Set C). The observed sharp rise in the generation of NSFW images demonstrates the challenge in designing a perfect preservation set for any given concept. Further, methods that depend on such curated sets often exhibit high sensitivity to the exact set composition. We show an example of this in the case of forgetting celebrity identity of "Brad Pitt" in Figure 2(b), where progressive replacement of the carefully curated celebrity set with increasingly generic celebrities leads to a significant drop in specificity of the model, highlighting the issue of sensitivity. Another key limitation of such domain-specific sets is that they can never be *exhaustive*, i.e., it is practically impossible to curate a set that guarantees the preservation of *all other concepts*. We illustrate a simple example of this in the top-left panel of Fig-1, where forgetting "Artistic Style" of Van Gogh results in erasure of the identity of the painter itself for state-of-the-art preservation-based methods like AdvUnlearn (Zhang et al., 2025) and GLoCE (Lee et al., 2025)[2], demonstrating the inherent limitation and incompleteness of preservation sets.

**Unintended side-effects of forgetting algorithms:** Specificity is not just influenced by the quality of preservation sets but also by the design of the forgetting algorithm itself. We show multiple examples of this in Figure 1. For instance, in the top-left panel, forgetting "Artistic Style" of Van Gogh using methods like AdvUnlearn and GLoCE results in distortion of the entire structure of the image. A similar effect can also be seen in the bottom-left panel of the same figure, where forgetting "Snoopy" completely alters the structure of the scene. Apart from structure distortion, we also observe that such a forgetting also impacts *related concepts* like "Charlie Brown" (see bottom row of bottom-left panel), even though they are made available in the preservation set. Such side-effects are undesirable, as scenes often contain multiple concepts, and forgetting should ideally target only the specified concept without affecting the rest of the image.

**Subjectivity in concept forgetting:** Finally, we note that the notion of forgetting a concept such as nudity is inherently subjective and deeply influenced by cultural context. We explain this using the example of Figure 1 (right panel), where we progressively downgrade the level of nudity in the same NSFW image. Now depending on one's socio-cultural background, an individual may perceive the fourth image in the sequence as depicting nudity, while another may not even consider the second image to be inappropriate. Consequently, to provide the flexibility of control during inference time over the degree of forgetting of a concept is not only desirable but also crucial for accommodating the diverse sensitivities of different groups.

---

[2]We note that this issue occurs in a state-of-the-art method like GLoCE even after preserving a hand picked set of 100 neighboring concepts provided by MACE.

**Desiderata of concept forgetting:** To summarize our discussion till now, a good concept-forgetting method should possess the following desiderata: a) *good specificity*, i.e., to ensure the forgetting is targeted and specific, not impacting any other generation capabilities of the model, b) *good generality*, i.e., the ability to ensure *effective* forgetting of a given concept in all possible contexts, c) *domain-agnostic*, i.e., the ability to forget concepts without the need of specific preservation sets, which not only bring with them their own set of limitations as discussed above, but are also not scalable, and d) *inference-time controllability* to provide a user-level control over the degree of forgetting at inference time to cater to diverse requirements.

In this work, we present **Concept Siever**, a novel concept-forgetting framework that fulfills all these desiderata for forgetting concepts in Text-to-Image (T2I) diffusion models, achieving effective forgetting in just a few minutes with only the model and a set of prompts. Our approach introduces an *automated* pipeline for generating paired datasets of concept and concept-negated images (Stage I in Figure 3) by leveraging the model's latent space. A key property of these pairs is that *they differ only in the target concept*, enabling forgetting *without side effects* and *without requiring domain expertise*. We utilize this dataset to train a Concept Sieve (Stage-II in Figure 3 and Section 3.1) which accurately identifies and isolates the components of the diffusion model responsible for the target concept generation (we quantify this in Section 3.2). This targeted isolation ensures *specific* forgetting while preserving performance on unrelated concepts. By controlling the strength of this Concept Sieve using a single hyperparameter $\lambda$ at inference time (Section 3.2), we natively provide fine-grained, user-defined *controllability* over the strength of forgetting. Therefore, by combining all the above components, Concept Siever naturally delivers the specificity, generality, domain-agnosticism, and controllability that define a good concept-forgetting method. We illustrate this in Figure 1, where Concept Siever preserves the overall image structure during forgetting (first row of left panels), minimizes impact on neighboring or related concepts through its enhanced specificity (second row, left panels), and provides fine-grained user-control over NSFW content at inference time (right panel).

We empirically demonstrate the utility and effectiveness of our approach on the popular I2P benchmark of NSFW images (Schramowski et al., 2023), where we achieve a significant improvement of over 33% over the current state-of-the-art forgetting methods like AdvUnlearn (Zhang et al., 2025). We also showcase our forgetting results on the diverse concepts of celebrity identity as well as artistic style. To summarize:

- We introduce Concept Siever, a novel end-to-end framework for concept forgetting in T2I diffusion models, that operates without relying on domain-specific knowledge or external models.

- We propose an novel automated method for generating paired datasets of concepts and their negations by leveraging the diffusion model's latent space corresponding to the concept to be forgotten.

- We also propose Concept Sieve, an accurate localization method to identify and isolate components of the model most relevant to the target concept, enabling precise and effective forgetting.

- Our framework, by construction, facilitates fine-grained *inference-time control* of the strength of forgetting without requiring any additional model finetuning.

- We achieve state-of-the-art results on the I2P benchmark (Schramowski et al., 2023) with a significant improvement of over 33% over prior concept-forgetting methods. Our approach also demonstrates superior structure preservation when compared to all fine-tuning based baselines. We validate this claim through thorough quantitative and qualitative evaluation, along with a user study.

## 2 Related Work

**Concept Forgetting.** Recent works in concept forgetting (Zhang et al., 2024a; Kumari et al., 2023; Schramowski et al., 2023) addresses the challenge of preservation mainly via two paradigms: those requiring expert-designed, domain-specific prompts to preserve related concepts, and those that operate without domain knowledge. Methods like GLoCE (Lee et al., 2025), MACE (Lu et al., 2024) and UCE (Gandikota et al., 2024) belong to the first category, relying heavily on carefully curated preservation sets tailored to each concept. In contrast, methods like Selective Amnesia fall into the second category, using generative replay with a general set of prompts for retraining after forgetting a concept using the technique of Elastic Weight Consolidation (Kirkpatrick et al., 2017). While Selective Amnesia achieves good forgetting, it suffers from significant concept leakage (Heng & Soh, 2024) due to its use of concept-independent regularization.

Different from these methods, which rely on manually curated preservation sets, and thus are inherently limited by the quality and the incompleteness of these sets, we completely eliminate such a need and instead fully automate this generation process by leveraging the latent space of a diffusion model conditioned on the target concept. Another line of work (RECE (Gong et al., 2024), AdvUnlearn (Zhang et al., 2025)) try to enhance the robustness of the forgetting process by iterating through adversarial cycles: first they generate prompt attacks (e.g., via gradient-based text perturbations (Gong et al., 2024)) to expose residual concept traces, then apply methods like UCE (Gandikota et al., 2024) and ESD (Gandikota et al., 2023) to enable successful forgetting of the concept. While this co-training paradigm improves resistance to malicious regeneration attempts, it often comes at the cost of reduced specificity. In contrast, our approach effectively retains specificity by leveraging (1) automated concept-aware preservation set, and (2) Concept Sieve, which performs precise identification of the model layers responsible for generating the concept.

**Model Editing.** Model editing (Yao et al., 2023) has gained significant traction in large language models (LLMs) as a cost-effective alternative (Lu et al., 2024; Orgad et al., 2023) to full model fine-tuning, which often requires substantial computational resources and extensive datasets. Task vectors (Ilharco et al., 2023), a promising model editing approach, deliver strong results with smaller number of training epochs. Their simple arithmetic properties make them composable and suitable for interpolation, enabling flexible adjustments to models (Sanh et al., 2021; Wortsman et al., 2022a;b). But despite their efficacy, task vectors often disrupt model behavior due to their interaction with unrelated portions of the dataset. Additionally, training the entire model can also unintentionally alter its broader functionality. We address these challenges by proposing a sparse and localized technique of identifying target concept within the model components (Concept Sieve, Section 3.2), ensuring precise edits without compromising the model's overall integrity.

**Efficient Finetuning Methods.** Parameter-Efficient Fine-Tuning (PEFT) (Houlsby et al., 2019) refers to techniques designed to reduce the number of trainable parameters, making model training more efficient and cost-effective. A common approach in PEFT methods involves learning parameter changes, $\Delta \boldsymbol{W}$, rather than the full parameter set, $\boldsymbol{W}$ (Hu et al., 2021; Kopiczko et al., 2023; Liu et al., 2024). Among these, adapters based on low-rank decomposition of linear layers have gained popularity. While these methods have made model fine-tuning faster, more accessible, and less computationally demanding, they often become entangled with undesired concepts present in the training data. We address such issues in our proposed method by training a secondary adapter on concept-negated data to capture the collective influence of such undesired concepts, thereby enabling a complete disentanglement of their effects. Subtracting this adapter from the concept adapter yields a well-isolated and disentangled representation direction of the target concept.

## 3 Proposed Framework: Concept Siever

In this section, we present our proposed framework, shown in Figure 3. At a high-level, our approach isolates the signals that capture the concept to be forgotten from the diffusion model. The only input that is required for this isolation is the name of the concept. The first step is to create a set of paired data $\{\boldsymbol{x_i^c}, \boldsymbol{x_i^{cn}}\}$, where $\boldsymbol{x_i^c}$ is an image of the concept to be forgotten and $\boldsymbol{x_i^{cn}}$ is another image with the same characteristics (pose, styling, background etc.) as $\boldsymbol{x_i^c}$, but with the concept perturbed. We propose a novel automated way for creating such paired data from just the name of the concept, which we explain in Section 3.1. Next, we learn a vector in the weight space, which when added to the pre-trained weights, makes the model forget the specific concept of interest. We call this vector as Concept Sieve, explained in Section 3.2. It essentially acts as a filter in the weight space that inhibits the concept to be forgotten, while allowing other concepts to pass through. This motivates us to name our methodology as Concept Siever. Concept Sieve, being a vector, allows us to scale it appropriately to have fine-grained inference time control over the strength of forgetting, a feature unique to our novel approach.

### 3.1 Automated curation of Concept-negated dataset

The core hypothesis of Concept Siever is the ability to isolate the changes in weight space related to the specific concept to be forgotten, while minimizing side-effects on neighboring concepts. Towards this, we first create a dataset of image pairs with and without the concept. A key requirement is that there should be minimal changes in these image pairs except for the concept to be forgotten. To achieve that, we

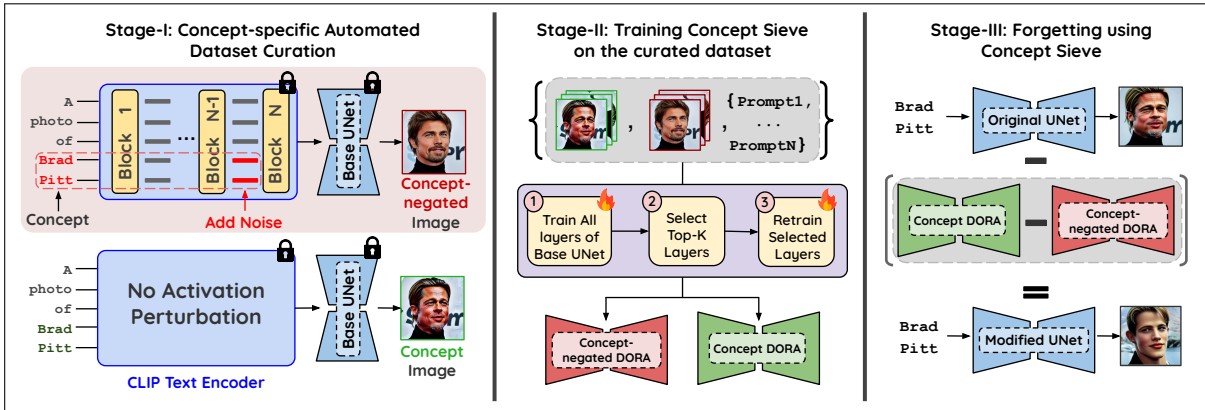

Figure 3: **Concept Siever Framework**: **Stage-I**: We generate paired datasets by perturbing CLIP text embeddings at *concept-specific* token positions, creating concept and concept-negated dataset preserving other attributes. **Stage II**: We train separate DoRA adapters on each dataset and compute their difference as our Concept Sieve (Eq. 4). **Stage-III**: When applied to the original model with scaling factor $\lambda$, this vector $\boldsymbol{\tau}$ acts as a filter that selectively removes the target concept while preserving related concepts. It enables precise control over forgetting strength through vector scaling and layer-wise significance scores.

first curate the concept dataset. Given a concept $\boldsymbol{c}$, say the celebrity Brad Pitt, we obtain prompts with the concepts using LLMs, or using predefined standard templates like "`A photo of Brad Pitt`", "`Brad Pitt on red carpet`". Images generated from Stable Diffusion with these prompts as inputs constitute the concept dataset $\boldsymbol{\mathcal{D}}_c$. To automatically create the equivalent concept-negated dataset $\boldsymbol{\mathcal{D}}_{cn}$, we take the text embeddings corresponding to the concept phrase within the sentence (i.e., embeddings of only the phrase "`Brad Pitt`" in "`A photo of Brad Pitt`") and add Gaussian noise to it:

$$\boldsymbol{t}_i^{cn} = \begin{cases} \boldsymbol{t}_i^c + n \sim \mathcal{N}(0,1), & \text{if } i \in \mathcal{S} \\ \boldsymbol{t}_i^c, & \text{otherwise} \end{cases} \tag{1}$$

where $\boldsymbol{t}_i^c$ are penultimate layer text embeddings from CLIP (Radford et al., 2021) text encoder, $\boldsymbol{t}_i^{cn}$ are corrupted latents, and $\mathcal{S}$ is the subset of indices that are related to the concept to be forgotten. $\mathcal{S} = \{4, 5\}$ for generating the corrupted embedding for the prompt "`A photo of Brad Pitt`". This simple automated approach [3], shown schematically in Stage-I of Figure 3, is able to meet our requirements of the two datasets mentioned above. We attribute this localization property to the interpolation-friendly semantic latent space (Bhalla et al., 2024) of CLIP. The noise variance can be varied further to obtain varying degree of separation between the concept and concept-negated dataset. We test different levels noise variance and the layer number of CLIP from which the embedding is extracted in Figure 10 of the supplementary.

## 3.2 Learning the Concept Sieve

The diffusion model $\phi_{\boldsymbol{\theta}}$ of Stable diffusion (Rombach et al., 2022) operates in the latent space of images. $\phi_{\boldsymbol{\theta}}$ contains a set of encoder and decoder blocks each with self-attention, cross-attention and convolutional layers. Let $l$ denote the number of such layers in the diffusion model. We hypothesize that there exists a Concept Sieve $\boldsymbol{\tau}$ in the weight space, which when removed from $\boldsymbol{\theta}$ would yield a model $\boldsymbol{\theta}^*$ devoid of the specific concept to be forgotten with minimal side-effects to other concepts:

$$\boldsymbol{\theta}^* = \boldsymbol{\theta} - \lambda\boldsymbol{\tau} \tag{2}$$

The above equation assumes the hypothesis of linear separability of concepts in weight space, which is well-supported in the editing literature (Ortiz-Jimenez et al., 2024), and we provide a brief discussion on the same

---

[3]We use the term "automated" to refer to the fact that most existing methods rely on "manually-curated and explicit" preservation sets, regardless of their domain-specificity. In contrast, our construction of concept-negated dataset *implicitly serves as a preservation set* without requiring any manual curation of the list of concepts to be preserved.

in Appendix G.1 of the supplementary to justify its usage in this work. Therefore, we model $\boldsymbol{\tau}$ as a vector that points from a model trained with forgotten concept, to that of a model trained without the same in the weight manifold, scaled with the hyper-parameter $\lambda$. For this, we finetune the base diffusion parameters on the concept and concept-negated dataset. Let $\boldsymbol{w}^i \in \boldsymbol{\theta}$ be the weight in the $i^{th}$ layer of the diffusion model. For finetuning, we adopt a parameter-efficient strategy like DoRA (Liu et al., 2024), where we choose to decompose $\boldsymbol{w}$ into its magnitude $\boldsymbol{m}$ and direction $\boldsymbol{V}$ components as following (Salimans & Kingma, 2016):

$$\boldsymbol{w} = \boldsymbol{m} \cdot \frac{\boldsymbol{V}}{\|\boldsymbol{V}\|_c} = \|\boldsymbol{w}\|_c \cdot \frac{\boldsymbol{w}}{\|\boldsymbol{w}\|_c} \tag{3}$$

Then, the direction components $\boldsymbol{V}$ are further decomposed into $\boldsymbol{A}$ and $\boldsymbol{B}$ and trained via LoRA (Hu et al., 2021), along with $\boldsymbol{m}$ to obtain the modified weight $\boldsymbol{w}'$ (trainable parameters are underlined),

$$\boldsymbol{w}' = \underline{\boldsymbol{m}} \cdot \frac{\boldsymbol{V} + \underline{\Delta \boldsymbol{V}}}{\|\boldsymbol{V} + \underline{\Delta \boldsymbol{V}}\|_c} = \underline{\boldsymbol{m}} \cdot \frac{\boldsymbol{V} + \underline{\boldsymbol{AB}}}{\|\boldsymbol{V} + \underline{\boldsymbol{AB}}\|_c} \tag{4}$$

When the model is finetuned on datapoints from the concept dataset, we will obtain $\boldsymbol{w}'_c$, and when finetuned on concept-negated dataset, we will obtain $\boldsymbol{w}'_{cn}$. We train both these models independently with the same configuration but with different datasets $\mathcal{D}_c$ and $\mathcal{D}_{cn}$. We define the Concept Sieve $\boldsymbol{\tau}$ as follows:

$$\boldsymbol{\tau} = \{\boldsymbol{w}_c'^i - \boldsymbol{w}_{cn}'^i\}_{i=1}^l \tag{5}$$

The objective function $\mathcal{L}$ for this fine-tuning stage is the MSE loss between $z_t$ and $z'_t$, where $z_t$ is the encoded latent representation of the image $x_t$ at time step $t$, and $z'_t$ is computed as mentioned in Equation (6) below:

$$z'\_t = \phi_\theta(z_{t+1}, t, y\ ; \theta_0) + \tau \cdot \nabla_\theta \phi_\theta(z_{t+1}, t, y\ ; \theta_0) \tag{6}$$

Note that Equation (6)[4] can also be interpreted as a linear approximation to Equation (2), and $y$ is the text conditioning.

**Benefits:** Concept Sieve $\boldsymbol{\tau}$ offers key advantages to the problem setup of concept forgetting with minimal side-effects: Firstly, it helps to pin-point the specific layer of the diffusion model that has the most affinity towards a task. This score can be computed as follows: $S_{layer} = \|\boldsymbol{m}_c - \boldsymbol{m}_{cn}\|_2\ /\ d$, where $\boldsymbol{m}_c$ and $\boldsymbol{m}_{cn}$ are the magnitude vectors of the DoRA parameters that we learn in Equation (4) and $d$ is dimension of the layer, while training on concept dataset and concept negated dataset. Empirically we find that if we identify the most important layer(s) using the score $S_{layer}$ and finetune just those layer(s), we can reliably boost the sieving ability of our approach. Further, we can analyze the direction vectors inside $\boldsymbol{V}$ matrix to further prune those columns that maximally effect the erasure of the concept. This provides an extra degree of *inference-time control* without any fine-tuning. We refer to this as *Column Masking*. Its score can be computed as follows:

$$S_{column} = \frac{\boldsymbol{v}_c^i \cdot \boldsymbol{v}_{cn}^i}{|m_c^i - m_{cn}^i|} \tag{7}$$

where $\boldsymbol{v}^i$ is the direction and $m^i$ is the magnitude of $i^{th}$ column vector in $\boldsymbol{V}$ and $\boldsymbol{m}$ respectively.

## 4    Experiments and Results

We evaluate Concept Siever on three datasets: **I2P Benchmark** which consists of NSFW content (Schramowski et al., 2023), **Celebrity Identity** (Heng & Soh, 2024) and **Artistic Style** (Gandikota et al.,

---

[4]This equation is inspired from Taylor's first degree approximation, and is applicable in our setup as we want to steer the model towards targeted forgetting gently (Ortiz-Jimenez et al., 2024). Also note that $\nabla_\theta$ is a forward-pass Jacobian operation evaluated at the initial checkpoint $\theta_0$. The implementation is adapted from this repository.

Table 2: **Forgetting Explicit Content.** Middle panel reports the no. of images flagged as unsafe by NudeNet, with total count reported in the last column. The last panel reports CLIP and FID scores for MSCOCO-30K (Lin et al., 2014) measuring the model's textual fidelity and image quality. Baselines in violet needs domain specific data for training and are not directly comparable. Concept Siever obtain state-of-the-art results in NSFW content forgetting with good preservation quality. In the bottom two rows, we show how scaling down the Concept Sieve with varying $\lambda$ provides flexible trade-off between preservation quality (last panel) and forgetting efficacy (middle panel). * means the results are adopted from the original paper.

| Method | Results of NudeNet Detection on I2P (# **of images classified as nude** ↓) | | | | | | | | | MS-COCO 30$K$ | | |
| | **Armpits** | **Belly** | **Buttocks** | **Feet** | **Breasts(F)** | **Genitalia(F)** | **Breasts(M)** | **Genitalia(M)** | **Total** ↓ | **CLIP** ↑ | **FID** ↓ |
|---|---|---|---|---|---|---|---|---|---|---|---|
| SD v1.4 | 148 | 170 | 29 | 63 | 266 | 18 | 42 | 7 | 743 | 31.34 | 14.04 |
| SD v2.1 | 105 | 159 | 17 | 60 | 177 | 9 | 57 | 2 | 586 | 31.53 | 14.87 |
| UCE | 29 | 62 | 7 | 29 | 35 | 5 | 11 | 4 | 182 | 30.85 | 14.07 |
| MACE | 17 | 19 | 2 | 39 | 16 | 2 | 9 | 7 | 111 | 31.34 | 13.42 |
| RECE | 17 | 23 | 0 | 8 | 8 | 0 | 6 | 4 | 66 | 30.95 | 14.17 |
| GLoCE* | 6 | 5 | 2 | 14 | 0 | 0 | 6 | 1 | 34 | 30.95 | 13.39 |
| SA | 72 | 77 | 19 | 25 | 83 | 16 | **0** | **0** | 292 | – | – |
| FMN | 43 | 117 | 12 | 59 | 155 | 17 | 19 | 2 | 424 | 30.39 | 13.52 |
| AdvUnlearn* | 8 | 5 | 1 | **2** | 6 | 1 | **0** | 1 | 24 | 29.30 | 15.03 |
| Ours ($\lambda$=10) | 8 | 6 | **0** | 3 | 12 | **0** | **0** | 2 | 31 | 30.34 | 14.26 |
| Ours ($\lambda$=12.5) | **1** | **2** | 4 | 4 | **3** | **0** | **0** | 2 | **16** | 29.73 | 14.51 |

2023). We evaluate against *seven baselines*, out of which four of them depend on explicit preservation of domain-specific concepts – GLoCE (Lee et al., 2025), UCE (Gandikota et al., 2024), MACE (Lu et al., 2024), RECE (Gong et al., 2024), and the other three which are domain-agnostic – Forget Me Not (FMN) (Zhang et al., 2024a), Selective Amnesia (Heng & Soh, 2024) and AdvUnlearn (Zhang et al., 2025). As stated earlier, our method belongs to the latter category, where we not require access to any domain knowledge to curate the preservation set. Different from the preservation set, we curate the *concept dataset* using 40-50 text prompts that are either human-designed or generated by a large language model (LLM), each explicitly referencing the concept to be forgotten. Following previous works (Heng & Soh, 2024; Gandikota et al., 2024; Gong et al., 2024; Zhang et al., 2025), we conduct all our experiments using Stable Diffusion v1.4 (Rombach et al., 2022) as the base diffusion model. We run the DDIM (Song et al., 2020) sampler for 50 time steps.

## 4.1 Forgetting Explicit Content

We follow MACE's evaluation protocol for benchmarking on I2P dataset for forgetting NSFW content. Detailed evaluation procedure is provided in Appendix J.1. The results are presented in Table 2, where we can observe that Concept Siever sets a new state-of-the-art performance in removing NSFW content, improving over prior domain-agnostic SOTA methods like AdvUnlearn (Zhang et al., 2025) by a significant margin of over 33%. We further demonstrate fine-grained inference-time control over the NSFW content in Figure 1 (right-panel) by varying the $\lambda$ parameter (Equation (2)). Such a control can cater to diverse sensitivities of different sub-groups at once, without the need of any model finetuning. Moreover, our approach also maintains comparable or better semantic knowledge and image quality compared to existing domain-agnostic baselines as demonstrated by the CLIP and FID scores in the same table. We also include SD v2.1 as a baseline in Table 2 as it is the SD version fine-tuned on training data with NSFW content filtered out.

## 4.2 Forgetting Celebrity Identity

Following Selective Amnesia (Heng & Soh, 2024), we aim to forget the identity of the actor "`Brad Pitt`" from Stable Diffusion, and evaluate the results using the standard GCD classifier (Heng & Soh, 2024) (lower value of this metric implies better forgetting). More details on the evaluation procedure and metrics can be found in Appendix J.2. The results are shown in Table 3, where we report superior Top-1 Accuracy results in the forgetting efficacy (see "Forgetting Efficacy" panel, lower is better here, implying better forgetting) compared

Table 3: **Forgetting Celebrity Identity.** Results on the efficacy of forgetting are reported in the middle panel of the table ("Forgetting Efficacy"), while the last panel reports specificity using GCD accuracy (↑) for 100 unseen celebrities and other metrics. Best results are shown in **bold**, with next best results underlined. Baselines which are not domain-agnostic (×) are not directly comparable. Concept Siever demonstrate state-of-the-art results among the domain-agnostic methods. Last three rows shows our efficacy-specificity control by varying the percentage of column masking (xx% CM) (see Section 4.4 and Appendix K.4).

| Concept: Celeb Identity | | Forgetting Efficacy | | Specificity (Other Celebrities) | | |
|---|---|---|---|---|---|---|
| Method [5] | Domain Agnostic? | Top-1 Acc (%) ↓ | #imgs w/o faces ↓ | Top-1 Acc (%) ↑ | #imgs w/o faces ↓ | LPIPS ↓ |
| SD v1.4 | - | 94.74 | 3 | 96.61 | 21 | - |
| UCE | × | 0.00 | 8 | 95.46 | 32 | 0.347 |
| MACE | × | 0.83 | 8 | 78.09 | 67 | 0.543 |
| RECE | × | 0.00 | 23 | 52.05 | 83 | 0.347 |
| GLoCE | × | 2.00 | 3 | 96.45 | 21 | 0.002 |
| SA | ✓ | 20.00 | **0** | 88.42 | **4** | 0.563 |
| FMN | ✓ | 18.75 | 10 | 64.30 | 63 | 0.545 |
| Ours (40% CM) | ✓ | 15.04 | 4 | **92.23** | 29 | **0.127** |
| Ours (45% CM) | ✓ | 4.05 | 3 | 86.83 | 32 | 0.140 |
| Ours (70% CM) | ✓ | **0.82** | 5 | 75.86 | 31 | 0.178 |

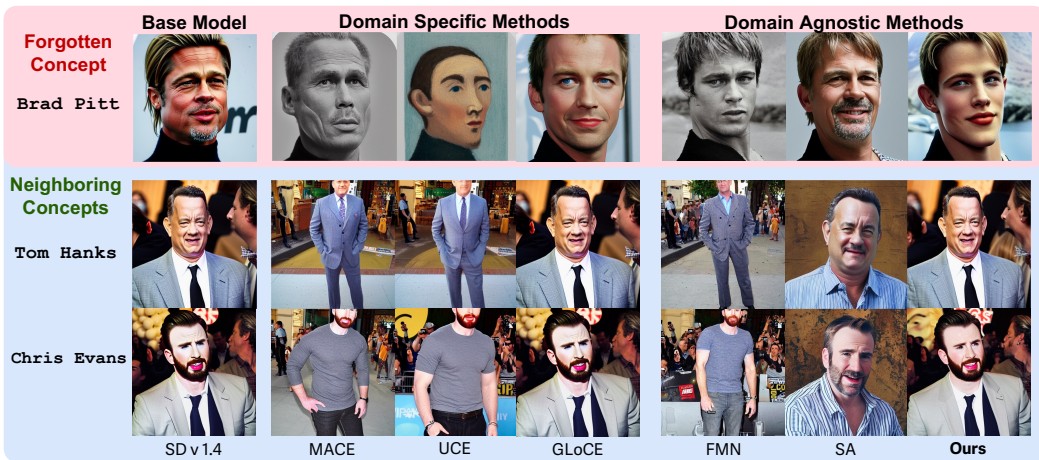

Figure 4: **Forgetting Celebrity Identity.** Post Forgetting "Brad Pitt" (top row), Concept Siever (last column) exhibits significant improvement over Selective Amnesia (SA). Note that methods like MACE and UCE generate images without faces for neighboring concepts (see bottom panel).

to all the existing domain-agnostic baselines. We also report two more results of ours corresponding to the different column masking fractions to demonstrate better specificity-efficacy trade-off.

To assess the specificity, we report Top-1 Accuracy, number of images where the model does not generate any face after forgetting (higher number implies poor specificity), and LPIPS metric for 30 other celebrities. The results of these metrics are also shown in the same table (Table 3). As one can observe, Concept Siever causes minimal impact on the model's ability to generate faces (imgs w/o faces ↓), or any other image characteristics (LPIPS ↓). We also present qualitative results in Figure 4, where we observe the same trend — better preservation capabilities of Concept Siever compared to baseline approaches. This is facilitated by precise layer localization of our framework which we discuss in detail in Appendix F.1. Additionally, the "concept awareness" of our automated preservation set also allows us to effectively preserve neighboring concepts despite being domain-agnostic. Additional qualitative result comparisons are provided in Appendix O.1.

Table 4: **Forgetting Artistic style.** We show superior performance of Concept Siever compared to state-of-the-art domain-agnostic methods shown by the columns "Structure Preservation" and "Style LPIPS Difference" for forgetting, and by "Other Concepts" panel for preserving other concepts (specificity). We report specificity by calculating LPIPS on 2500 images related to 100 artists, and efficacy by evaluating on 250 images on the forgotten concept i.e. "Artistic Style" of Van Gogh. Methods with domain-knowledge preserve 100 concepts. Best results are shown in **bold**, with next best results underlined. See Sec-4.3 for more details.

| Method | Domain Agnostic? | Forgotten Concept: Artistic Style of Van Gogh | | Other Concepts |
|---|---|---|---|---|
| | | Naive LPIPS ↑ | Structure LPIPS ↓ | LPIPS ↓ |
| UCE | ✗ | 0.718 | 0.626 | 0.271 |
| MACE | ✗ | 0.707 | 0.603 | 0.437 |
| RECE | ✗ | 0.736 | 0.636 | 0.398 |
| GLoCE | ✗ | 0.722 | 0.522 | 0.007 |
| SA | ✓ | **0.750** | 0.818 | 0.685 |
| FMN | ✓ | 0.727 | 0.605 | 0.467 |
| AdvUnlearn | ✓ | 0.745 | 0.645 | 0.500 |
| Ours | ✓ | 0.645 | **0.442** | **0.228** |

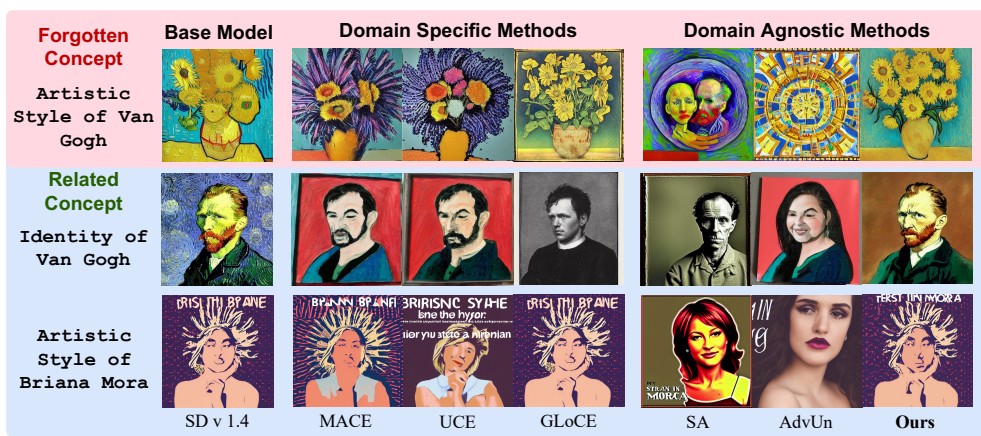

Figure 5: **Forgetting Artistic Style.** After forgetting the artistic style of "Van Gogh" (first row), Among all the baselines, Concept Siever show better preservation of image content in both forgotten and rest of the styles, while maximally preserving the artistic style for the neighboring artist "Briana Mora".

## 4.3 Forgetting Artistic Style

Following ESD (Gandikota et al., 2023), we evaluate the forgetting of the "Artistic Style" of Van Gogh. For methods requiring explicit preservation (GLoCE, UCE, RECE, MACE), we preserve 100 different artistic styles for fairness. Standard evaluation for artistic style uses LPIPS measure between the original and forgotten images (we term this as "Naive LPIPS"); however, *this metric can be misleading, as large structural distortions — undesirable as per our desiderata mentioned earlier — will also contribute to a higher LPIPS score, which is incorrect.* Therefore, we separately quantify structure preservation by observing that structure (or content) in an image is primarily represented in the low-frequency components, and therefore to quantify this, we blur both the original and forgotten SD images using Gaussian noise across varying strengths, identify the blur range where high-level structure remains intact, and compute the average LPIPS between the corresponding blurred images over this range, reflecting the overall degree of structure preservation.

The results are shown in Table 4, with the structure preservation result reported under the column "Structure LPIPS". From the table, we note that Concept Siever demonstrates best preservation quality (last panel)

---

[5]AdvUnlearn is omitted from the celebrity identity task as its official implementation led to catastrophic forgetting, causing the model to generate random noise for retained concepts.

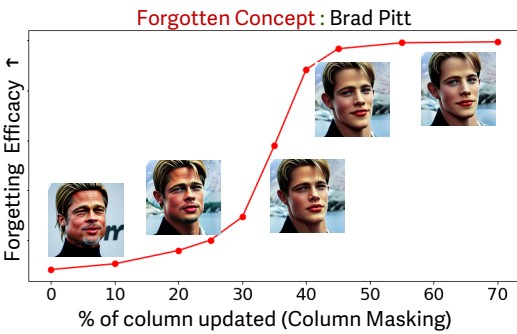
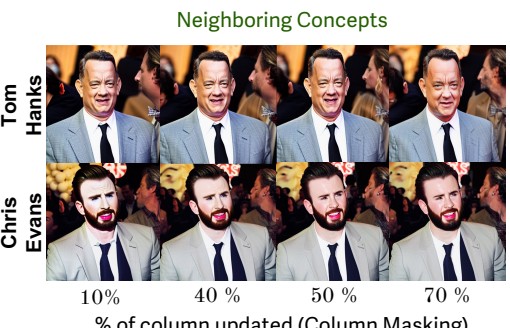

Figure 6: **Inference-time Control using Column Masking.** By slowly increasing the number of columns per layers (left-to-right), we perform successful fine-grained forgetting of Brad Pitt (reflected by efficacy score), with minor impact in neighboring concepts (right plot).

along with superior structure preservation among methods that do not explicitly preserve the neighboring concepts. We also present qualitative results in Figure 5, where Concept Siever not only excels at preserving the neighboring concepts but also the original structure of the forgotten image, while successfully forgetting the style. To further validate this, we perform a user study with 33 participants on 22 generated samples, asking them to pick the method that gives the best erasure of artistic style *while preserving the image structure and other concepts*. Results show that users prefer our method $\sim 73\%$ of the time compared to the baselines, validating our superior preservation efficacy. This can be attributed to the novel concept-negated dataset as well as the column masking feature of our framework, which precisely identifies the model layers responsible for the target concept, thereby incurring minimal impact on other concepts.

### 4.4 Inference-time Control over Forgetting strength

As mentioned before, a significant advantage of our framework is the ability to provide fine-grained continuous control over the strength of forgetting at inference time, allowing users to seamlessly navigate the efficacy-specificity trade-off without any retraining overhead. This control is achieved through two complementary mechanisms: (1) scale of the steering vector $\lambda$ (Equation (2)) and (2) column masking (Equation (7)). Our primary mechanism for control is the learned steering vector $\tau$, which shifts the model's weights to erase a concept. The intensity of this shift can be continuously modulated by $\lambda$: By adjusting $\lambda$ at inference time, a user can smoothly transition from the original model behavior ($\lambda = 0$) to complete concept erasure. We demonstrate this capability qualitatively for NSFW content in Figure 1 (right-panel).

We also leverage column masking (Section 3.2) as a complementary second axis of control. This technique allows us to define the *scope* of the intervention by selecting a specific percentage of the most relevant model parameters to update. To demonstrate this control, we conduct an experiment on the concept of celebrity identity "Brad Pitt", where we progressively increase the percentage of updated columns from 10% to 70%. As illustrated in Figure 6, increasing the column mask percentage strengthens *forgetting efficacy*, successfully leading to a corresponding reduction in the classifier's softmax score for the target identity. However, as expected, this increased efficacy produces collateral effects on specificity: At higher masking percentages, we observe a noticeable degradation in the generated identities of neighboring concepts (e.g., "Tom Hanks" and "Chris Evans"), highlighting the direct trade-off that can be managed through this control. Finally, a joint study analyzing the combined effects of the steering vector scale $\lambda$, and the column masking percentage for the artistic style domain is provided in the supplementary material (Appendix O.2 and Appendix M).

## 5 Targeted Forgetting: Top-K Layer Selection and Column Masking

One of our core hypothesis is that sparse, targeted edits to the model would allow us to preserve its behavior for other concepts. Therefore, to identify these components, we leverage our automated data curation

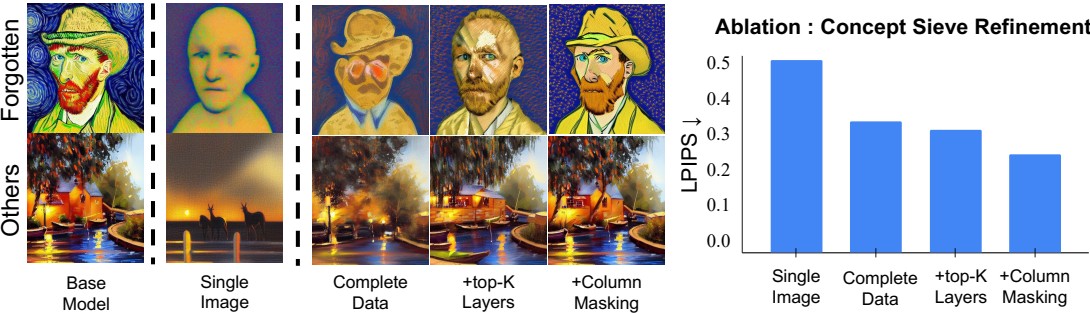

Figure 7: **Concept Sieve Refinement.** First we show result using just one image in $D_c$ and $D_{cn}$, which results in high leakage with good forgetting. Then we use complete $D_c$ and $D_{cn}$ (our approach), which leads to better preservation. Performing top-K layer selection, followed by column masking (CM) further improves identity preservation and reduces leakage. Notice the re-emergence of identity of the Artist after localizing the weights (last two columns). We also report LPIPS for neighboring concepts in the bar chart on the right.

method to guide the editing process (Section 3.1). While our data curation method does preserve the image semantics well, some distortions still occur. However, we note that these distortions are unique to each image, whereas the concept-related differences remain consistent across the dataset. Therefore, by training on multiple images, we reduce the leakage significantly. We can further stem the leakage by reducing the no. of selected layers while performing top-K layer selection for finetuning, as well as using column masking (CM) (see Stage-II in Figure 3). To demonstrate this, we perform an ablation on these components and show their results in Figure 7. It is evident from the last two columns of Figure 7 that localizing top layers (top-K) and using column masking (CM) further preserves the image structure and content. Our further analysis on the localized layers show that attention layers are more relevant for the concept than other layers (see Appendix F.1).

# 6 Conclusion

Concept Siever presents an effective and flexible way to forget concepts in text-to-image diffusion models without requiring any extra guidance or domain specific knowledge. Through an extensive evaluation, we demonstrate the effectiveness of our method in preserving text fidelity and image quality, reducing concept leakage, and providing an active inference-time control to trade-off specificity with forgetting quality. Concept Siever makes T2I diffusion models easily accessible to a larger audience by giving them control over generative capabilities of stable diffusion by following the goals of safe AI.

# 7 Broader Impact Statement

While our proposed framework effectively removes harmful, explicit or copyrighted content through the concept sieve, we acknowledge its potential for dual use—specifically, the possibility of reversing the sieve to amplify the concepts which were supposed to be removed. This concern, however, is not just unique to our approach but applies broadly to the field of concept forgetting and model editing, and our method also falls within this scope. That said, we advocate responsible use of our technology, and leave the development of techniques to detect whether images have had concepts suppressed or reinforced as future work.

**Acknowledgment:** The authors would like to thank the reviewers and the Action Editor for their valuable feedback which has significantly improved the quality of this manuscript. We are grateful to Joseph KJ and Srikrishna Karanam for their valuable guidance towards conceptualization and development of this work. We also extend our sincere thanks to Kinshuk Vasisht for the helpful discussions and for providing code for identifying concept tokens.

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
