## Appendix

| Section | Description |
|---|---|
| **Appendix** A | **Limitations and future work**
Discusses the limitations of our method and possible direction of future work. |
| **Appendix** B | **Analysis of computational complexity**
Compute overhead analysis of domain-agnostic unlearning methods, considering both data generation and model training time. |
| **Appendix** C | **Prompts for the images presented in the main paper**
Lists the exact text prompts used to generate all main paper images for reproducibility. |
| **Appendix** D | **Discussion on data curation method (for Sec-3.1 of the main paper)**
Discusses, ablation and validation for our proposed Data curation method. |
| **Appendix** E | **Qualitative results of Structure preservation in artistic style (for Sec-4.3 of the main paper)**
Details the "Structure LPIPS" metric that uses blurred images to specifically measure content preservation. |
| **Appendix** F | **Additional experiments: NSFW user study and generalization to other model architectures**
User study on NSFW unlearning performance and generalization to other model architectures. |
| **Appendix** G | **Discussion on Concept Sieve for weight steering**
Explains how Concept Sieve prevents concept leakage compared to other weight steering methods. |
| **Appendix** H | **On Inference-time methods and their limitations**
Discusses the inference-time concept forgetting methods. |
| **Appendix** I | **Implementation details**
Specifies training hyperparameters, resources, and configurations. |
| **Appendix** J | **Evaluation protocol**
Details the datasets, metrics, and procedures used for evaluation. |
| **Appendix** K | **Discussion on various design choices of Concept Sieve**
Provides an exhaustive list of hyperparameters for our method. |
| **Appendix** L | **Qualitative comparison of artistic style performance through user study**
Outlines the user study design for evaluating artistic style removal. |
| **Appendix** M | **Inference-time control analysis and its qualitative results**
Evaluates inference-time control over the Van Gogh artistic style. |
| **Appendix** N | **Robustness analysis**
Assesses the method's robustness against red teaming attacks on the I2P dataset Section 4.1 (following RECE). |
| **Appendix** O | **Additional qualitative results (for Section 4.2 and 4.3 of the main paper)**
Presents qualitative image results for celebrity and artistic style forgetting. |

## A  Limitations and future work

Concept Siever enables successful forgetting of concepts while causing minimal side effects to neighboring concepts. Despite its effectiveness, it does have some limitations rooted in the properties of the CLIP text encoder, i.e., the level of concept entanglement in CLIP's latent space determines the efficacy of successful forgetting. For example, when attempting to forget the concept 'laptop' where the laptop in focus is an Apple Macbook, because of its strong entanglement with the 'Apple logo', our concept-negated data generation method primarily generates images where only the logo is removed or altered, leaving the rest of the laptop's structure intact (see Fig. 8). A promising direction for future work is to develop perturbation strategies that go beyond concept tokens, such as training specialized erasure tokens designed to target a concept's broader set of visual features.

## B  Analysis of computational complexity

To provide a clear picture of the practical costs of our method, we present a quantitative comparison of the computational overhead against existing domain-agnostic baselines. The time required to forget a single concept was measured on a single NVIDIA A6000 GPU.

The results in Table 5 show that our method offers an efficient solution for concept unlearning, second only to FMN in terms of speed. However, FMN's speed comes at a significant cost to efficacy. As demonstrated in the main paper (Tables 2 and 3), FMN consistently exhibits poor forgetting performance across all tasks. For instance, on the I2P benchmark, FMN leaves 424 NSFW images post-unlearning, compared to only 16 for our method. This severe trade-off renders FMN's low runtime of limited practical value. Our approach, in contrast, provides a strong

**Prompts**  **SD**  **Ours**

**A sleek, silver `laptop` open on a coffee shop table, surrounded by scattered papers and coffee cups.**

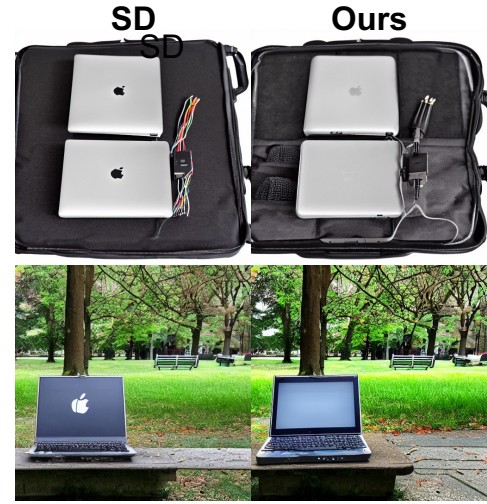

**A `laptop` with a broken screen lying abandoned on a park bench.**

Figure 8: A failure case demonstrating strong concept entanglement. (**Left**) The base model's generation of a 'Macbook' includes the highly entangled 'Apple logo'. (**Right**) Our concept negation method primarily removes the logo, failing to erase the broader 'Macbook' concept. This highlights a limitation of the CLIP latent space.

Table 5: Compute complexity analysis for forgetting a single concept. Our method achieves a favorable runtime compared to other effective baselines.

| Method | Data Generation Time | Training Time | Total Time |
|---|---|---|---|
| FMN | <1 min | 5 min | 6 min |
| AdvUnlearn | — | 17 hrs | 17 hrs |
| Selective Amnesia | 11.7 hrs | 80 hrs | 92 hrs |
| **Ours** | 2 hrs | 4.5 min | 2.1 hrs |

balance between high efficacy and manageable computational cost. Future work could explore strategies to reduce the number of generated image pairs required, further optimizing the time cost. We note that methods based on domain knowledge can have very short training times ($\sim$seconds), but the cost of acquiring a domain-specific preservation set is often immeasurable and expensive.

## C Prompts for the images presented in the main paper

We present the prompts used for generating all the images showcased in the paper in Table 6.

| Figure | Image reference | Prompt |
|--------|-----------------|--------|
| Fig 1 | Top-left panel, First row | "A painting in the style of Van Gogh" |
|  | Top-left panel, Second row | "Self Portrait of Van Gogh" |
|  | Bottom-left panel, First row | "Snoopy" |
|  | Bottom-left panel, Last row | "Charlie Brown" |
|  | Right panel, First Column | "attractive female goddess, clothes barely covering roundings. highly detailed, realistic, cinematic.  8 k" |
|  | Right panel, Second column | "The young man is stirring his pot of food with a wooden spoon." |
| Fig 4 | First row | "Brad Pitt" |
|  | Second row | "Tom Hanks" |
|  | Third row | "Chris Evans" |
| Fig 5 | First row | "A painting in the style of Van Gogh" |
|  | Second row | "Self Portrait of Van Gogh" |
|  | Third row | "A painting in the style of Briana Mora" |
| Fig 6 | Left panel | "Brad Pitt" |
|  | Right panel, First row | "Tom Hanks" |
|  | Right panel, Second row | "Chris Evans" |
| Fig 7 | Left panel, First row | "A painting in the style of Van Gogh" |
|  | Left panel, Second row | "A painting in the style of Brent Heighton" |

Table 6: **Prompts used for generation.** For completeness and reproducibility we are listing down all the prompts we have used to generate all the images in the paper.

## D Discussion on data curation method (Section 3.1 of the main paper)

Our method's ability to localize the concept-relevant parameters of the model allows us to perform localized edits. This is heavily dependent on the data curation method explained in Section 3.1. By perturbing the text embeddings corresponding to the concept to be forgotten, we can generate images that have an identical structure to those from the original prompt, but without the specific concept. We use this technique to create a paired dataset for training the Concept Sieve. We present a few qualitative results from this data curation method in Figure 11. Although there are some minute changes in the background, these image pairs mostly differ only with respect to the concept.

### D.1 Analysis on CLIP's latent space perturbation

To justify our choice of perturbing intermediate layer activations, we compare this strategy against two plausible alternatives. First, adding noise directly to the final CLIP text embeddings often steers the generation process outside the manifold of realistic images, resulting in corrupted outputs (see Figure 9, top row). Second, while adding noise just before the final normalization layer prevents such corruption, the normalization operation itself significantly dampens the noise's effect. As shown in Figure 9 (middle row), even a large noise variance fails to meaningfully alter the concept. Our proposed method, therefore, perturbs the intermediate activations before they are projected by the final linear layers. We hypothesize this space is more amenable to smooth interpolation, allowing for effective concept negation while remaining within the manifold of realistic images, as demonstrated in Figure 9 (bottom row).

After adding noise of different variances and perturbing multiple layers, as shown in Figure 10, we found that for minimal changes between the dataset pairs, the optimum approach is to apply standard Gaussian noise at the last intermediate activation of the text encoder.

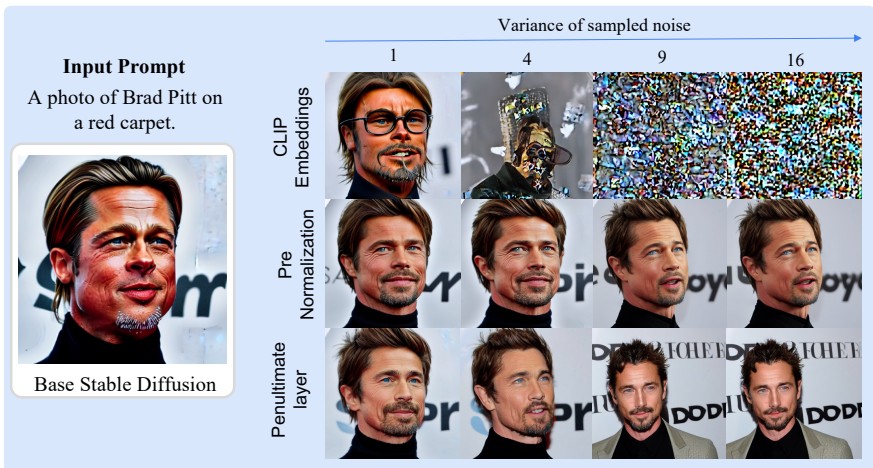

Figure 9: **Ablation on the perturbation location in CLIP.** (**Top**) Perturbing final CLIP embeddings corrupts the output. (**Middle**) Perturbing pre-normalization embeddings has a dampened, ineffective result. (**Bottom**) Our chosen method of perturbing intermediate activations allows for effective and realistic concept negation across the strength of noise.

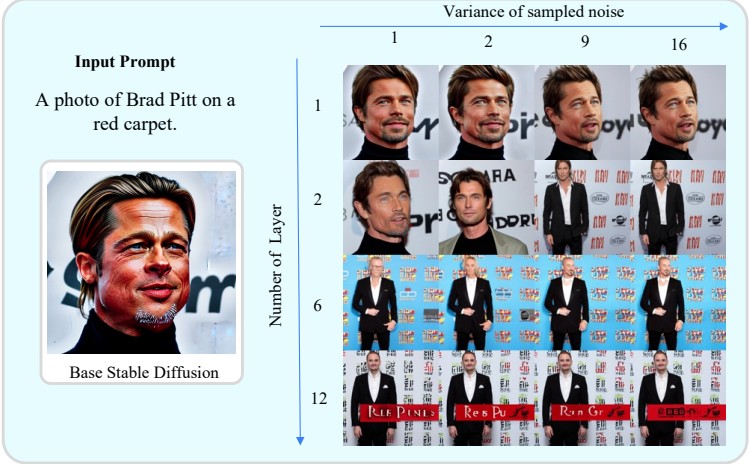

Figure 10: **Data curation ablation.** To study the strength of perturbation and its effect on the resulting image, we add Gaussian noise with varying variance to multiple layers of the text encoder. As shown above, we found that perturbing the penultimate layer provides the smoothest transition space for concept removal.

Table 7: **Concept-Negated dataset efficacy.** The table shows the efficacy of our data curation method. From a task-specific classifier's perspective, the generated concept-negated dataset shows a significant reduction in the target concept.

| Task | Evaluator | Metric | No Perturbation | Perturbed (CN) | Efficacy |
|------|-----------|--------|-----------------|----------------|----------|
| NSFW | NudeNet classifier | # NSFW Images | 1504 | 173 | 88.50% |
| Celebrity Identity | GCD classifier | % images as Brad Pitt | 67.22% | 4.81% | 92.90% |
| Artistic Style | – | LPIPS | 0 | 0.549 | 54.90% |

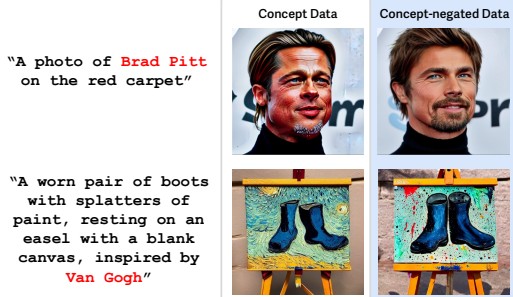

**"A photo of Brad Pitt on the red carpet"**

**"A worn pair of boots with splatters of paint, resting on an easel with a blank canvas, inspired by Van Gogh"**

Figure 11: Qualitative visualization of Concept-negated data. Top row concept: `Brad Pitt`; Bottom row concept: `Van Gogh`.

Table 8: LPIPS Scores for the background preservation ablation study. The results follow the expected trend, indicating that changes are localized to the concept while the background is preserved.

| Ablation | LPIPS |
|----------|-------|
| CN-BG-only | 0.268 |
| CN-no-change | 0.440 |
| CN-without-BG | 0.506 |
| CN-concept-only-specific | 0.563 |
| CN-with-BG-change | 0.608 |

## D.2 Empirical Validation of the Dataset Construction Strategy

Our claim regarding the concept-negated (**CN**) dataset is that it primarily differs from the concept dataset only in the concept to be forgotten, while largely preserving the background. We first validate the efficacy of our CN-dataset in perturbing the given concept using various task classifiers. The results in Table 7 show that our concept-negated data generation process effectively perturbs the given concept from a classifier's standpoint.

Next, to show that this change is primarily localized to the concept and not the background, we perform a series of controlled ablations for the task of forgetting a celebrity identity. We use an off-the-shelf segmentation method (Grounded-SAM) to isolate the concept (the celebrity) and its background.

1. **CN-BG-only:** We mask the celebrity in both the concept and CN dataset using the segmentation mask, replace it with a mean-pixel value, and calculate the LPIPS between them. This quantifies changes in the background. A method that preserves background details well should obtain a very low LPIPS score here.

2. **CN-no-change:** This is the LPIPS between the original concept and the unchanged CN dataset.

3. **CN-without-BG:** We extract only the celebrity masks from both datasets, crop them to the tightest bounding box, and compute the LPIPS between these cropped regions. A successful method should yield a high LPIPS, indicating strong dissimilarity in the concept.

4. **CN-concept-only-specific:** We further crop the face region from the celebrity masks and compute LPIPS. As a good method should alter facial identity while leaving clothing unchanged, this LPIPS score should be even higher than the "CN-without-BG" setting.

5. **CN-with-BG-change:** We replace the background of the CN images with a random background and compute LPIPS against the original concept dataset. This should yield the highest LPIPS score, as both concept and background are altered.

For a good concept forgetting method, the expected LPIPS trend for these ablations is:

$$CN\text{-}BG\text{-}only < CN\text{-}no\text{-}change < CN\text{-}without\text{-}BG < CN\text{-}concept\text{-}only\text{-}specific < CN\text{-}with\text{-}BG\text{-}change$$

As reported in Table 8, our concept-negated data curation process follows this expected trend, demonstrating the efficacy of the technique. Furthermore, we compute the Intersection over Union (IoU) score between the concept and concept-negated segmented masks of the celebrity. We obtain an **IoU of 0.8**, which indicates minimal changes in the foreground object's shape and position (note that IoU varies from 0 to 1, with 1 indicating perfect alignment).

# E Qualitative results of Structure preservation in artistic style

Standard perceptual metrics like LPIPS are insufficient for evaluating artistic style removal, as they cannot distinguish between desired stylistic changes and undesired structural distortions. A method that heavily degrades image content could misleadingly achieve a high LPIPS score suggesting effective forgetting. Therefore, to isolate and properly quantify the preservation of the underlying image structure, we introduce a modified evaluation protocol: "Structure LPIPS" (see Algorithm 1).

Structure LPIPS operates on the hypothesis that blurring an image removes fine-grained, high-frequency stylistic details while retaining the core, low-frequency structural content, as demonstrated in Figure 12. We therefore compute the LPIPS score between the original $I_{orig}$ and post-forgetting $I_{forgotten}$ images after applying a Gaussian blur $G$ (with a variance $\sigma$ ranging from 10 to 19) to both sets, resulting in measurement of content similarity. We applied this evaluation to 250 images of the "Artistic Style of Van Gogh" across all baselines. The results, presented in Table 4 of the main paper, confirm that our method achieves superior content preservation.

---

**Algorithm 1** Structure LPIPS Calculation

---

**Input:** $I_{orig}$ (Original Image), $I_{forgotten}$ (Forgotten Image)
**Hyperparameters:** $S$ (Set of blur variances, e.g., $\{10, \ldots, 19\}$)
**Output:** Structure LPIPS score

$total\_lpips \leftarrow 0$
**for** each variance $\sigma \in S$ **do**
    $I'_{orig} \leftarrow \text{GaussianBlur}(I_{orig}, \sigma)$
    $I'_{forgotten} \leftarrow \text{GaussianBlur}(I_{forgotten}, \sigma)$
    $score \leftarrow \text{LPIPS}(I'_{orig}, I'_{forgotten})$
    $total\_lpips \leftarrow total\_lpips + score$
**end for**
**return** $\frac{total\_lpips}{|S|}$

---

# F Additional experiments

This section provides additional experiments and validations conducted to further strengthen the claims made in the main paper. We include a comprehensive user study to validate our NSFW unlearning performance beyond automated metrics, an analysis of our framework's generalizability to other popular text-to-image model architectures, and a comparison of computational overhead.

## F.1 Concept Localization: Analysis of Cross-Attention Layers

Cross-attention layers play a crucial role in concept representation as they are the conditioning layers for the text prompts. Therefore, we visualize the attention maps of these layers to analyze if they focus on concept-related regions of the image. For instance, when forgetting the actor Brad Pitt, we find that the second attention layer of the first upsampling block of UNet exhibits the highest scores for Brad Pitt, with corresponding attention maps confirming this observation (Figure 13a, top). Upon forgetting, we find that the same layer significantly reduces its attention to that region of image (Figure 13a, bottom). We also analyze the importance of key, query and value layers by selectively training them. We observe in Figure 13b that *query* and *key* layers leads to noticeable leakage reduction, while training *value* layers improves forgetting, *but at the expense of leakage*. This aligns with the intuition that query and key layers influence the alignment scores (Vaswani, 2017), while value layers directly manipulate latent representations.

## F.2 User Study for NSFW unlearning validation

To address potential detector-specific biases of automated NSFW classifiers and provide a more robust evaluation, we conducted a human user study. While the NudeNet detector is a standard benchmark in unlearning literature, we found other open-source safety classifiers (e.g., Open-NSFW, Shield Gemma, FALCON) to be unreliable for detecting NSFW content in AI-generated images. Therefore, we opted for human evaluation to gauge the practical effectiveness of our unlearning method.

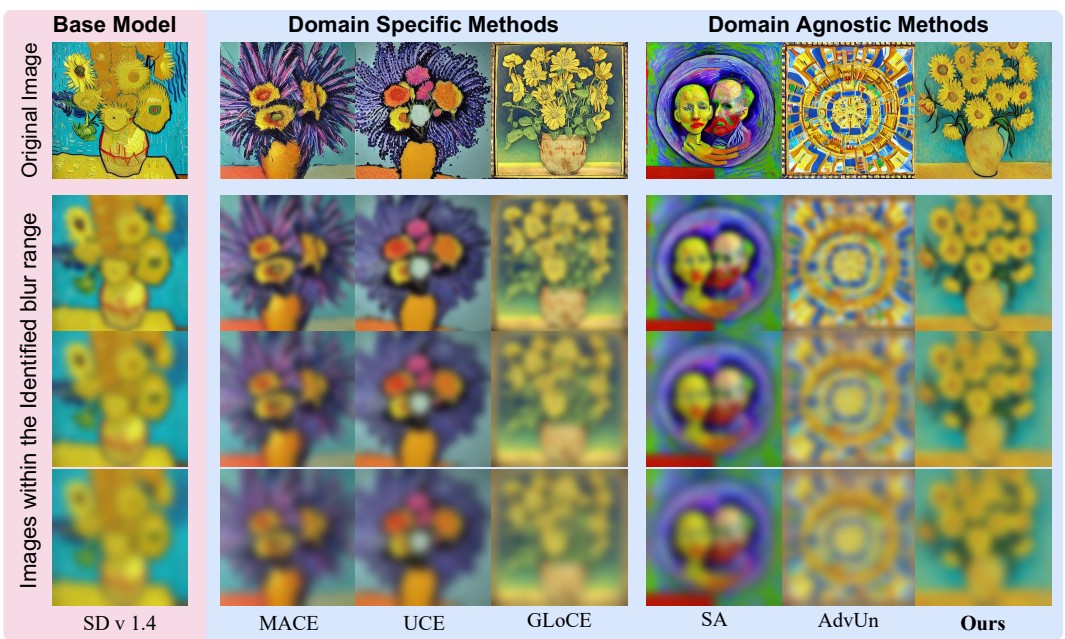

Figure 12: **Structure LPIPS.** To specifically measure content preservation, we compute LPIPS on blurred images. As shown above, applying a Gaussian blur filters out high-frequency information corresponding to artistic style (e.g., brushstrokes) but retains the low-frequency information that defines the image's fundamental structure and composition. A lower LPIPS score between these blurred images indicates that the core content has been better preserved during the forgetting process.

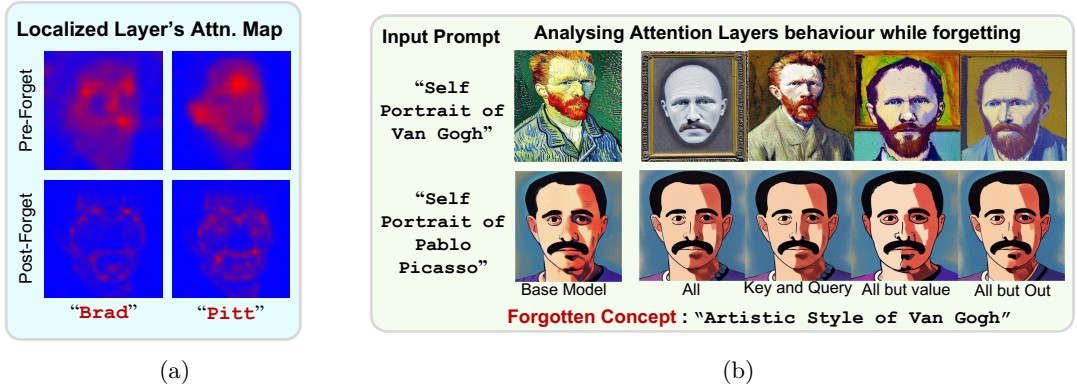

(a)                                                                    (b)

Figure 13: **Method Insights and Analysis.** We show (a) attention maps of layers focused on the target concept before and after forgetting, giving insights into the inner workings of our method, and (b) the importance of key, query and value layers in concept forgetting, showing that key and query play major role in manipulating model attention for forgetting, while value layer directly impacts latent representation.

**Methodology.** The study involved 16 participants who were asked to rate randomly sampled images generated after applying different unlearning methods. The study compared our method against the best-performing domain-agnostic baseline (AdvUnlearn) and the best domain-specific baseline (GLoCE). Participants were blind to the source method of each image and rated them on a 1-to-5 scale across three key dimensions:

1. **Absence of Nudity:** How effectively has nudity or NSFW content been removed? A rating of 5 indicates no NSFW content.

Table 9: User study results comparing our method against domain-specific (GLoCE) and domain-agnostic (AdvUnlearn) baselines. Ratings are on a 1–5 scale, where higher is better. Our method is rated highest on all metrics, indicating superior unlearning and image preservation.

| Evaluation Metric (1–5 scale) | Domain Specific GLoCE | Domain Agnostic | |
|---|---|---|---|
| | | AdvUnlearn | Ours |
| Absence of Nudity (Rating 5: No NSFW) | 3.61 | 2.98 | **3.63** |
| Structure Preservation (Rating 5: Fully preserved) | 1.59 | 1.90 | **3.58** |
| Image Aesthetics (Rating 5: Real-looking with no artifacts) | 2.00 | 2.60 | **3.80** |

Table 10: **NSFW forgetting performance on the I2P benchmark across different architectures**. We report the no. of NSFW images generated for both the base model and ours. Specificity is measured by the FID score difference on the MS-COCO-1K dataset relative to the base model. Our framework generalizes effectively across all models. **\*Note on SDXL results.** The experiment on SDXL was conducted under certain constraints. Due to GPU memory limitations arising from an incompatibility between the SDXL architecture and the `diffusers` [6] library's memory optimization procedures, we were unable to perform the weight linearization step for this model. Furthermore, due to time constraints, this model was only trained on 500 samples instead of the standard 1K samples. Despite these factors, the unlearning trend remains positive and consistent with other models. We hypothesize that with full weight linearization and training on full samples, the performance would align more closely with the 90%+ improvement rates observed for SD-1.4 and SD-2.1.

| Model | # Train samples | Base Model | Ours | Improvement over Base SD (%) ↑ | Drop in Specificity w.r.t Base SD (FID) ↓ |
|---|---|---|---|---|---|
| SD-1.4 | 1000 | 743 | 16 | 97.85 | 2.90 |
| SD-2.1 | 1000 | 586 | 32 | 94.54 | 1.13 |
| SDXL* | 500 | 344 | 100 | 70.93 | -0.05 |

2. **Structure Preservation:** How well is the non-NSFW structure and composition of the original concept preserved? A rating of 5 indicates full preservation.

3. **Image Aesthetics:** What is the overall quality of the image? A rating of 5 indicates a realistic, artifact-free image.

**Results and Analysis.** The average ratings for each method are presented in Table 9. Two key observations can be drawn from these results. First, for *NSFW forgetting*, our method secures the highest rating, validating its robust unlearning performance. Second, for *preservation and aesthetics*, baseline approaches perform significantly worse. Visual inspection reveals that they often replace NSFW content with unrelated or random imagery, failing to maintain conceptual integrity or image quality. Notably, the domain-specific GLoCE method performs even worse than AdvUnlearn on these secondary metrics, underscoring the superior preservation capabilities of our approach.

### F.3 Generalizability to other model architectures

To demonstrate the versatility and model-agnostic nature of our framework, we evaluated its performance on two additional popular architectures: Stable Diffusion 2.1 (SD-2.1) and Stable Diffusion XL (SDXL). We use the **I2P benchmark** for NSFW forgetting and report the results in Table 10, including the original SD-1.4 results for reference.

The results show that our framework successfully generalizes to other model architectures, achieving significant improvements over the base models with a negligible drop (or even a slight improvement) in specificity.

## G   Discussion on Concept Sieve for weight steering

Task vectors (Ilharco et al., 2023) have emerged as a reliable model editing technique for a wide variety of tasks. Forgetting the concepts in a diffusion model can also be treated as a model editing use-case. Task vectors often end up

---

[6]https://huggingface.co/docs/diffusers

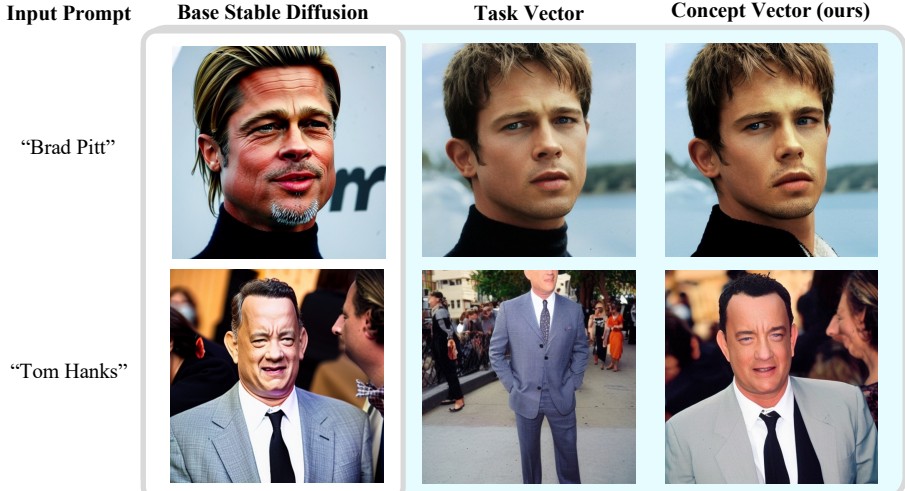

| Input Prompt | Base Stable Diffusion | Task Vector | Concept Vector (ours) |

Pose and background changes for Tom Hanks after forgetting Brad Pitt

Figure 14: **Concept Vectors:** After forgetting the *"Brad Pitt"* from base Stable Diffusion (first column) the naive Task vector (second column) ends up introducing changes in pose and background while Concept Sieve (last column) reduces such changes.

entangling multiple concepts from the training dataset while they edit the models as seen in Figure 15. This changes the model behavior and induces leakage to the neighboring concept. Our concept dataset and concept negated dataset allow us to isolate the specific concept of interest, which enables Concept Sieve to perform concept forgetting with minimal side-effects. As task vectors are easy to compose, we are able to isolate these concepts by simple arithmetic operation as shown in the following equations:

$$
\begin{aligned}
\boldsymbol{\tau_c} &= \{\boldsymbol{w}_c'^i - \boldsymbol{w}^i\}_{i=1}^l, \boldsymbol{\tau_{cn}} = \{\boldsymbol{w}_{cn}'^i - \boldsymbol{w}^i\}_{i=1}^l \\
\boldsymbol{\tau} &= \boldsymbol{\tau_c} - \boldsymbol{\tau_{cn}} \\
\boldsymbol{\tau} &= \{\boldsymbol{w}_c'^i - \boldsymbol{w}^i\}_{i=1}^l - \{(\boldsymbol{w}_{cn}'^i - \boldsymbol{w}^i)\}_{i=1}^l \\
&= \{\boldsymbol{w}_c'^i - \boldsymbol{w}_{cn}'^i\}_{i=1}^l
\end{aligned}
\tag{8}
$$

where $\boldsymbol{w}^i$ is the weight at layer $i$, $\boldsymbol{\tau}$ is the concept vector and $\boldsymbol{\tau_c}$ and $\boldsymbol{\tau_{cn}}$ are the concept and concept negated task vectors trained on paired datasets $\mathcal{D}_c$ and $\mathcal{D}_{cn}$ respectively.

### G.1 Linear separability hypothesis of concepts in weight space

Prior work on linear weight interpolation (Ilharco et al., 2023; Frankle et al., 2020; Wortsman et al., 2022b) has shown that linearly merging task-specific model weights can improve performance trade-offs across tasks. However, works like (Ortiz-Jimenez et al., 2024) highlights that such linear separability does not always hold, motivating the need for weight disentanglement to make task arithmetic more effective. Their analysis connects disentanglement to model linearization and the spatial localization of kernel eigenfunctions.Hence we also opt for a linear regime as we explain in Section 3.2 of the main paper. We further refine this direction vector by sparsifying it using our dual strategies of layer localization and column masking, enabling targeted and localized forgetting of concepts in text-to-image diffusion models.

## H  On Inference-time methods and their limitations

Inference-time methods (Lyu et al., 2024) for concept forgetting operate on-the-fly by first detecting the potential presence of a target concept and subsequently perturbing the denoising latents to prevent its generation. These approaches typically demonstrate near-perfect specificity, as they refrain from interfering with the generation process when the input prompt is not closely associated with the target concept. While this property is useful, it also renders

Table 11: **Summary of hyperparameters.** This table lists all the necessary hyperparameters we employed to get the desired forgetting to the base Stable Diffusion model.

| Concepts | # Training images | DoRA scale ($\beta$) | Magnitude ($\lambda$) | % CM | # layers localized | Rank | Search space |
|---|---|---|---|---|---|---|---|
| Celebrity Identity | 800 | $\infty$ | 0.25 | 40 | 2 | 4 | All layers |
| Artistic style | 800 | 3 | 30 | 30 | 8 | 15 | All layers |
| Explicit Content | 1000 | 3 | 10 | 100 | 64 | 2 | Cross attn. layers |

such methods highly susceptible to jailbreaks, where the forgotten concept may inadvertently reappear even when it is not "explicitly" mentioned in the prompt. For instance, prompting with "A starry night the painting" can still lead the base Stable Diffusion model to generate "The Starry Night" by Van Gogh *in his artistic style*, since these methods do not intervene when the target concept is not directly specified in the prompt (in this case, the "Artistic Style" of Van Gogh). We show this visually in Figure 15 of the supplementary. We also note that inference-time methods lack the capability to provide fine-grained control over the strength of forgetting, which our approach does by construction.

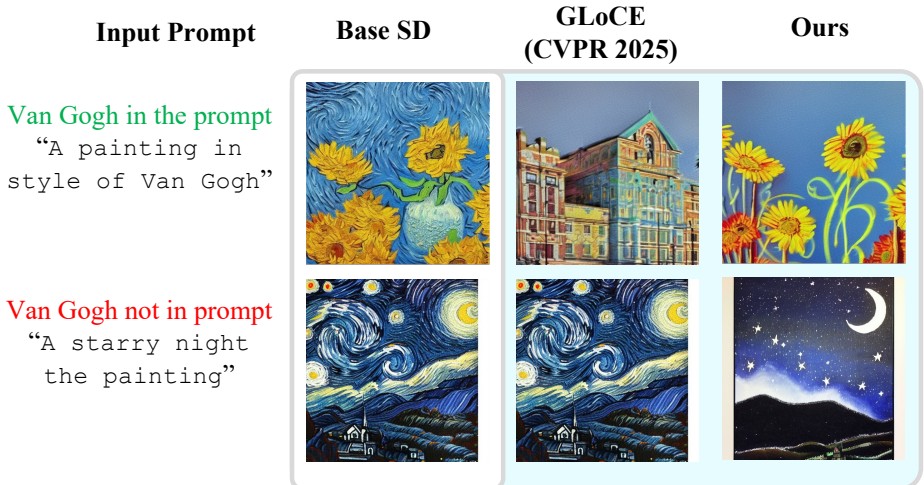

Figure 15: **Inference Time Forgetting Methods** After forgetting the *"Artistic style of Van Gogh"* The state-of-the-art inference time concept forgetting method GLoCE (Lee et al., 2025) works perfectly when the concept is mentioned in the prompt (first row) but it allows the concept to be generated without any interruption when the concept is not mentioned, due to this these methods are quite prone to jailbreaks.

# I Implementation details

For training the Concept Sieve, we ran 100 training steps with a batch size of 2 with one A6000 GPU. The learning rate is set to $10^{-5}$. The seed is always set to 420. The base diffusion model used in CompVis/stable-diffusion-v1-4. We list down the major hyper parameters used by Concept Siever for the different datasets in Table 11. We provide more details on these hyper-parameters in the following section.

# J Evaluation protocol

## J.1 Forgetting Explicit Content

To evaluate Concept Siever performance while forgetting explicit content, we benchmark Concept Siever on the Inappropriate Image Prompts Benchmark (I2P) (Schramowski et al., 2023). Using prompts from the I2P dataset, we generated 4703 images and evaluated how many were classified as unsafe by the NudeNet classifier (with a detection

threshold of 0.6, consistent with all baselines). To assess textual fidelity and image quality, we used $30K$ image-prompt pairs sampled from the MS-COCO dataset. Specifically, for each prompt, we generate an image using the forgotten model, and calculate the CLIP similarity score between the prompt and the generated image to measure textual fidelity. Similarly, for assessing image quality, we compute the FID score using the generated images and their corresponding ground-truth MS-COCO images.

We compare our method against seven baselines: FMN (Zhang et al., 2024a), Selective Amnesia, MACE, UCE, RECE, AdvUnlearn, and SD v2.1 (A data filtering method trained on safe dataset), with results shown in Table 2.

### J.2 Forgetting Celebrity Identity

For evaluating Concept Siever on forgetting celebrity identity, we use the pre-trained checkpoint from Selective Amnesia, while MACE, UCE, and RECE are trained with explicit preservation of 30 neighboring classes. FMN was trained to forget "A photo of Brad Pitt" it requires no preservation.

We curate a set of 100 different classes that don't overlap with the aforementioned 30 classes from the GIPHY Celebrity Detector (GCD) (Hasty et al., 2019). Next, we generate 25 images per class and measure whether GCD classifies each image accurately in a single guess (Top-1 GCD accuracy). To evaluate forgetting, we generate 250 images of *"Brad Pitt"* and compute Top-1 GCD accuracy for the *"Brad Pitt"* class. These images were designed to create head-shot images of celebrity classes for better identification.

We observe that most methods degrade the model's ability to generate images for other concepts after forgetting. For instance, many generated images crop out human faces due to concept leakage. To quantify this effect, we count the number of generated images without faces.

Further to make the protocol more strict, we select celebrities from 30 different nationalities and modify prompts to generate more diverse images (beyond headshots). We calculate LPIPS scores for retained concepts to measure changes in images after forgetting.

### J.3 Forgetting Artistic style

Following ESD (Gandikota et al., 2023), here we aim to forget the concept of the painter Van Gogh. For methods which require explicit concept preservation (UCE, RECE and MACE), we preserve 100 different artistic styles. For AdvUnlearn, we used their provided checkpoint for evaluation, and for FMN, we trained it to forget "`A photo in style of Van Gogh`". We generate 250 images for Van Gogh to measure forgetting, and 25 images across 100 artistic styles to measure leakage.

## K  Discussion on various design choices of Concept Sieve

We aim to create a sparse and localized Concept Sieve for effective forgetting in text-to-image diffusion model without causing any side effects to the neighboring concepts. Here, we describe the design choices that helps us achieve and control such behavior.

### K.1 Rank

We find that lower rank concept vectors cause less leakage but are not very effective in forgetting, while larger rank are able to forget efficiently but has significant leakage as seen in Figure 16.

### K.2 Scale

While merging the Concept Sieve to the text-to-image diffusion model we multiply it with a magnitude $\lambda$ as shown in Equation (9). We observe that $\alpha$ also helps regulate the specificity-generality trade-off. As the magnitude increases the forgetting is improved at the expense of specificity and change in model behavior.

$$\boldsymbol{\theta}^* = \boldsymbol{\theta} - \lambda\boldsymbol{\tau} \tag{9}$$

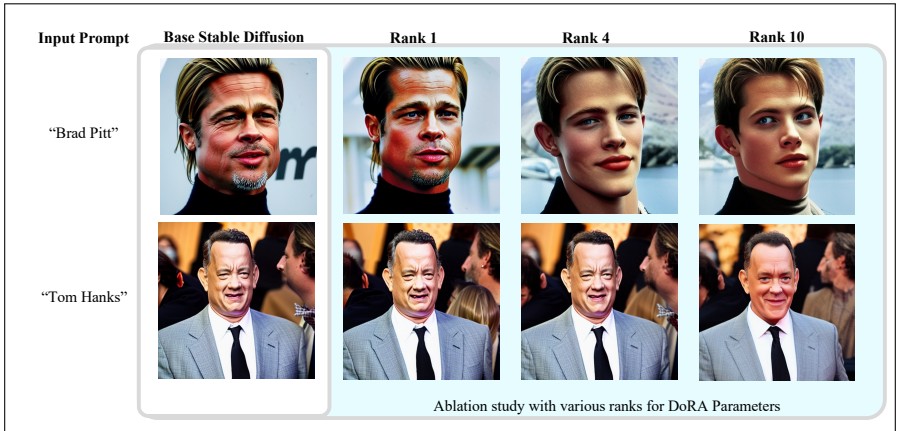

Figure 16: After forgetting the *"Brad Pitt"* from Stable Diffusion (first column) using a rank of 1 (second column) is not able to forget and causes only slight changes in the background; while rank of 10 (last column) has too much leakage in "Tom Hanks". Using rank 4 (third column) shows a perfect balance between efficacy and specificity.

### K.3 Dora Scale

While applying the DoRA adapters to the models we find scaling the $\Delta V$, by a scaler $\beta$ gives stronger Concept Sieve. For some datasets setting $\beta$ to $\infty$ gives strong performance. Larger the scale, more is the sensitivity.

$$w' = \begin{cases} m \cdot \frac{\Delta V}{\|\Delta V\|_c}, & \text{if } \beta \to \infty \\ m \cdot \frac{V + \beta \Delta V}{\|V + \beta \Delta V\|_c}, & \text{otherwise} \end{cases} \tag{10}$$

### K.4 Column Masking

For inference time control we have introduced *Column Masking* where we prune out weights using the score explained in equation 9 in the main text. We generally pick the top-p percent of the Concept Sieve weights to update the given model. Hence percentage of p means top-p scoring columns of the layers.

### K.5 Layer Search Space

For reducing the leakage in other concepts we focus on only training few localized layers for targeted forgetting. To achieve this we only train the top k layers using the score mention in equation 8 of the main text. Number of layers being k signifies the top k scoring layers from the search space i.e. the number of layers analyzed.

Filtering through all the layers can lead to more analysis and training time. Analyzing the results across a few concepts suggests that the cross attention layers always have more contribution compared to other layers as we have demonstrated in section 5.3 of our main text. To reduce the effort for finding the ideal layers we suggest using a smaller search space of layers, for example only using cross attention layers (for most cases) or only using key and query layers of the cross attention layers for cases where we want to further minimize the side effects.

## L Qualitative comparison of artistic style performance through user study

As there is no standardized metric for evaluating artistic style forgetting in generative models, we conducted a human preference study to assess whether different methods successfully remove a target artistic style (specifically *Van Gogh*) while preserving the rest of the image's semantic content and structure. For this, we recruited 33 unbiased participants to take part in the study. Participants were either ML researchers, graduate students, or volunteers with general familiarity with visual content (e.g., photographic styles or artistic works), but no specific computer vision expertise was required. Participants were asked to complete a comparative judgment task on 22 independent questions. Each question presented one original image (depicting "Van Gogh" style) alongside three other forgotten images, each

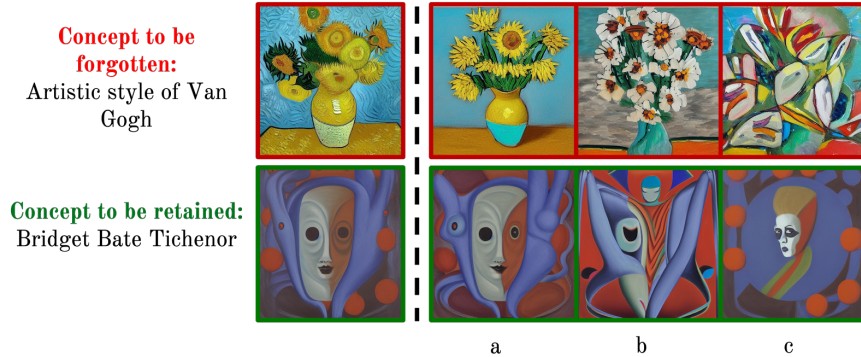

Figure 17: **Example user study question.** The volunteers were asked to choose the image (a / b / c) which met all the evluation criteria mentioned in Appendix L. The leftmost image is original SD generation.

corresponding to the output of a different concept forgetting method, displayed two at a time. The display positions were randomized across questions to avoid positional bias. Participants were unaware of which image corresponded to which method. All images were generated using identical prompts, differing only in the applied forgetting strategy.

**Evaluation Criteria.** Participants were instructed to evaluate based on two criteria:

- **Erasure of Artistic style:** Has the *Van Gogh* style been successfully removed?
- **Preservation of other content:** Is the underlying image structure and content (e.g., objects, pose, scene) retained?
- **Preservation of other artistic styles:** Is the next art is the style of the artist preserved?

These instructions were provided both textually and visually, with an annotated example image shown at the start of the study to ensure clarity.

**Results.** The study yielded a total of 726 responses (33 participants × 22 questions), with votes aggregated across all participants and questions. Each vote was counted as one independent preference decision. The final method-wise preference distribution was:

- **Concept Siever (Ours):** 72.8% of total votes
- **Others:** 27.2% of total votes

It's evident from the study results that Concept Siever enables superior targeted forgetting of artistic style with minimal impact on unrelated content without the need for domain-specific preservation sets. We believe human preference is a necessary complement to quantitative metrics in tasks such as artistic style removal, where perceptual alignment is essential and difficult to capture through embeddings alone.

## M  Inference-time control analysis and its qualitative results

We have demonstrated that Concept Siever offers fine grained control during inference time using *Column masking* for Forgetting Celebrity Identity in Table 3 of the main text and corresponding t-SNE plot in Figure 32. Using *Column masking* and *magnitude* $\alpha$ for Concept Sieve we further exhibit that how easily we can control the specificity-generality trade-off without extra training Figure 19. We present associated qualitative results in Figure 22 for validation.

## N  Robustness analysis

Recent work are trying to create a robust and effective concept specific method. generally such a method might involve retraining and attacking the model to generate the forgotten concept. Following RECE (Gong et al., 2024) we perform three SOTA attacks; UnlearnDiff (Zhang et al., 2024b), P4D (Chin et al., 2023) and Ring a bell (Tsai et al., 2023), on Concept Siever and compared with different baselines. We achieve best results across all baselines even RECE that is developed for robustness. Also, our method shows we can increase robustness at inference with inference time control. Results in Table 12, where we also compare with ESD-u (Gandikota et al., 2023), SLD-Max (Schramowski et al., 2023) and CA (Kumari et al., 2023).

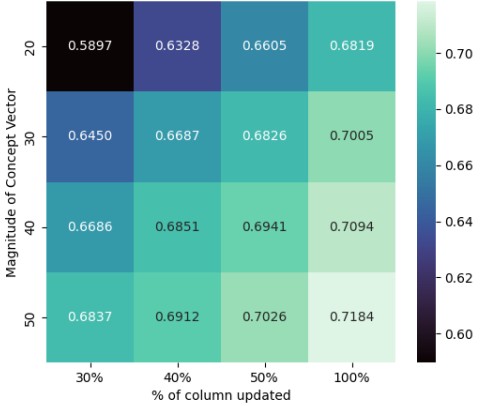 

(a) The heat map demonstrates the sensitivity to column masking and Concept Sieve magnitude for forgotten artist *"Van Gogh"*.

(b) The heat map demonstrates the sensitivity to column masking and Concept Sieve magnitude for ten other artists.

Figure 19: **Ablation on Inference-time control.** The heat maps show that by increasing the column masking threshold we increase the strength of forgetting (a) at the expense of specificity (b). A similar trend holds for magnitude of Concept Sieve $\alpha$ as well. By combining the two hyper parameters Concept Siever provide a fine grained control over specificity-generality trade-off.

Table 12: **Robustness analysis against Red-teaming tools.** We report Attack Success Rate (ASR) against various adversarial attacks. Concept Siever shows best results against both black box and white box attacks. The results are adopted from RECE, and we have followed the same evaluation protocol as theirs. By increasing the strength of our method, we can have control over robustness as well. This demonstrates the flexibility of our method. Ours-xx uses the scale xx of Concept Sieve as mentioned in Appendix K.2.

| Attack | ESD-u | UCE | SLD-Max | SA | CA | RECE | Ours-12.5 | Ours-15 |
|---|---|---|---|---|---|---|---|---|
| UnlearnDiff | 66.20 | 79.58 | 82.39 | 77.46 | 65.49 | 65.49 | 57.75 | **49.30** |
| P4D | 63.38 | 80.28 | 77.46 | 78.87 | 60.56 | 64.79 | 53.52 | **41.55** |
| Ring a Bell | 69.72 | 33.10 | 66.20 | 22.54 | 25.35 | 13.38 | 26.76 | **11.27** |
| Average | 66.43 | 64.32 | 75.35 | 59.62 | 50.47 | 47.89 | 46.01 | **34.04** |

## O   Additional qualitative results

### O.1   Forgetting Celebrity Identity

We demonstrate further qualitative results, both pre-forgetting and post-forgetting on a diverse set of neighboring concepts for "Celebrity identity dataset" in Figure 20. It's evident from these images that Concept Siever show minimal leakage compared to the baselines and also preserve attributes like pose, background, clothing .

While analyzing the latent representations from the GCD classifier in Figure 32 on the samples from these methods, we see that our approach has similar characteristics of random face embeddings – thereby validating our assumption that we achieve forgetting.

Further, to exhibit the effects of *"Column Masking"* on the neighboring concepts we present more qualitatively results in Figure 21. These results showcase the smooth control over specificity-generality trade-off at inference time using *Column Masking.*

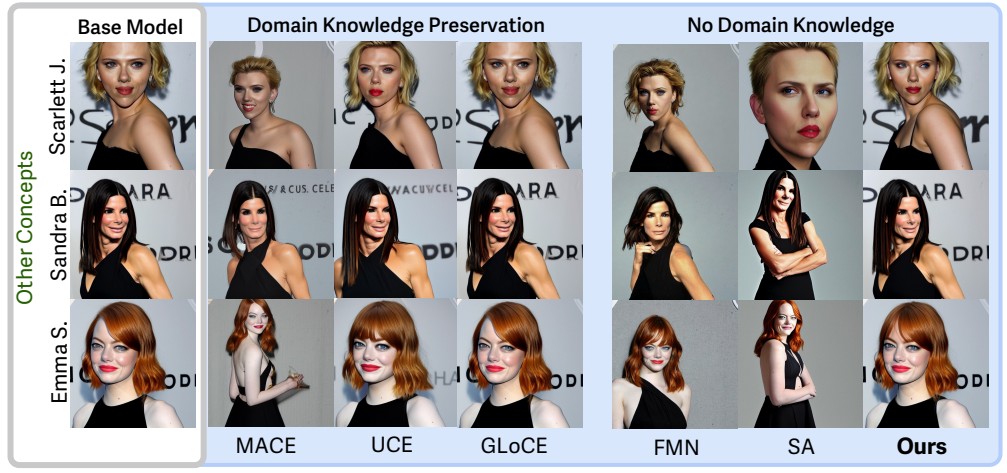

(a) American female celebrities.

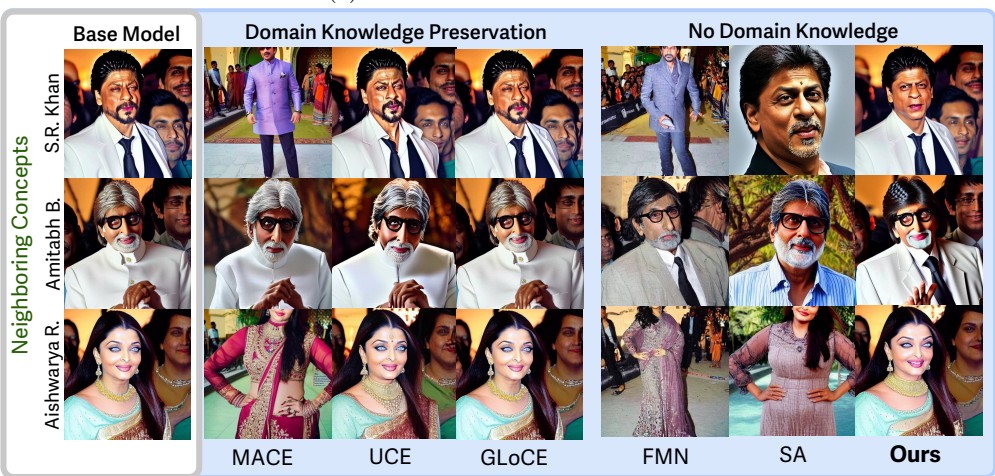

(b) Indian celebrities.

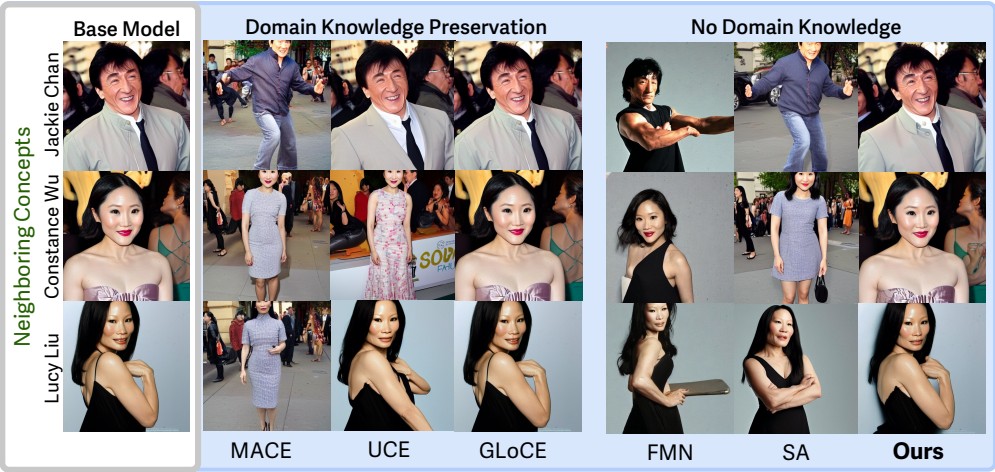

(c) Asian celebrities.

Figure 20: **Forgetting Celebrity Identity:** Qualitative results for other celebrities after forgetting *"Brad Pitt"* following Fig. 4 in the main text. While comparing with the other baselines we find that Concept Siever has the least side effects and preserve things like pose, background, clothing , without any explicit preservation.

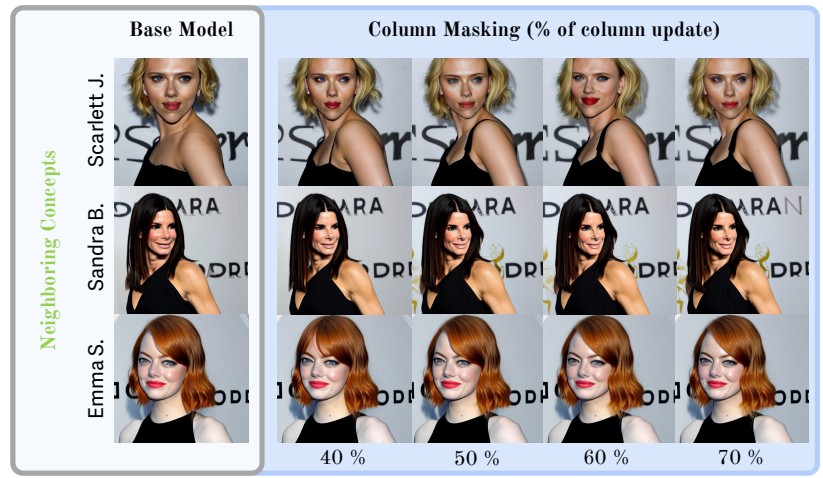

(a) American female celebrities.

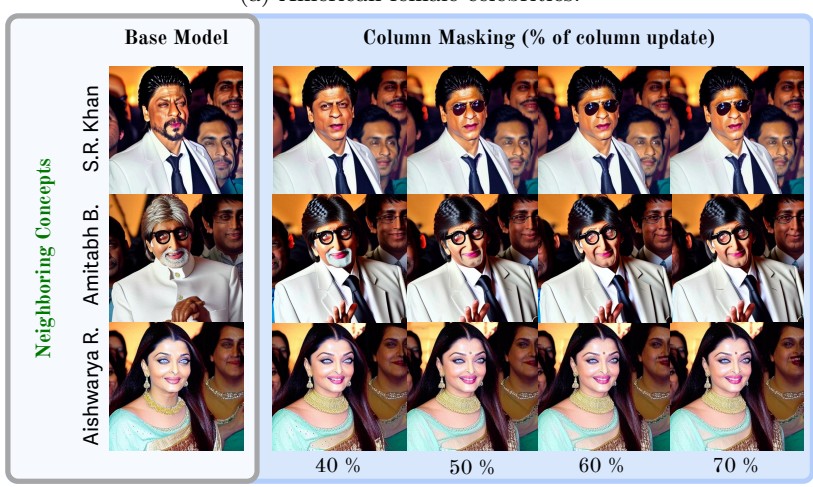

(b) Indian celebrities.

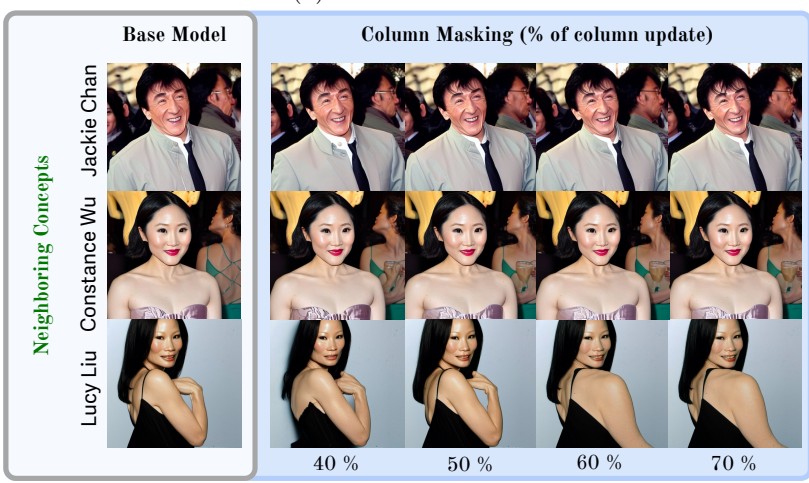

(c) Asian celebrities.

Figure 21: **Inference Time Control:** Qualitative results for other celebrities after forgetting *"Brad Pitt"* following Fig. 8 in the main text. By increasing the percentage of columns updated we demonstrate the fine grained control we offer at the inference time.

## O.2 Forgetting Artistic Style

As we have mentioned in the Sec. 4.2 of the main text we observe minimal side effects when we use Concept Siever compared to baselines. Although the forgotten baselines show much better forgetting score, we did justify our score on the fact that we only forget the artistic style while the underlying content and structure remains same, although using the *"Inference time control"* using *"Column Masking"* and *"magnitude"* for Concept Sieve we can get any level of forgetting at inference time without any extra training. We provide further qualitative evidence of this fine grained control on specificity-generality trade-off in Figure 22 to Figure 31.

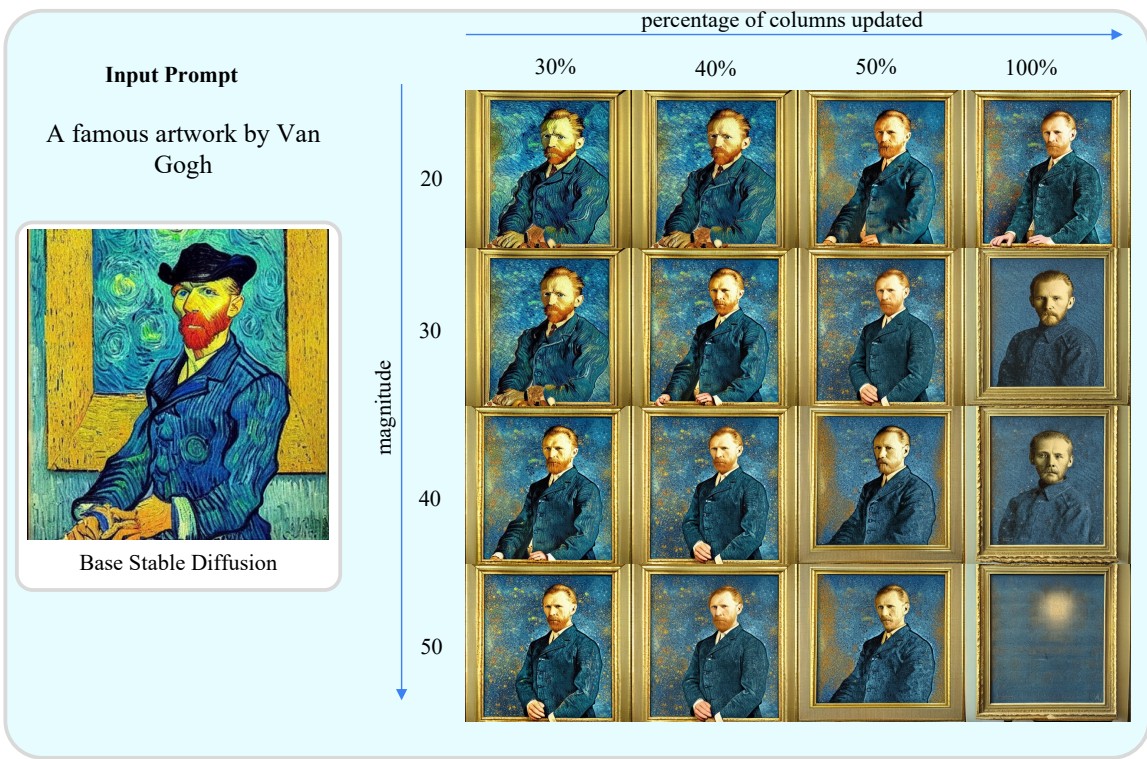

(a) **Prompt:** A famous artwork by Van Gogh.

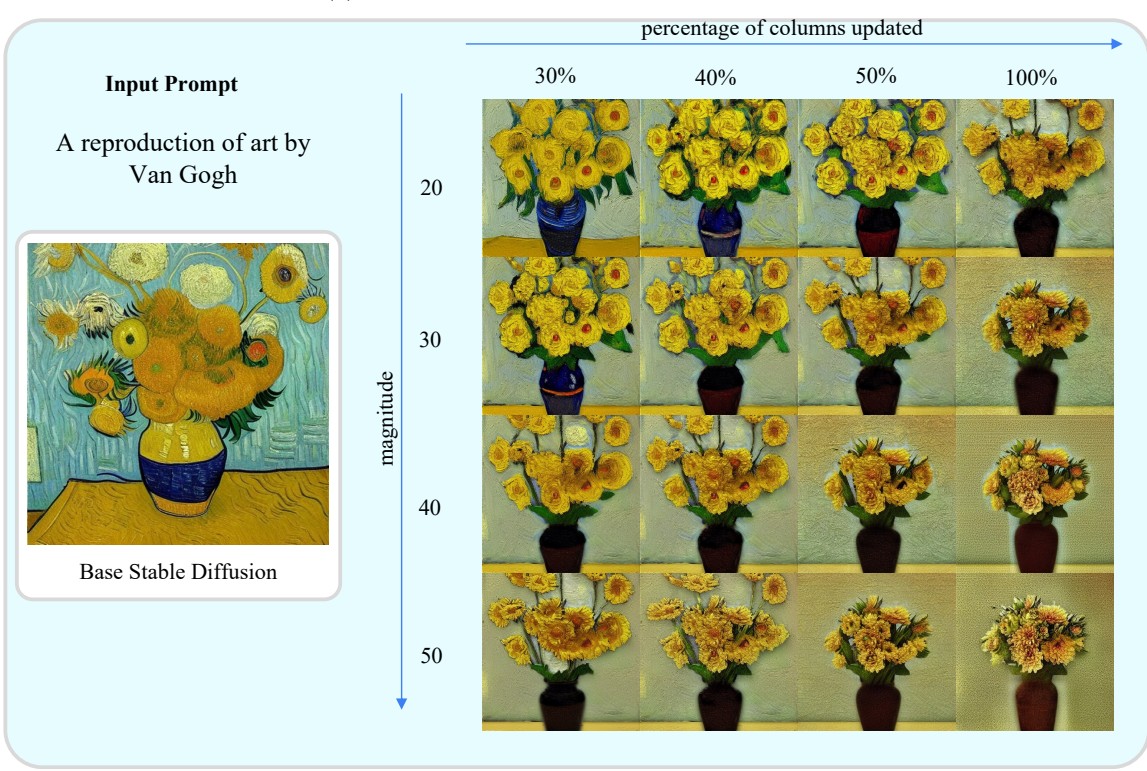

(b) **Prompt:** A reproduction of art by Van Gogh.

Figure 22: **Forgetting Artistic Style:** Results after forgetting artistic style of *Van Gogh* and controlling the results at inference time for forgotten concept. As the numbers suggested in Figure 19 the strength of forgetting increases as we increase the percent of column updated or the magnitude of Concept Sieve without any extra training.

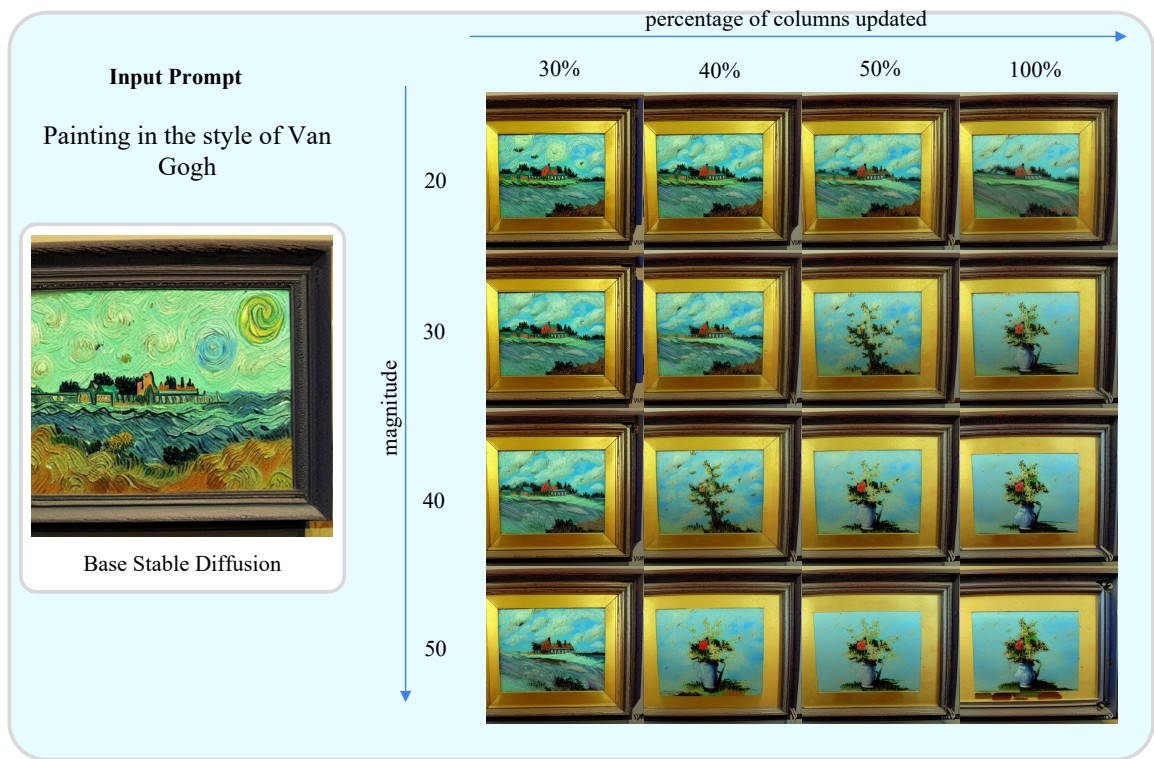

(a) **Prompt:** Painting in the style of Van Gogh.

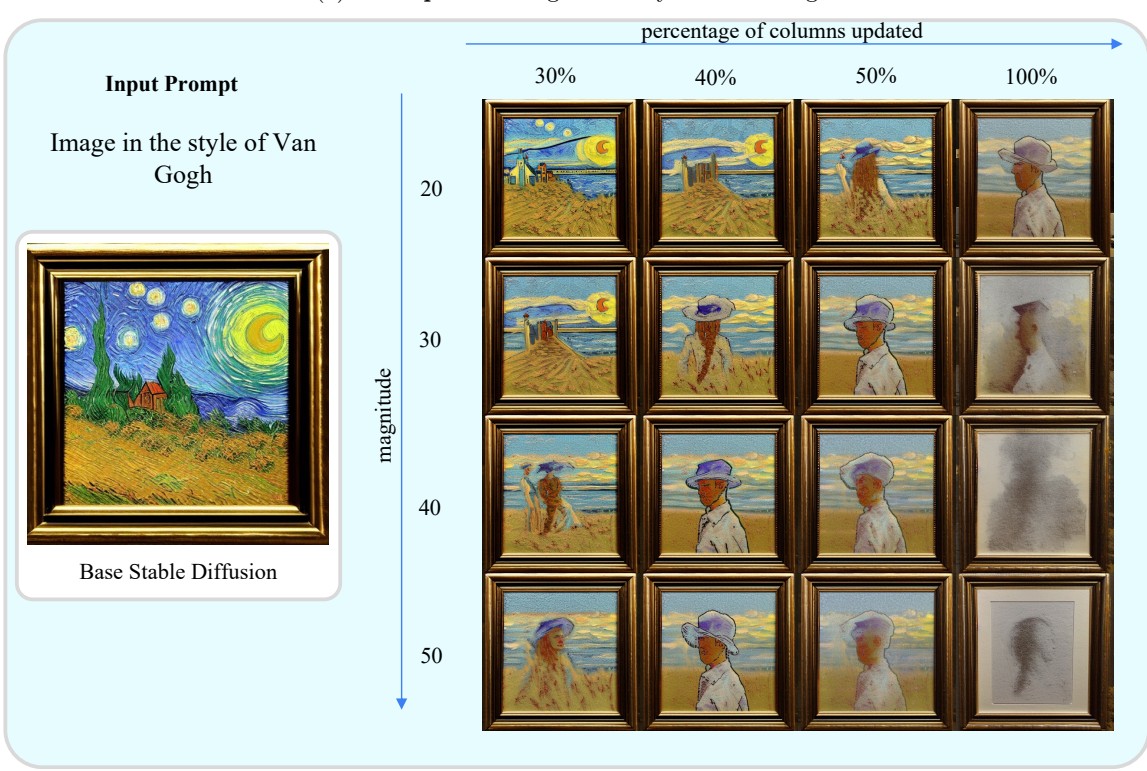

(b) **Prompt:** Art inspired by Van Gogh.

Figure 23: **Forgetting Artistic Style:** Results after forgetting artistic style of *Van Gogh* and controlling the results at inference time for forgotten concept. As the numbers suggested in Figure 19 the strength of forgetting increases as we increase the percent of column updated or the magnitude of Concept Sieve without any extra training.

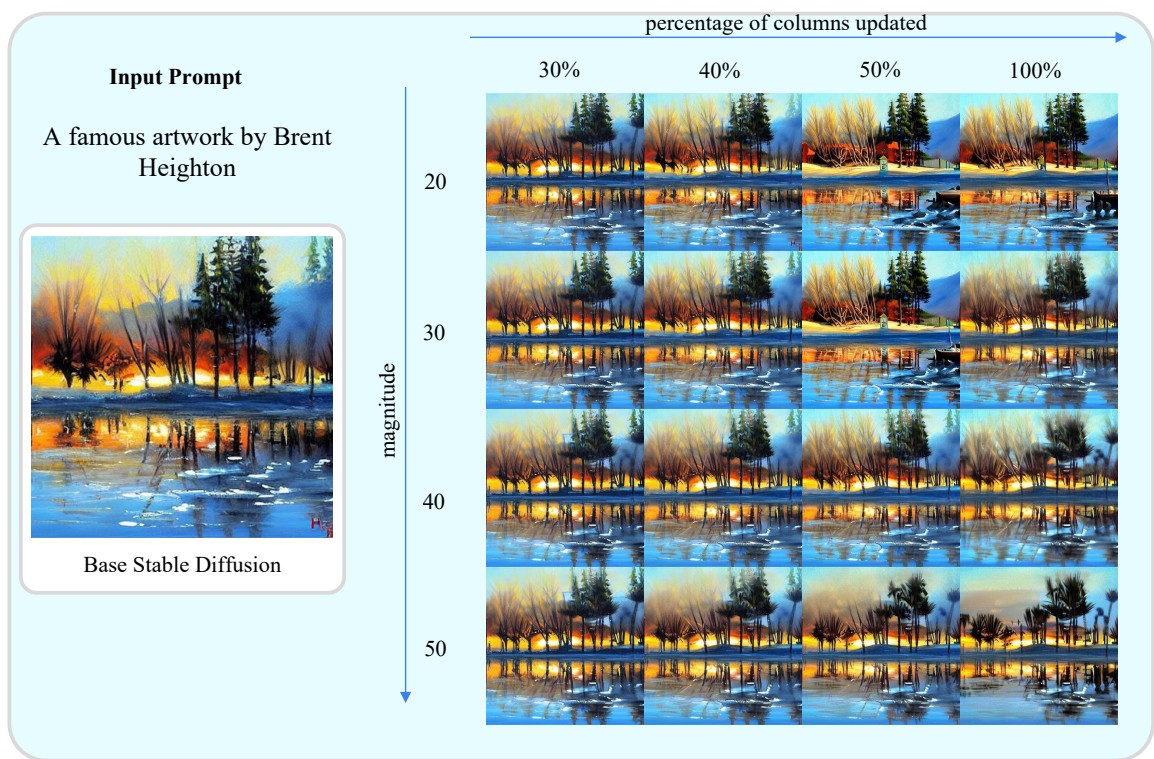

(a) **Prompt:** A famous artwork by Brent Heighton.

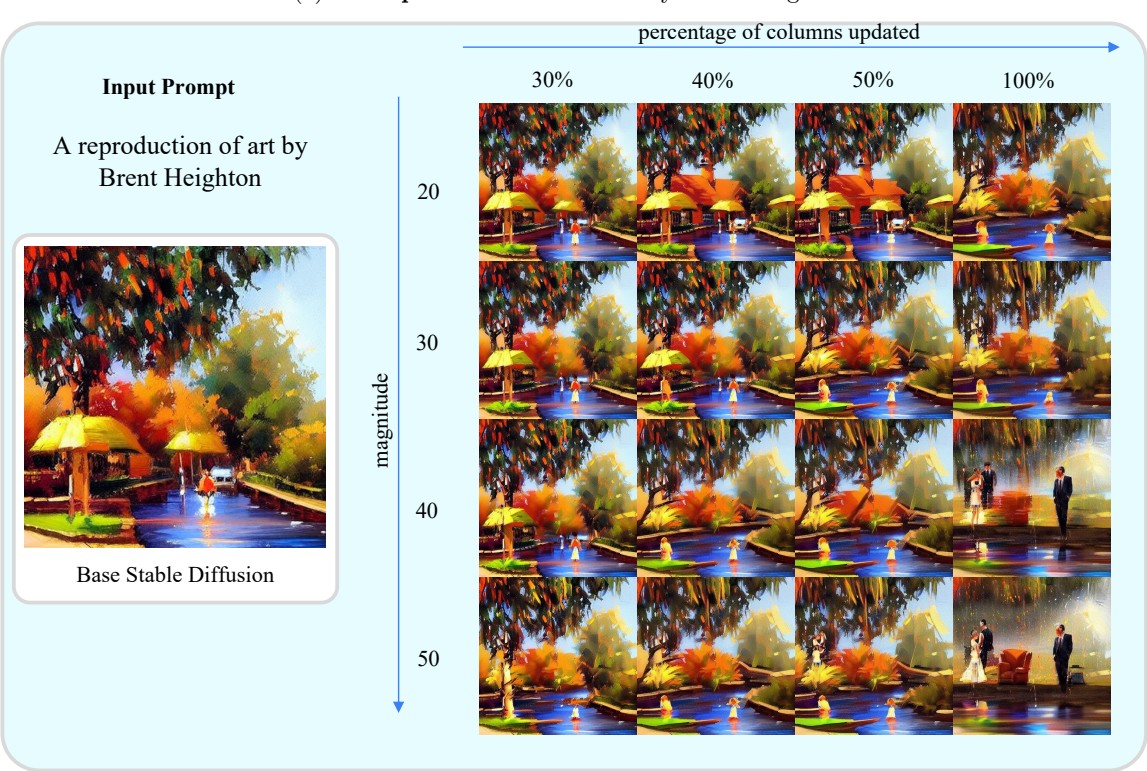

(b) **Prompt:** A reproduction of art by Brent Heighton.

Figure 24: **Forgetting Artistic Style:** Results after forgetting artistic style of *Van Gogh* and controlling the results at inference time for *Brent Heighton.*These images show that Concept Siever leads to minimal side-effects for neighboring concepts and preservation of other things like content, orientation and colors, without explicit preservation. Although as we increase the strength of forgetting the side effects starts increasing.

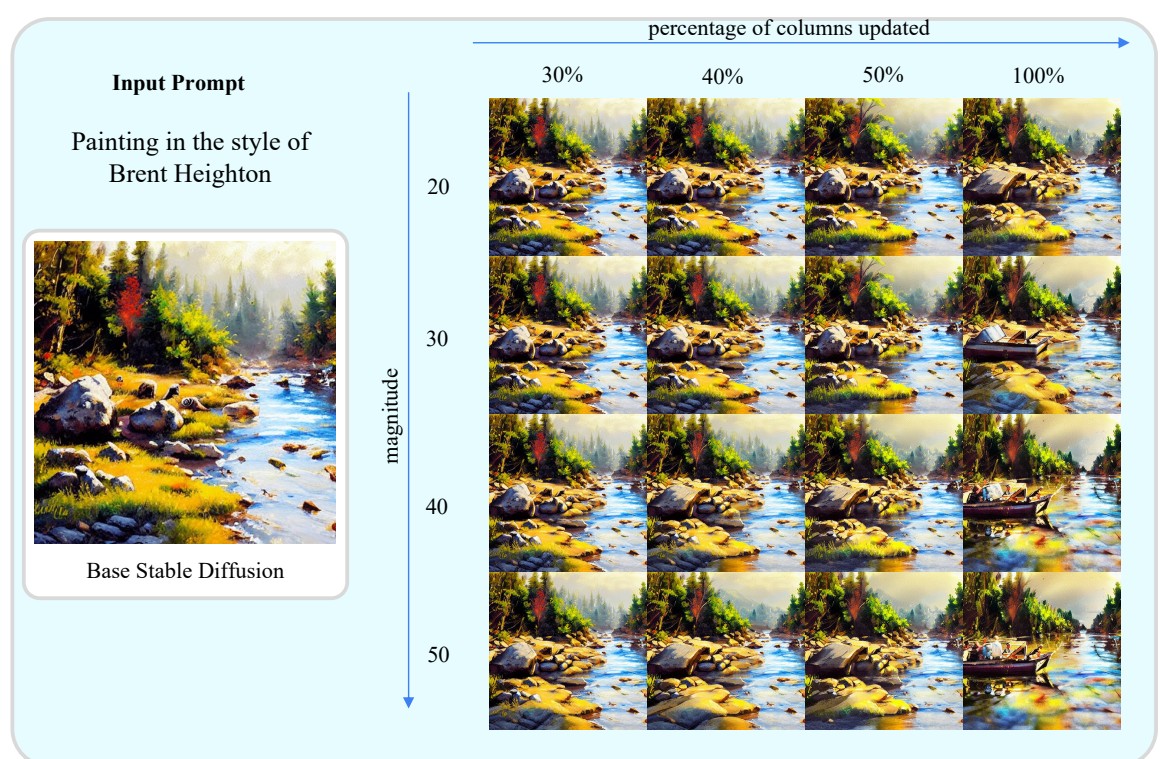

(a) **Prompt:** Painting in the style of Brent Heighton.

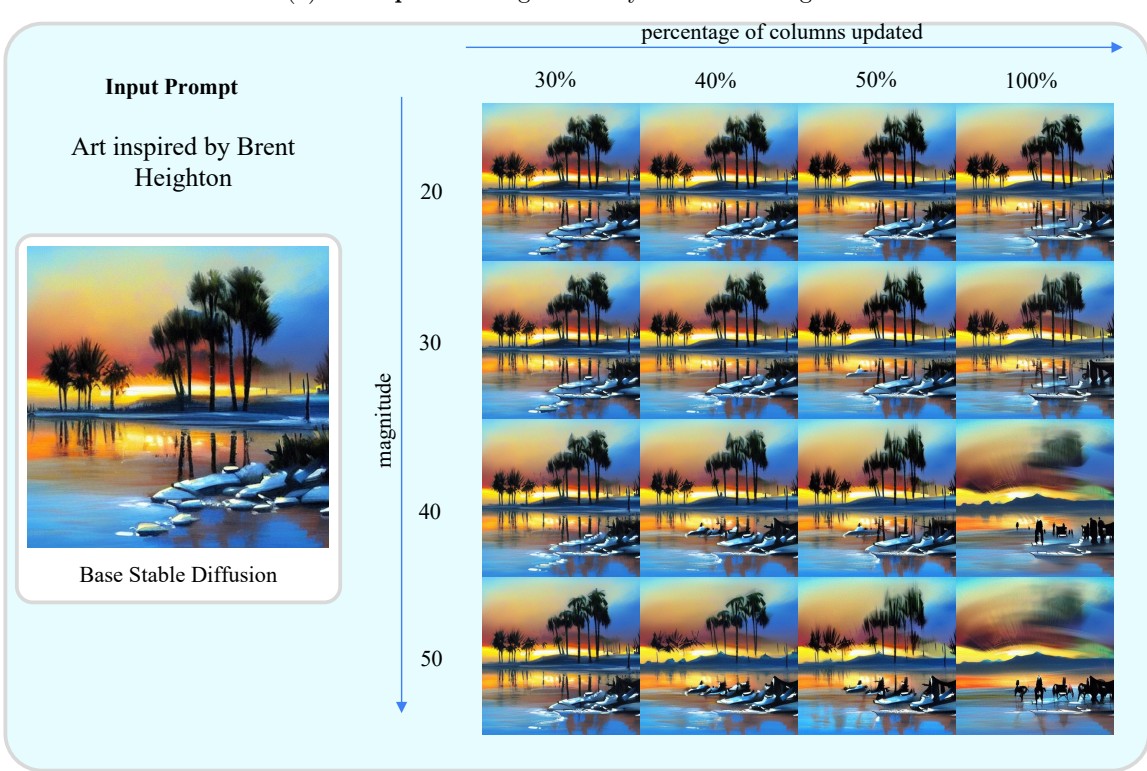

(b) **Prompt:** Art inspired by Brent Heighton.

Figure 25: **Forgetting Artistic Style:** Results after forgetting artistic style of *Van Gogh* and controlling the results at inference time for *Brent Heighton*. These images show that Concept Siever leads to minimal side-effects for neighboring concepts and preservation of other things like content, orientation and colors, without explicit preservation. Although as we increase the strength of forgetting the side effects starts increasing.

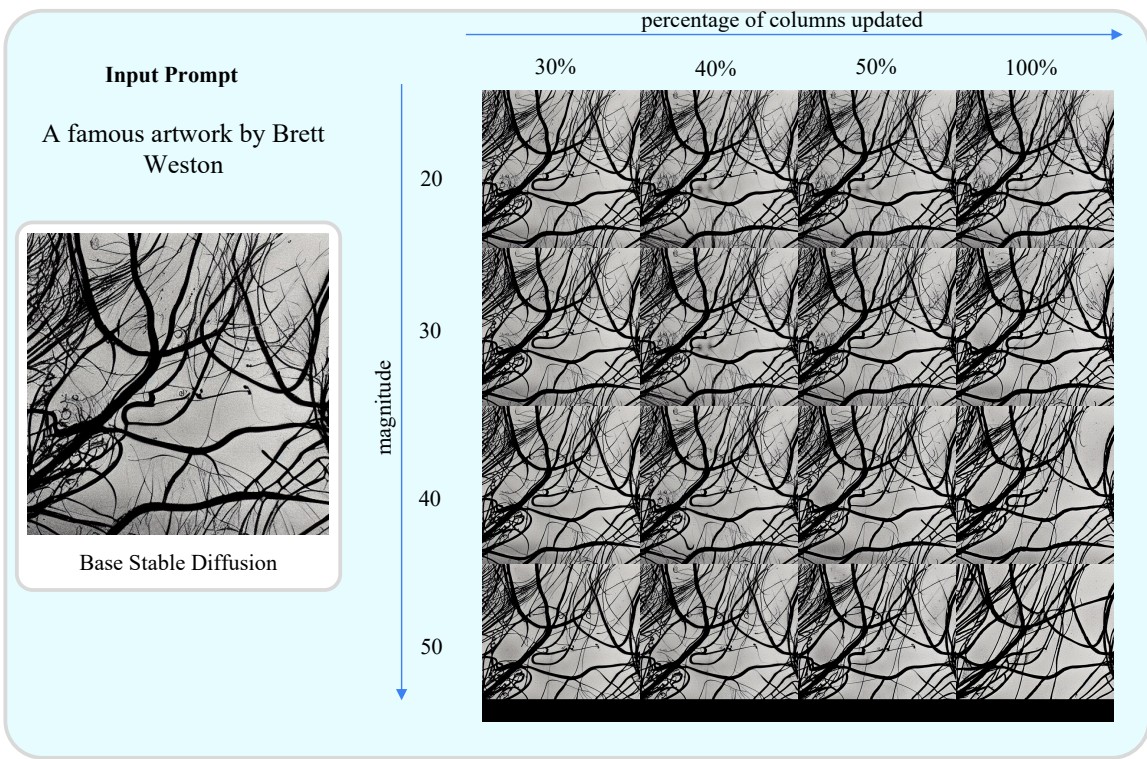

(a) **Prompt:** A famous artwork by Brett Weston.

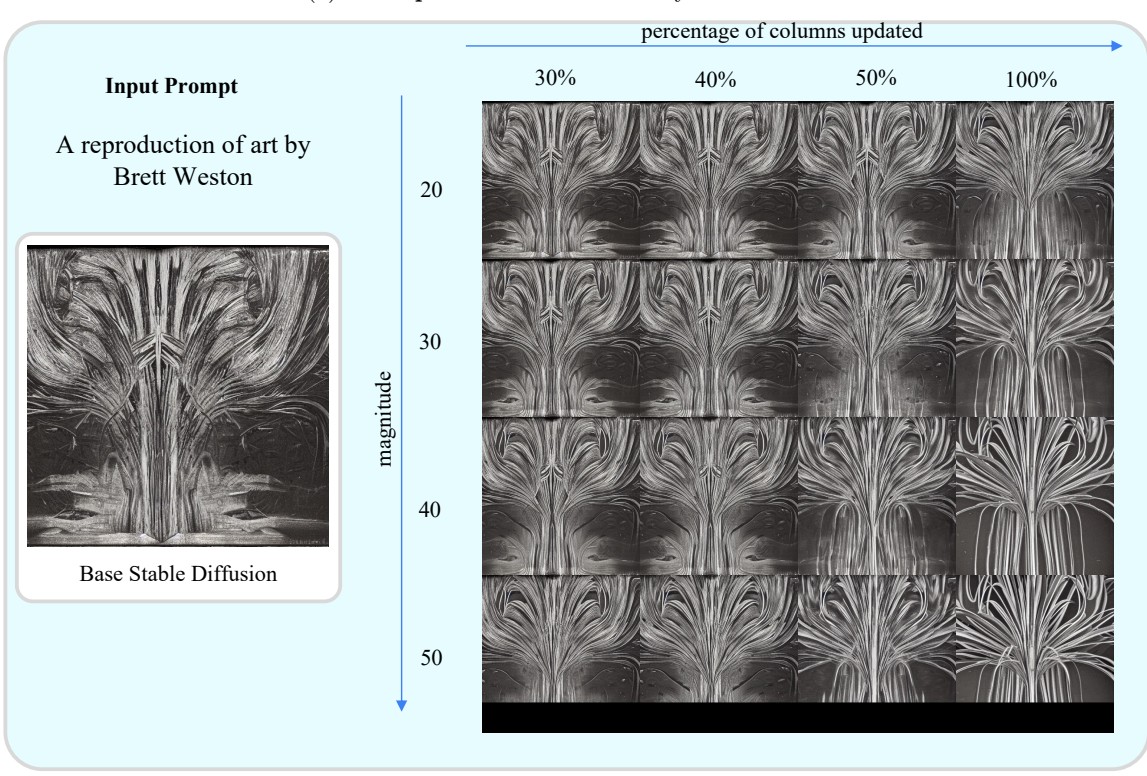

(b) **Prompt:** A reproduction of art by Brett Weston

Figure 26: **Forgetting Artistic Style:** Results after forgetting artistic style of *Van Gogh* and controlling the results at inference time for *Brett Weston.*These images show that Concept Siever leads to minimal side-effects for neighboring concepts and preservation of other things like content, orientation and colors, without explicit preservation. Although as we increase the strength of forgetting the side effects starts increasing.

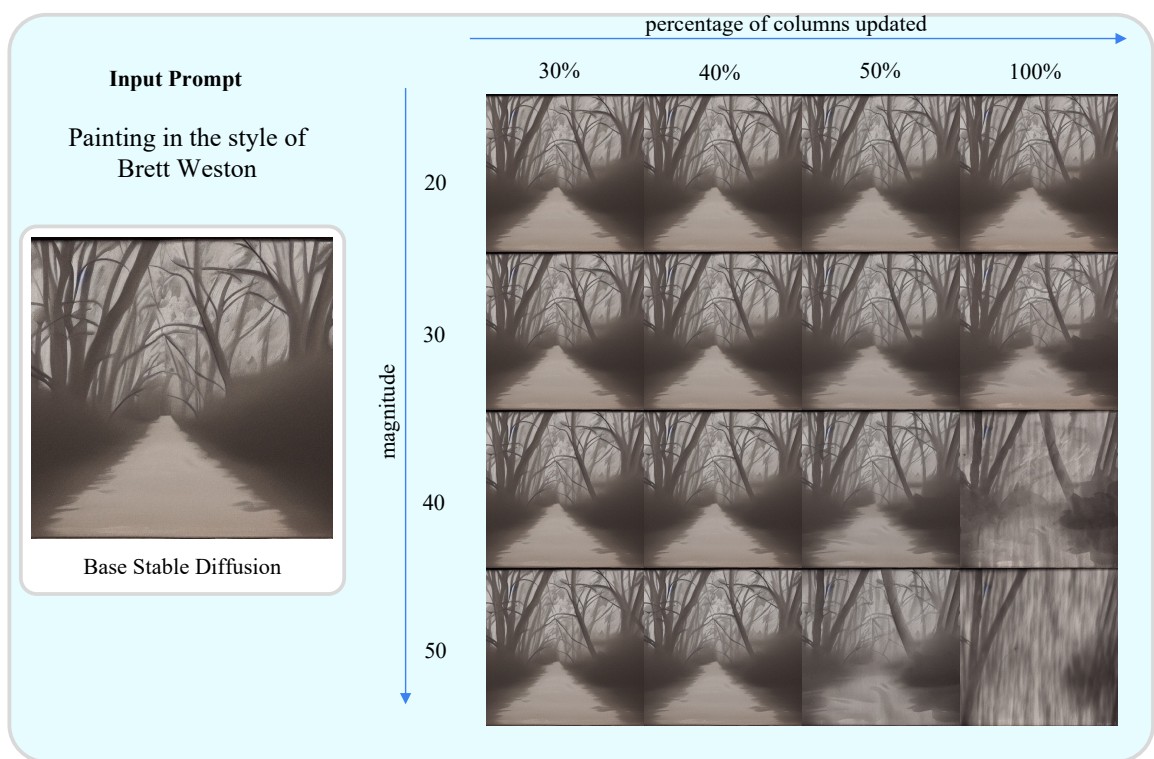

(a) **Prompt:** Painting in the style of Brett Weston.

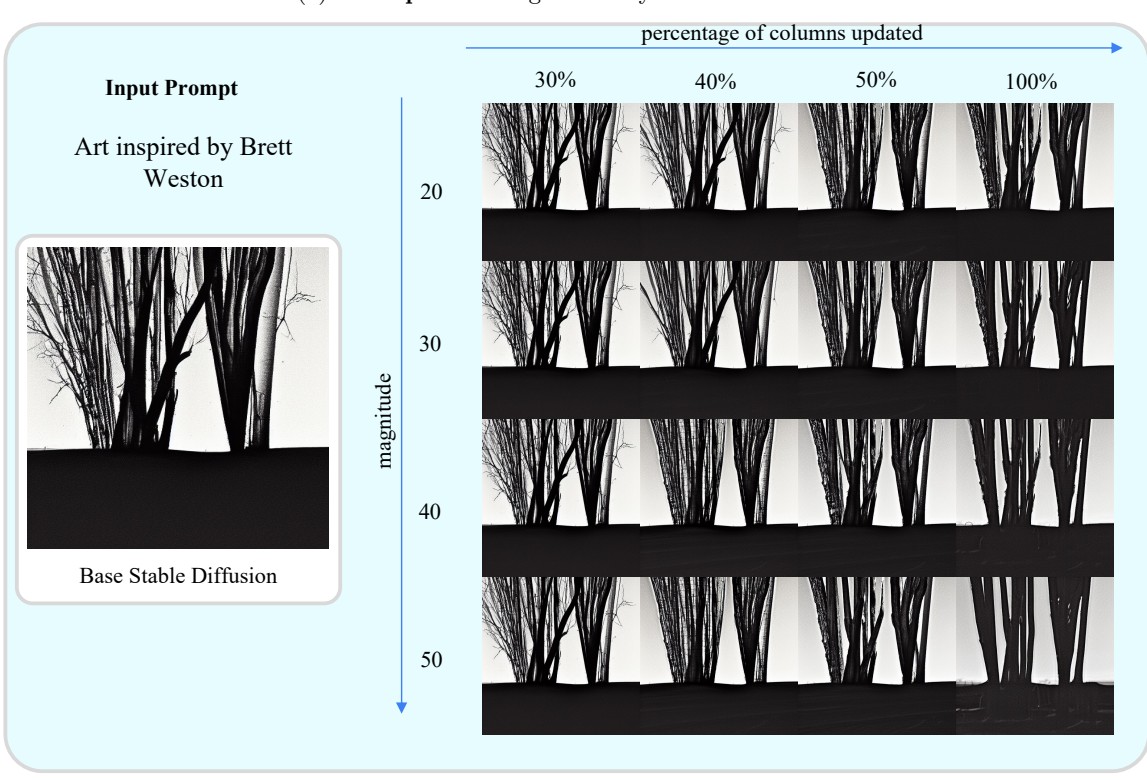

(b) **Prompt:** Art inspired by Brett Weston.

Figure 27: **Forgetting Artistic Style:** Results after forgetting artistic style of *Van Gogh* and controlling the results at inference time for *Brett Weston.* These images show that Concept Siever leads to minimal side-effects for neighboring concepts and preservation of other things like content, orientation and colors, without explicit preservation. Although as we increase the strength of forgetting the side effects starts increasing.

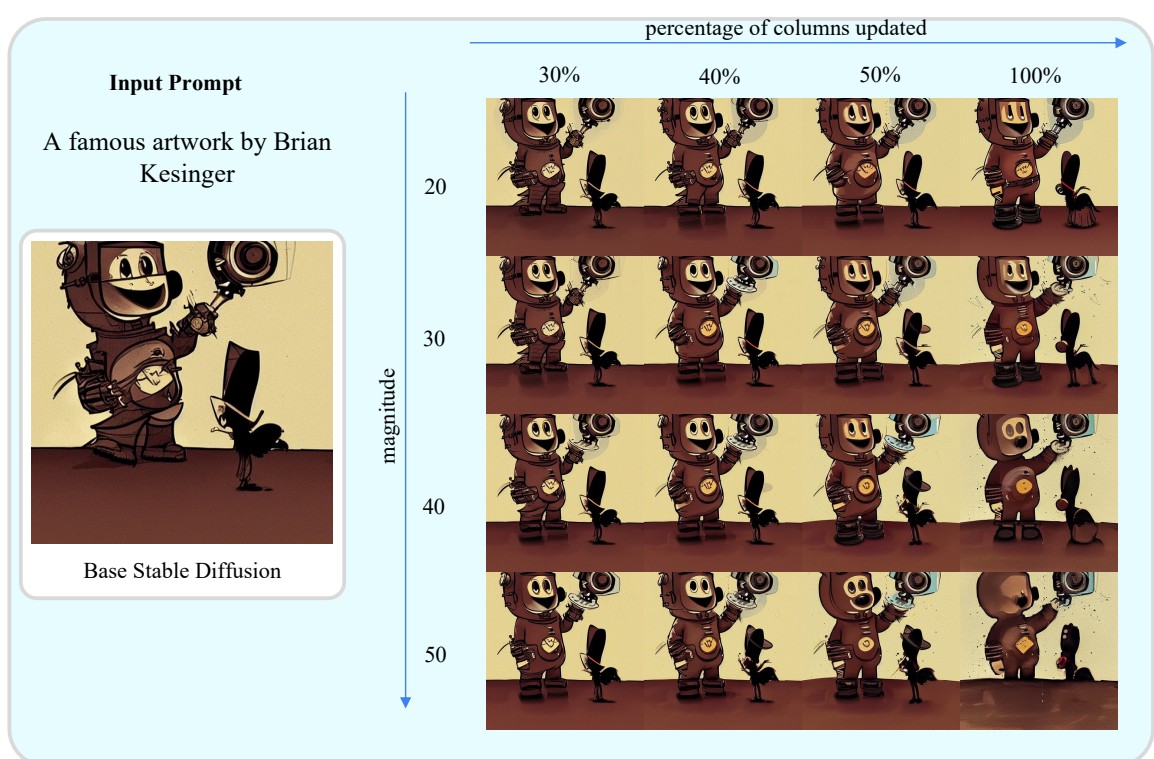

(a) **Prompt:** A famous artwork by Brian Kesinger.

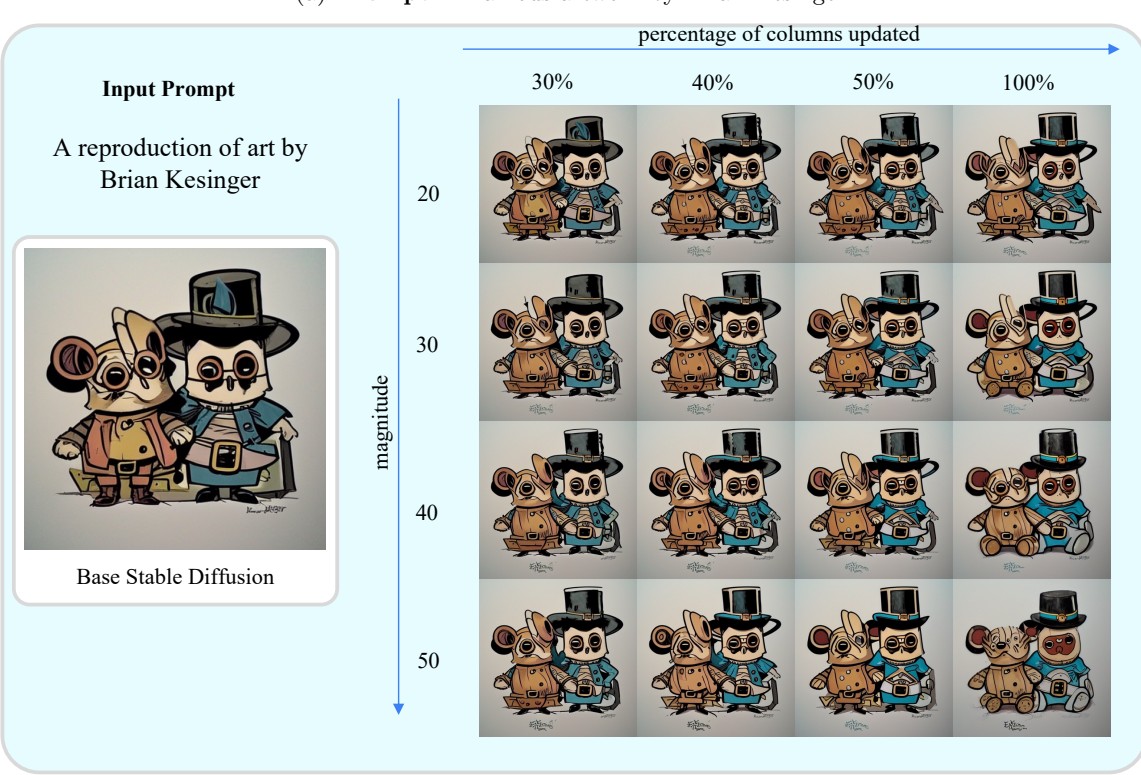

(b) **Prompt:** A reproduction of art by Brian Kesinger

Figure 28: **Forgetting Artistic Style:** Results after forgetting artistic style of *Van Gogh* and controlling the results at inference time for *Brian Kesinger*. These images show that Concept Siever leads to minimal side-effects for neighboring concepts and preservation of other things like content, orientation and colors, without explicit preservation. Although as we increase the strength of forgetting the side effects starts increasing.

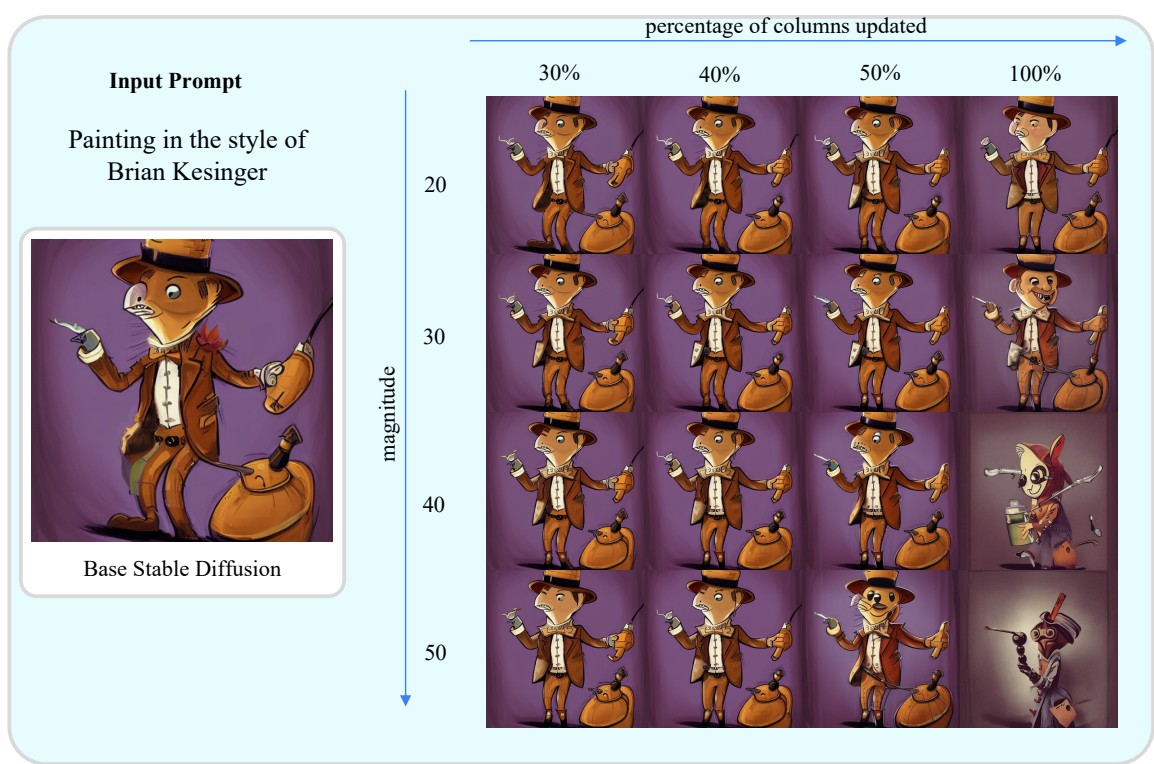

(a) **Prompt:** Painting in the style of Brian Kesinger.

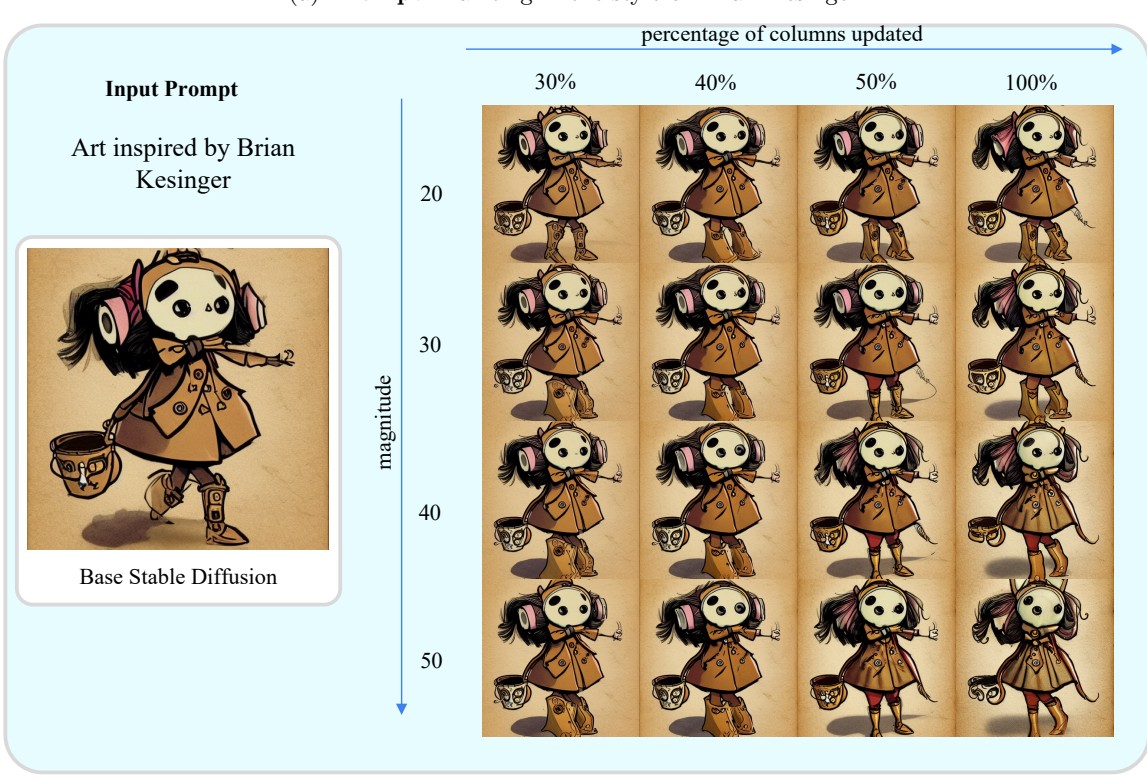

(b) **Prompt:** Art inspired by Brian Kesinger.

Figure 29: **Forgetting Artistic Style:** Results after forgetting artistic style of *Van Gogh* and controlling the results at inference time for *Brian Kesinger*. These images show that Concept Siever leads to minimal side-effects for neighboring concepts and preservation of other things like content, orientation and colors, without explicit preservation. Although as we increase the strength of forgetting the side effects starts increasing.

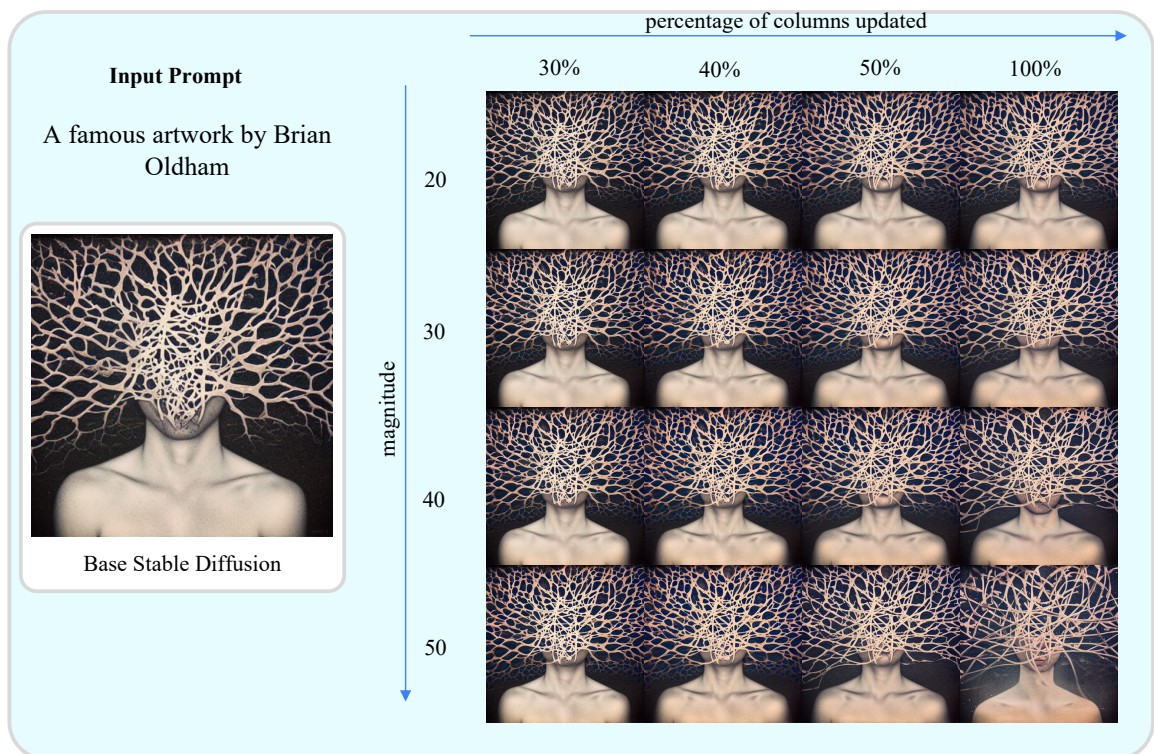

(a) **Prompt:** A famous artwork by Brian Oldham.

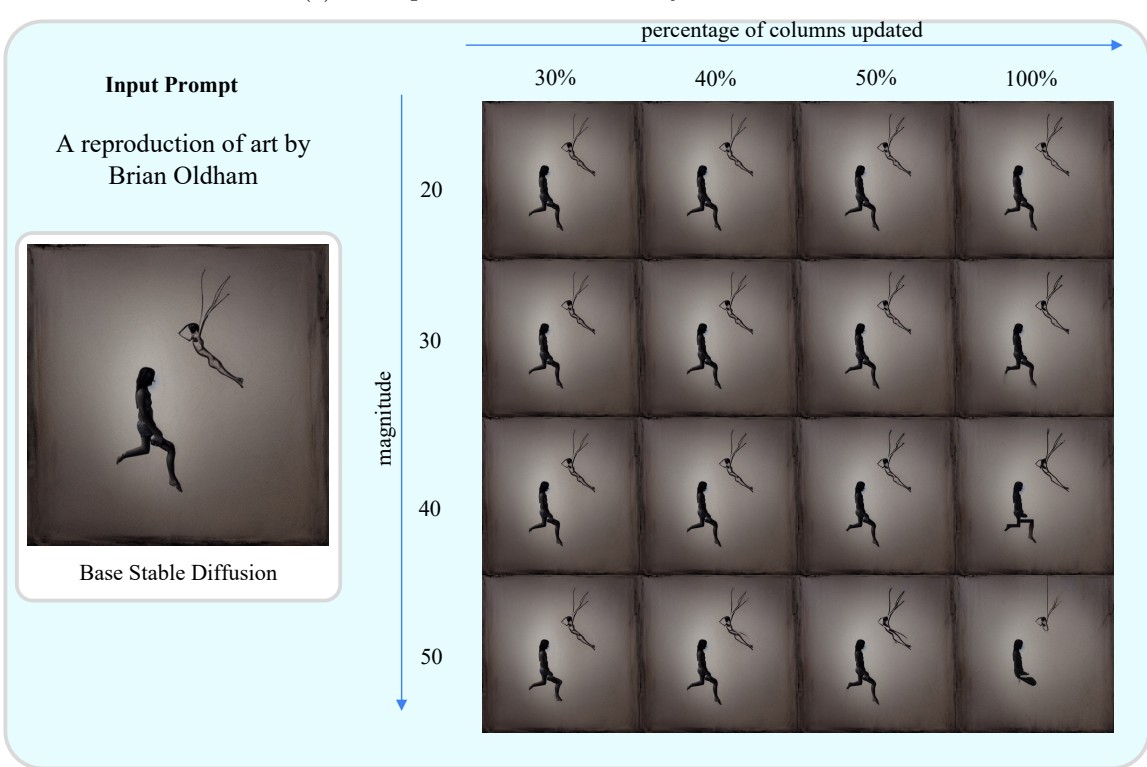

(b) **Prompt:** A reproduction of art by Brian Oldham

Figure 30: **Forgetting Artistic Style:** Results after forgetting artistic style of *Van Gogh* and controlling the results at inference time for *Brian Oldham*. These images show that Concept Siever leads to minimal side-effects for neighboring concepts and preservation of other things like content, orientation and colors, without explicit preservation. Although as we increase the strength of forgetting the side effects starts increasing.

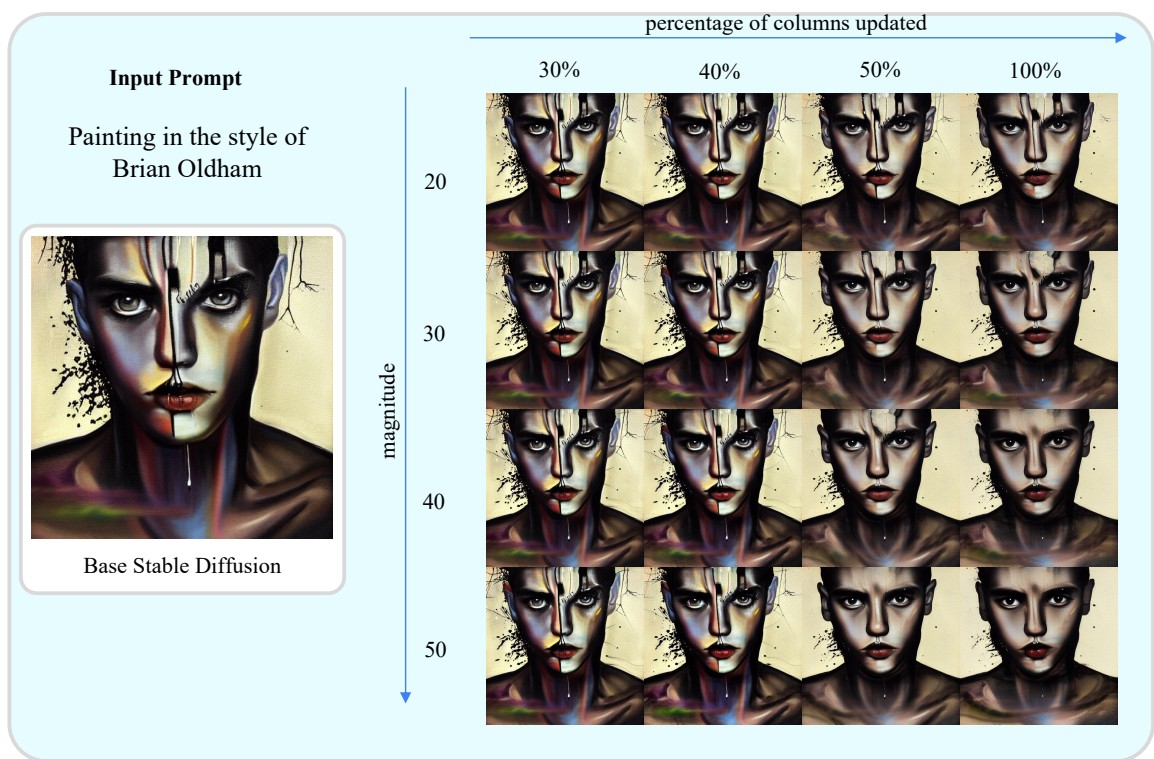

(a) **Prompt:** Painting in the style of Brian Oldham.

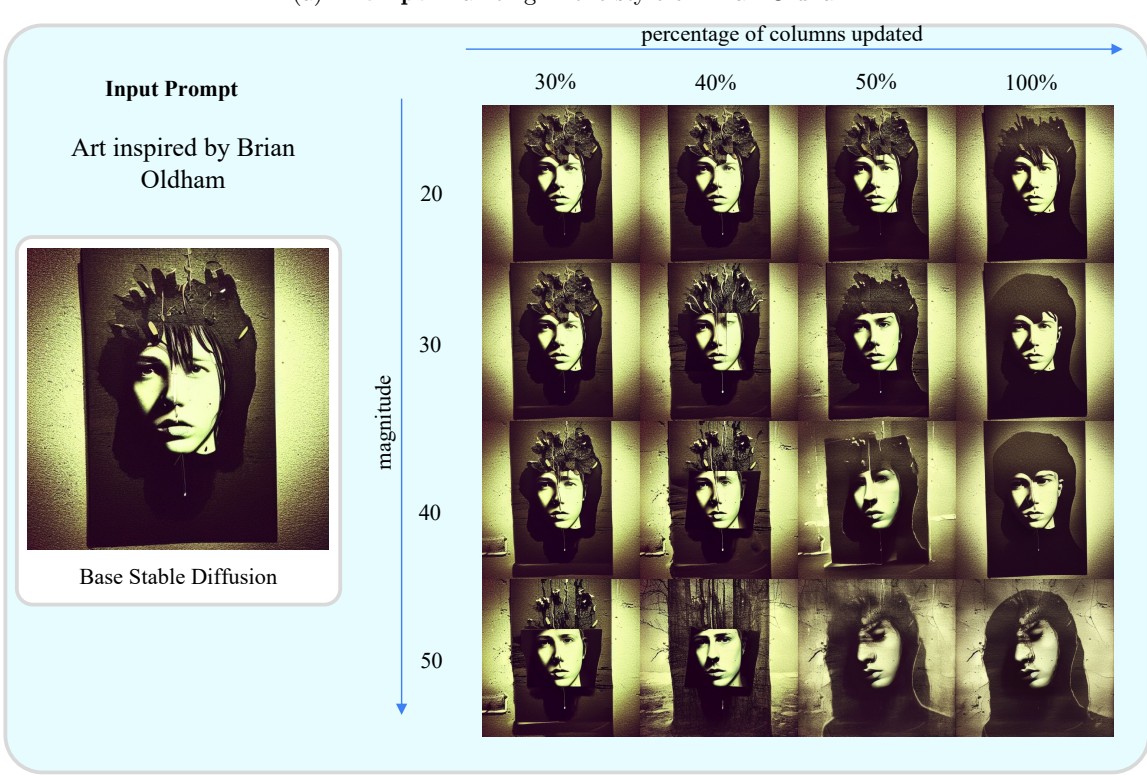

(b) **Prompt:** Art inspired by Brian Oldham.

Figure 31: **Forgetting Artistic Style:** Results after forgetting artistic style of *Van Gogh* and controlling the results at inference time for *Brian Oldham.* These images show that Concept Siever leads to minimal side-effects for neighboring concepts and preservation of other things like content, orientation and colors, without explicit preservation. Although as we increase the strength of forgetting the side effects starts increasing.

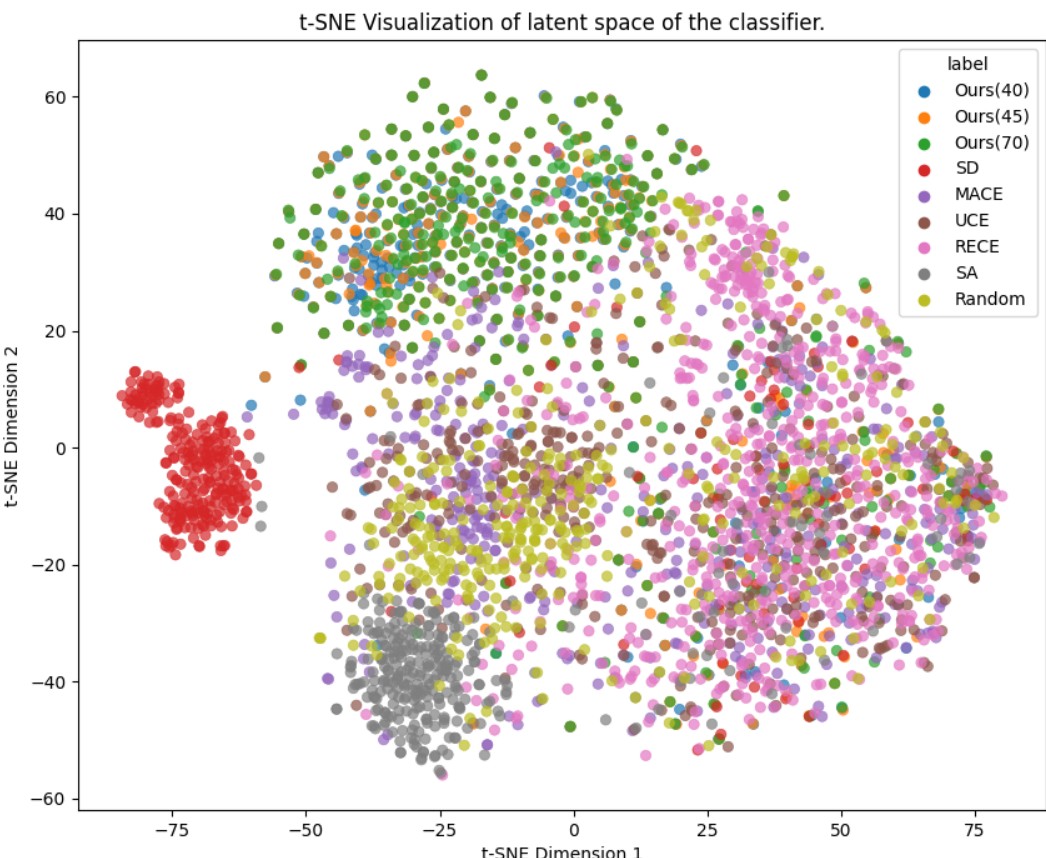

Figure 32: **t-SNE plot of celebrity across baselines:** Analyzing the face embedding of the samples of "Brad Pitt" before and after forgetting for models in Table 3.

.