# OpenReview forum: "Concept Siever : Towards Controllable Erasure of Concepts from Diffusion Models without Side-effect"
_TMLR — Accepted by TMLR_

### Review · Reviewer_kSZQ · 2025-08-16

**Summary Of Contributions:**

This paper tackles the problem of concept erasure in diffusion models. The authors propose Concept Siever, an end-to-end framework that automatically constructs concept/anti-concept pairs by perturbing CLIP embeddings of the target concept tokens. Two DoRA adapters are then trained on these pairs, and their difference defines a "forgetting direction" that can be applied at inference time with a controllable strength parameter $\lambda$. Experiments on NSFW content removal (I2P), celebrity identity suppression, and artistic style forgetting demonstrate significant improvements over domain-agnostic baselines while maintaining image quality and semantic alignment. Additional user studies and qualitative results highlight controllable forgetting strength and structure preservation.

**Additional Comments:**

- How robust is the proposed method to adversarial prompts (e.g., those generated by MMA-Diffusion [1])?

Ref:

[1]: Yang et al., MMA-Diffusion: MultiModal Attack on Diffusion Models. CVPR 2024.

**Audience:**

Yes

**Audience Explanation:**

I believe audiences working on diffusion models and concept unlearning would be interested in it.

**Broader Impact Concerns:**

The submission does not include an ethics or broader impact statement, which is concerning given the sensitivity of the problem it addresses. Concept forgetting has clear societal and regulatory implications, particularly in contexts such as NSFW suppression, privacy protection, and copyright compliance. The paper also include unsafe images for demonstration (e.g., nude person), but no disclaimer or warning is presented. I suggest the authors include a Broader Impact Statement that discuss the potential ethical concerns and how to mitigate them.

**Claims And Evidence:**

Yes

**Claims Explanation:**

The claims in the submission are mostly supported by evidence, but not fully convincing. On the positive side, the paper provides quantitative results on the I2P benchmark and compares against several baselines, with clear improvements in NSFW unlaerning while maintaining CLIP scores and FID. The qualitative examples are compelling, and the ablation studies on the controllable parameter $\lambda$ and masking strategies demonstrate that the method works as intended. However, the evidence has important gaps: the evaluation relies almost entirely on a single NSFW detector (NudeNet), which may bias the results, and no robustness checks with alternative detectors are presented. The experiments are also limited to Stable Diffusion 1.4, raising doubts about scalability to newer backbones. Finally, the assumption that Gaussian noise perturbation yields valid anti-concept samples is not validated with direct analysis. Overall, the evidence is clear but not fully convincing.

**Requested Changes:**

- Provide stronger validation of the NSFW unlearning performance beyond a single NudeNet detector. At minimum, include evaluations with multiple independent classifiers or human annotation to rule out detector-specific bias.
- Demonstrate the method’s generalization beyond Stable Diffusion 1.4 by including experiments on at least one newer backbone such as Stable Diffusion 2.1 or SDXL, showing whether the approach scales without major modification.
- Justify the anti-concept construction strategy with empirical evidence. For example, evaluate whether perturbing embeddings with Gaussian noise consistently removes only the intended concept across paraphrased or synonymous prompts.

---

> ### Author Response · Authors · 2025-09-10
> **Authors' response to Reviewer kSZQ (1/3)**
>
> We sincerely thank the reviewer for their prompt and thoughtful feedback, recognizing the strengths of our work. We are pleased that the reviewer finds our quantitative improvements on I2P, the qualitative results, and the ablation studies clearly demonstrate the effectiveness of our proposed approach. We address their concerns and questions below:
>
> - _**Stronger validation of NSFW unlearning performance**_
> > _include evaluations with multiple independent classifiers or human annotation to rule out detector-specific bias._
>
> We thank the reviewer for raising this important point. Indeed, we agree that validation beyond a single detector is important to rule out detector-specific biases. We had reported our results using the NudeNet detector as that is the primary (and the only) detector used in the current unlearning literature to evaluate NSFW performance. Upon reviewer’s request, we evaluated several open-source classifiers in the safety field (Open-NSFW, Shield Gemma, and CLIP-based FALCON) for NSFW detection in AI-generated images. However, we found that these classifiers performed poorly in detecting AI-generated NSFW content, with very low detection rate even in the input image, although the input images clearly contained NSFW content when checked manually . Therefore, we resorted to a **user study** with 16 participants who rated randomly-sampled NSFW-forgotten images on three dimensions:
>
> (1) Absence of nudity,
>
> (2) Preservation of (non-NSFW) structure, and
>
> (3) Overall image aesthetics (artifacts, realism).
>
> Ratings were given on a 1–5 scale (with rating 5 indicating very good performance on that metric), and the study compared our method with AdvUnlearn (best domain-agnostic) and GLoCE (best domain-specific), with participants blind to the source of each image. Results with average rating for each evaluation metric are presented in the table below.
>
> |**Evaluation Metric**|**Domain-specific**|**Domain-agnostic**||
> |:-|:-:|:--:|:-:|
> |**(rating 1-5)**|**GLoCE**|**AdvUn**|**Ours**|
> |Absence of Nudity (Rating 5: No NSFW)|3.61|2.98|**3.63**|
> |Structure Preservation (Rating 5: Fully preserved)|1.59|1.90|**3.58**|
> |Image Aesthetics (Rating 5: Real-looking with no artifacts)|2.00|2.60|**3.80**|
> |||||
>
> We draw two key observations from these results:
>
> a) _Superior forgetting performance:_ Our method secures the highest rating in NSFW forgetting, scoring 3.63 compared to 2.84 for AdvUnlearn and 3.61 for GLoCE, demonstrating robust unlearning performance with independent validation.
>
> b) _Better preservation and aesthetics:_ We note that the baseline approaches perform significantly worse on structure preservation and aesthetics. Their visual inspection shows that they often replace NSFW content with unrelated random images, failing to preserve concepts or image quality, further underscoring the superior preservation quality of our approach. Interestingly, GLoCE performs even worse than AdvUnlearn on these metrics despite its domain-specific nature.
>
> ---
>
> - _**Generalizability of results beyond SD 1.4**_
>
> Thanks for raising this. It is indeed important to demonstrate the generalizability of the framework to multiple model architectures. Therefore, we evaluate our model’s performance on both SD 2.1 and SD-XL on the **I2P benchmark** of NSFW forgetting, and report the results in the table below. For reference, we also report the SD-1.4 performance results.
>
> |**I2P**||||||
> |:-:|:-:|:-:|:-:|:-:|:-:|
> |**Model**|**# Training samples**|**Base Model**|**Ours**|**Improvement over Base SD (%) $\uparrow$**|**Drop in Specificity (FID) w.r.t Base SD on COCO-1K $\downarrow$**|
> |SD-1.4|1000|743|16|97.85|2.90|
> |SD-2.1|1000|586|32|94.54|1.13|
> |SDXL*|500|344|100|70.93|-0.05|
> |||||||
>
> From the table, we can observe that our framework does indeed generalize to other model architectures as well, with negligible drop in specificity on the MS COCO-1K dataset w.r.t. the base model. We’ll add these results to our updated draft.
>
> *Note on SD-XL: We faced GPU memory issues in linearizing the model weights of SD-XL due to incompatibility of SD-XL with diffuser library’s memory optimization procedure. Therefore, we could not do it successfully using the compute resources available at our end, and that's why we report results for the non-linearized version. Also, due to limited time, we could only train the SD-XL model on 500 image samples instead of the standard 1K samples. Despite these factors, the trend remains positively consistent. We believe that linearizing the SD-XL model weights and training on 1K samples would bring the performance trend of our framework for that model closely on par with other architectures (i.e., 90%+ improvement over Base SD).
>
> Thanks again for the helpful suggestions in improving the work further.
>
> ---

---

> ### Author Response · Authors · 2025-09-10
> **Continuation of Authors' response to Reviewer kSZQ (2/3)**
>
> - _**Empirical validation of the concept-negated dataset construction strategy**_
>
> Thanks for the question. Our claim regarding the concept-negated (**CN**) dataset is that it primarily differs from the concept dataset only in the concept to be forgotten, while largely preserving the background. The first validation of the efficacy of our CN-dataset in perturbing the given concept is already provided in Table-8 of our supplementary, where we evaluate the concept and concept-negated dataset under various task classifiers (NudeNet classifier for detecting NSFW content and GCD classifier to detect celebrity identity) to evaluate their concept-perturbation efficacy. We provide the table below again for ease of reference (Note: We have updated the table a bit (header names and more columns) to make it more clear. We’ll update this change in the draft as well).
>
> |||||||
> |:-|:-:|:-:|:-:|:-:|:-:|
> |**Task**|**Evaluator**|**Evaluation Metric**|**No Perturbation**|**With Perturbation**|**CN Efficacy**|
> |NSFW|NudeNet classifier|# NSFW Images |1504|173|88.50%|
> | Celebrity Identity |GCD classifier| % images as Brad Pitt|67.22%|4.81%|92.90%|
> |Artistic Style|--|LPIPS|0|0.549|54.90%|
> |||||||
>
> From the table, we can observe that our concept-negated data generation process effectively perturbs out the given concept from the classifier standpoint.
>
> Next, to show that this change is primarily coming from the concept part of the image and not from anywhere else, we perform a series of controlled ablations for the task of forgetting the celebrity identity. We generate the concept and CN datasets for this and use an off-the-shelf segmentation method (Grounded-SAM) to isolate the concept (celebrity here) and its background, and perform the following experiments:
>
> (1) **CN-BG-only:** We mask the celebrity in both the concept and CN dataset using the segmentation mask, and replace it with a mean-pixel value (here ImageNet mean), and calculate LPIPS between them. This approximately quantifies the changes in the image other than the celebrity (concept). A method which preserves background details well should obtain a very low LPIPS score here.
>
> (2) **CN-no-change:** This is the LPIPS between the concept and unchanged CN dataset.
>
> (3) **CN-without-BG:** We extract only the celebrity masks from both datasets, crop them to the tightest bounding box around each celebrity, and compute the LPIPS between the cropped regions. A successful forgetting method should yield a high LPIPS between the two sets, indicating strong dissimilarity in the forgotten concept. Importantly, for a good forgetting method where backgrounds should be preserved, the above LPIPS value of "CN-no-change" should be strictly lower than this LPIPS score, since inclusion of unchanged BG will lower the LPIPS score.
>
> (4) **CN-concept-only-specific:** Since the concept of interest is the identity of the celebrity, we further crop the face region and compute LPIPS between the cropped faces in the concept and CN datasets. Relative to the broader "CN-without-BG" setting, this LPIPS score should be even higher, as a good forgetting method will alter only the facial identity while leaving bodily features and clothing largely unchanged. Please note that this ablation is specific to celebrity identity and may not generalize easily to other, more abstract concepts.
>
> (5) **CN-with-BG-change:** Finally, we replace the background of the CN images with a random background and then compute the LPIPS between the concept dataset and this background-replaced CN dataset. If a method fails to preserve background details, the resulting score will be similar to that of "CN-no-change", where LPIPS is computed against the unmodified CN dataset. By contrast, for a good forgetting method, this ablation should yield the highest LPIPS score among all settings, since both the concept and the background are being altered.
>
> **For a good concept forgetting method**, here’s the expected LPIPS trend for these ablations:
>
> > _CN-BG-only (1) < CN-no-change (2) < CN-without-BG (3) < CN-concept-only-specific (4) < CN-with-BG-change._
>
> Below, we report these LPIPS scores obtained **for our concept-negated dataset**:
>
> |**Ablation**|**LPIPS**|
> |:-|:-:|
> |CN-BG-only|0.268|
> |CN-no-change|0.440|
> |CN-without-BG|0.506|
> |CN-concept-only-specific|0.563|
> |CN-with-BG-change|0.608|
> |||
>
> As can be seen, our concept-negated data curation process follows the expected  ablation trend, demonstrating the efficacy of the technique. We also compute the IoU score between the concept and concept-negated segmented masks of the celebrity to test the change in foreground pixels. We obtain an **IoU of 0.8**, which does indeed indicate minimal changes in the foreground concept pixels (note that IoU varies from 0 to 1, with 1 indicating perfect alignment). We have added some qualitative examples for concept and concept negated data in Fig-29 of supplementary material.
>
> ---

---

> ### Author Response · Authors · 2025-09-10
> **Continuation of Authors' response to Reviewer kSZQ (3/3)**
>
> - _**Robustness of the proposed method to adversarial prompts**_
>
> Thank you for the question. To test this, we randomly sampled 100 adversarial prompts out of 1000 from the MMA Diffusion NSFW benchmark [1] (we picked the prompts from their huggingface repository [2]) and evaluated both the base SD-1.4 model and our NSFW-forgotten model on these prompts.
>
> For the base model, we found that a total of 99 out of 100 generated images were classified as NSFW by the NudeNet classifier, showcasing the adversarial nature of the prompts. We passed the same set of prompts through our model, and observed that **none of the corresponding generated images were tagged unsafe by the NudeNet classifier**. This validates the robustness of the proposed method to adversarial prompts. We're happy to provide further qualitative samples from this experiment also if needed.
>
> ---
>
> **Summary**
>
> We hope that with the additional results and clarifications provided above, we have addressed the concerns of the reviewer and look forward to their response.
>
> ---
>
> **Citations**
>
> [1]: Yang et al., MMA-Diffusion: MultiModal Attack on Diffusion Models. CVPR 2024.
>
> [2] https://huggingface.co/datasets/YijunYang280/MMA-Diffusion-NSFW-adv-prompts-benchmark.

---

> > ### Comment · Reviewer_kSZQ · 2025-09-11
> > **Thank you for your response!**
> >
> > Dear authors, thank you so much for your thorough response! I think you've done a good job in rebuttal and I believe the new experiments have addressed my concerns. The only minor suggestion is the concern on broader impact. As I mentioned in my initial review, the paper contains NSFW materials, which may somewhat disturb readers. I suggest the authors include (1) a disclaimer that warns readers of potential NSFW content (in the abstract or somewhere else), and (2) a Broader Impact Statement that discuss the potential ethical concerns and how to mitigate them. Thank you.

---

> ### Author Response · Authors · 2025-09-11
> **Thank you for the prompt response!**
>
> We sincerely thank the reviewer for their prompt response and are happy to see that their concerns are addressed.
>
> The suggestions on having a disclaimer and a Broader Impact Statement is indeed valid.  We will include a disclaimer after the abstract, warning the reader about the presence of NSFW content in the paper. Here is the exact text that we will be adding:
>
> > _**Warning: This paper contains examples of NSFW (not safe for work) content generated by text-to-image diffusion models.**_
>
> ---
>
> Next, we provide the following broader impact statement:
>
> **Broader Impact Statement:** While our proposed framework effectively removes harmful, explicit or copyrighted content through the concept sieve, we acknowledge its potential for dual use—specifically, the possibility of reversing the sieve to amplify the concepts which were supposed to be removed.  This concern is not just unique to our approach but applies broadly to the field of concept forgetting and model editing, and our method also falls within this scope. We advocate responsible use of our technology, and leave the development of techniques to detect whether images have had concepts suppressed or reinforced as future work.
>
> We will update the existing impact statement in Appendix-O of the supplementary with the above text.
>
> Thank you again for your time and efforts in reviewing our work. Happy to clarify any further queries that you might have.

---

> > ### Comment · Reviewer_kSZQ · 2025-09-11
> > **Thank you for your response!**
> >
> > Dear authors, thank you for the response! I have no further concerns.

---

### Review · Reviewer_Y7R6 · 2025-08-28

**Summary Of Contributions:**

# Summary of contribution
This work proposes an end-to-end concept-forgetting model for white-box text-to-image latent diffusion models called Concept Siever that does not require a domain-specific preservation set per concept, while enabling strong concept unlearning and preservation of untargeted concepts simultaneously. The proposed latent perturbation-based data generation protocol and the controllability of test-time forgetting strength bring practical benefits towards desired generative models.

# Strengths
* The proposed method achieves strong concept forgetting performance without requiring domain-specific real data for forgetting and remaining concepts.
* Dynamic control for the strength of forgetting during inference time enables flexible applications for various needs of users.

# Weaknesses
* Missing validation on the linearly separable weight hypothesis (the main working condition of the proposed Concept Sieve)
  * The authors formulate the Concept Sieve adapted new model weight as a linear combination of the original weight vector and concept sieve weight vector.
  * However, they do not discuss the realism of this hypothesis, nor how the violation of this hypothesis potentially affects the quality of forgetting.
  * For example, in the weight interpolation literature, there is a well-known hypothesis, '_linear mode connectivity_' [1,2], to support the validity of the linear weight merging-based approach. There is also a line of work that claims weight space linear separability does not naturally hold [3], and controlling the separability can affect the quality of weight vector arithmetic. The author can make use this kind of discussion to add rigor to their justification on methodology.
* Computational overhead
  * The proposed protocol requires data generation (hundreds of, and one thousand images) from a diffusion model, which requires somewhat huge computation.
  * It requires per-concept model training as well, which induces non-trivial resources.
  * If users curated a dataset using the proposed pipeline, they may prefer to use a training-free method, e.g., GLoCE, on top of it, rather than utilizing the proposed training-heavy method concept sieve.
* Overclaims
  * The authors introduce their data generation technique as an **automated method** in the abstract.
    * However, it requires hand-made text prompts for each concept to generate from them (the authors said they used 40-50 prompts for this). Although we can generate them with LLMs, it still requires prompt specification to LLMs, and obviously, this is hard to be seen as a fully automated method.
    * Besides, the variance of the Gaussian noise and the layer index for perturbation should also be tuned. As shown in the Appendix, those two hyperparameters affect the overall quality of generations significantly.
  * The authors emphasize that their data generation protocol ensures that the pair of concept and negated-concept images only differ in terms of the target concept while achieving minimal changes to other semantics. However, this claim was not validated, and **there is no theoretical guarantee that all the pairs from multiple concepts have only a difference in the target concept.**
* Presentation issue
  * [`Performance reporting inconsistency`]
    * The authors include AdvUnlearn in Table 2 and Table 4 but omit it in Table 3, which makes the generality of the claim "state-of-the-art among domain-agnostic methods" questionable.
    * In Table 4, the authors provide a new metric, 'structured LPIPS', as an alternative to standard LPIPS, but they only provide it on the forgetting concept evaluation, while not showing structured LPIPS for the remaining concepts.
  * [`Missing formulation of structured LPIPS`] Although the 'structured LPIPS' is a new metric proposed by the authors to justify their claim on the semantic content preservation capability of the model, they do not provide a formal formulation for the metric anywhere, which makes reader hard to figure out the validity of the design of structured LPIPS.

---

> Reference

1. Linear Mode Connectivity and the Lottery Ticket Hypothesis, Frankle et al. 2020
2. Robust fine-tuning of zero-shot models, Wortsman et al. 2022
3. Task Arithmetic in the Tangent Space: Improved Editing of Pre-Trained Models, Ortiz-Jimenez et al. 2023

**Audience:**

Yes

**Audience Explanation:**

Although I brought up some weaknesses of this work, the TMLR's audience may find some interesting insights and practical benefits from Concept Siever, given that it doesn't require domain-specific real data (though it requires some computational budget), inference-time dynamic control on the forgetting strength, and the promise of a task weight vector arithmetic-based concept forgetting approach.

**Broader Impact Concerns:**

Including this work, all the works related to concept-forgetting have potential risks that can be maliciously utilized by adversarial users. Simply using the reverse direction of the learned concept sieve can amplify the sensitive concepts of a model, so I think the author should discuss this potential misuse in a separate broader impact section.

**Claims And Evidence:**

Yes

**Claims Explanation:**

As mentioned in the "Summary Of Contributions" section, there are a couple of issues with the validation of the authors' claim:
1. the claim on the automated data curation pipeline => practically not automated.
2. the claim on the generated pairs of concept/negated-concept images "only differ in the target concept" is not rigorously validated by empirical or theoretical evidence.
3. tha claim on "achieving the state-of-the-art among the domain-agnostic methods" is not quite convincing, given inconsistent reporting of the previous SOTA method AdvUnlearn

**Requested Changes:**

[Requesting items to secure my recommendation for acceptance]
1. Weakening the claims on the data-generating pipeline -- non-automated and no guarantee on 'only differ in target concept'.
2. Comparing the computational cost and wall-clock runtime of Concept Siever to baseline methods at least in the Supplementary material, and guide readers to that section who may wonder about it.

[Requesting items to simply strengthen the work in my view]
1. Adding a brief discussion on the validity of the linear separability of the diffusion model weight to make the justification of the method stronger.
2. Consistently reporting the most competitive baseline, AdvUnlearn.
3. Adding formulation of the structured LPIPS or adding implementation details for that metric.

---

> ### Author Response · Authors · 2025-09-10
> **Authors' response to Reviewer Y7R6 (1/2)**
>
> We thank the reviewer for their time and valuable feedback on our work. We are pleased to find that the reviewer finds our forgetting performance as significant among the current domain-agnostic baselines, and our controllable test-time forgetting as practical. We address their concerns and questions below:
>
> - _**On data generation pipeline as an automated method**_
>
> We understand the reviewer’s concern here. We agree that all the methods, including ours, do need prompts to generate concept dataset, and our data-generation process also has some (interpretable) hyperparameters to control the strength of forgetting. However, we use the term “automated” to refer to the fact that most existing forgetting methods rely on “manually-curated explicit preservation sets” (domain-specific or generic). In contrast, our construction of concept-negated dataset _implicitly serves as a preservation set without requiring any manual curation of the list of concepts to be preserved_. We’ll add this clarification in the updated draft of the paper.
>
> ---
>
> - _**On validity of the concept-negated dataset differing only in concept to the concept-dataset**_
>
> We provide a systematic analysis of how our concept-negated dataset generation strategy ensures forgetting of target concept with minimal side-effects through a series of five experiments which we have detailed in response to Reviewer kSZQ (please see the response titled "_Empirical validation of the concept-negated dataset construction strategy_"). Though not proven theoretically, these empirical experiments clearly bring out the ability of our approach to create paired data that maximally differs only in the concept to be forgotten.
>
> ---
>
> - _**Computational overhead**_
>
> We acknowledge the reviewer’s concern regarding the compute overhead for forgetting each concept, and present below a quantitative comparison of the overhead of our method against existing domain-agnostic baselines for forgetting a single concept on a single A6000 GPU:
>
> | **Method**| **Data Generation Time** | **Training Time** | **Total time** |
> |-|-|-|-|
> | FMN| 5 images x 7sec| 5 mins| 6 mins|
> | AdvUnlearn|0 | 17 hrs| 17 hrs|
> | Selective Amnesia | 6000 images x 7sec| 80 hrs|92 hrs|
> | Ours| 1000 images x 7sec| 4.5 mins| 2.1 hrs|
> |||||
>
> As can be seen, our method achieves a _favorable runtime among existing domain-agnostic forgetting approaches_, second only to FMN. However, despite low runtime, _FMN consistently exhibits poor forgetting performance across all three tasks_ as shown in our paper (Table-2 and 3). For instance, in the I2P benchmark, FMN-based forgetting results in 424 NSFW images compared to 16 in ours. Similarly, in celeb identity forgetting, it achieves a forgetting efficacy of 18.75% compared to 4% of ours (lower is better here), with similar results in artistic style as well. This renders FMN's low runtime of limited practical value. Future work can look towards enabling forgetting with less no. of generated images.
>
> ---
>
> - _**On combining our data generation pipeline with training-free method like GLoCE**_
>
> Thanks for the question. We first note that methods like GLoCE rely on a domain-specific preservation set. This set is _explicit in nature_, where one needs a list of related or unrelated concepts to be preserved while forgetting a given concept, whereas our approach of creating concept and concept-negated dataset is _not the same_ as this explicit list of preserving concepts, but rather a way to create a paired dataset which varies primarily in the concept to be forgotten. We utilize this paired set to create a steering vector to update the model, whereas GLoCE performs a preservation step using the explicit preservation set.
>
> Apart from this, as we discussed in detail in the paper’s introduction, explicit preservation sets do come with their own limitations. Not just that, GLoCE often replaces the entire image with an unrelated random image, leading to poor preservation of structure and other concepts (as shown in Figure-1 and 4 of the paper). We provide supporting evidence for this in a user study, which is presented in our response to Reviewer kSZQ (see the title “_Stronger validation of NSFW unlearning performance_”).
>
> Therefore, while current training-free methods do offer low runtime, they are not really compatible with the nature of concept-negated dataset, and their significant side effects make them less suitable for effective concept forgetting.
>
> ---

---

> ### Author Response · Authors · 2025-09-10
> **Continuation of Authors' response to Reviewer Y7R6 (2/2)**
>
> - _**Discussion on the validity of linearly separable weight hypothesis**_
>
> We thank the reviewer for providing relevant pointers on addressing the discussion on the validity of linearly separable weight hypothesis and its relevance in our work. We provide such a discussion below motivated by the provided pointers, and will also elaborate it in the draft to make it explicit and clear:
>
> Prior work on linear weight interpolation [1,2] has shown that linearly merging task-specific model weights can improve performance trade-offs across tasks—a phenomenon termed as task arithmetic. However, [3] highlights that such linear separability does not always hold, motivating the need for weight disentanglement to make task arithmetic more effective. Their analysis connects disentanglement to model linearization and the spatial localization of kernel eigenfunctions.
>
> Building on this foundation, we adapt this hypothesis for concept forgetting by leveraging paired concept and concept-negated datasets to provide accurate steering directions to the linearized model vector. We present a mathematical formulation of this approach in Sec-C of the supplementary. We further refine this direction vector by sparsifying it using our dual strategies of layer localization and column masking, enabling targeted and localized forgetting of concepts in text-to-image diffusion models.
>
> ---
>
> - _**Clarification on reporting pattern for AdvUnlearn**_
>
> We omit AdvUnlearn on celebrity identity (Table-3) because, despite training as per AdvUnlearn’s recommended hyper-parameter settings, we were unable to create a stable model checkpoint using their official codebase where the model was able to perform any targeted forgetting. Instead, the model consistently produced random noise outputs for other celebrities once Brad Pitt was forgotten, a behavior we observed across multiple identities. For this reason, we did not report AdvUnlearn results for celebrity identity task. We’ll add a note about this in the final draft.
>
> ---
>
> - _**Clarification on reporting pattern of Structure LPIPS in Table-4**_
>
> The key reason why we introduce Structure LPIPS is to measure how structurally consistent the image looks to the original image after forgetting the concept. For the remaining concepts (Other Concepts column in Table-4), we want to measure how close it is to the original image, not just structurally, but also on other aspects like style, content, etc.; hence standard LPIPS suffices for checking how these multiple attributes are preserved maximally.
>
> ---
>
> - _**Missing formulation of Structure LPIPS**_
>
> We thank the reviewer for pointing this out. While section-4.3 of the main paper provides the methodological description of this metric, we provide below the exact algorithm of Structure LPIPS, and have updated the same in the supplementary draft (Appendix A) for completeness.
>
> ### **Structure LPIPS: Algorithm**
>
> > **Input:** $I_{orig}$ (Original Image), $I_{forgotten}$ (Forgotten Image)
> >
> > **Hyperparameters:** $S$ (Set of blur variances, e.g., $\{10, \dots, 19\}$)
> >
> > **Output:** Structure LPIPS score
> >
> > ---
> >
> > $total\_{lpips} \leftarrow 0$
> >
> > **for** each variance $\sigma \in S$ **do**
> >
> > &nbsp;&nbsp;&nbsp;&nbsp; $I'\_{orig} \leftarrow \text{GaussianBlur}(I_{orig}, \sigma)$
> >
> > &nbsp;&nbsp;&nbsp;&nbsp; $I'\_{forgotten} \leftarrow \text{GaussianBlur}(I_{forgotten}, \sigma)$
> >
> > &nbsp;&nbsp;&nbsp;&nbsp; $score \leftarrow  \text{LPIPS}(I'\_{orig}, I'\_{forgotten}) $
> >
> > &nbsp;&nbsp;&nbsp;&nbsp; $total\_{lpips} \leftarrow total\_{lpips} + score$
> >
> > **end for**
> >
> > **return** $\frac{total\_{lpips}}{|S|}$
> ---
>
> - _**Broader Impact Statement**_
>
> While our proposed framework effectively removes harmful or copyrighted content through the concept sieve, we acknowledge its potential for dual use—specifically, the possibility of reversing the sieve to amplify forgotten concepts. This concern is not unique to our approach but applies broadly to the field of concept forgetting, and our method also falls within this scope. We will add this discussion to the paper draft.
>
> ---
>
> **Summary**
>
> We hope that with the additional results and clarifications provided above, we were able to address the concerns of the reviewer and we look forward to their response.
>
> ---
>
> **Citations**
>
> [1] Linear Mode Connectivity and the Lottery Ticket Hypothesis, Frankle et al. 2020.
>
> [2] Robust fine-tuning of zero-shot models, Wortsman et al. 2022.
>
> [3] Task Arithmetic in the Tangent Space: Improved Editing of Pre-Trained Models, Ortiz-Jimenez et al. 2023.

---

> ### Comment · Reviewer_Y7R6 · 2025-09-12
>
> I really appreciate the authors' professional rebuttals, including an additional experiment!
> All my concerns are addressed, and I believe that the authors will edit the draft accordingly.
>
> I have updated my initial review item: Are the claims made in the submission supported by accurate, convincing and clear evidence?: `No` -> `Yes`

---

> > ### Author Response · Authors · 2025-09-13
> > **Official Comment by Authors**
> >
> > Thank you very much for your time and inputs towards making our paper stronger. Glad that we could address the concerns. Please be assured that we will update the draft with all the new experimental analysis and clarifications that we had added during the review phase.
> >
> > Thank you!

---

### Review · Reviewer_Km7W · 2025-08-30

**Summary Of Contributions:**

The paper introduces Concept Siever as a new model editing-based framework to unlearn specific concepts from pre-trained text-to-image diffusion models. The method's primary contribution is an automated, domain-agnostic pipeline that does not require manually curated sets of images or prompts to generate a “concept-negated” dataset and preserve related concepts. This is achieved by adding Gaussian noise to the CLIP text embeddings of the target concept, which is hypothesized to disrupt the concept while preserving all other image attributes. Then, by fine-tuning two separate low-rank adapters on the original and negated datasets and subtracting their weights, the method derives a “sieve” vector that, when applied to the base model, erases the target concept.

Strengths:
1. The work targets aa relevant and challenging problem of concept erasure, which has significant implications for AI safety, copyright, and privacy.
2. The method achieves state-of-the-art results on several benchmarks.
3. The idea of generating a dataset via embedding perturbation is interesting and appears to be reasonable to the model editing toolkit.
4. The framework provides practical fine-grained control over the concept erasure process, which could be valuable for real-world deployment.

Weaknesses
1. The paper claims to solve the problem of reliance on “preservation sets”. However, what it does fundamentally is to automate the creation of an implicit preservation set, thus (likely) inheriting the original problem’s limitations rather than solving them.
2. The overall framework relies on a (unproven) assumption that concepts are linearly separable and disentangled within the model's latent and weight spaces. However, the paper provides no direct evidence to support this premise.
3. Some of the core claims are not well supported by the method and empirical studies. See below for details.

**Audience:**

Yes

**Audience Explanation:**

The paper’s findings would be of interest to a broad segment of the TMLR audience. The work's value is primarily the empirical results and its introduction of a practical technique. The communities that might find this work interesting include: AI safety practitioners, generative model and model editing researchers.

**Broader Impact Concerns:**

The work is well-motivated from an AI safety perspective, aiming to provide tools for removing harmful or copyrighted content. The potential for dual-use (e.g., for malicious censorship) is a general concern for the entire field of model editing and is not specific to this paper. The paper does not require a separate Broader Impact Statement.

**Claims And Evidence:**

No

**Claims Explanation:**

The paper successfully supports its claims of empirical efficacy, which is commendable. Yet it fails to provide convincing evidence for other fundamental claims, and some are even contradicted by its own results.

First, the paper’s primary motivation is to overcome the limitations of preservation sets (brittleness, incompleteness, domain expertise). It claims to solve this by eliminating the need for them. However, this appears to be a mischaracterization - the concept-negated dataset ($\mathcal{D}_{\rm cn}$) is an implicit preservation set that serves the exact same purpose: providing a large corpus of examples to constrain the model's update and preserve desired attributes. In this sense, the paper has not eliminated the preservation set; it has automated its creation. So, the method is still limited by the quality and completeness of this automated set, and it is difficult to confirm the core problem of ensuring specificity has well been addressed.

Second, the overall framework relies on a critical assumption that concepts are sufficiently disentangled in CLIP's latent space, such that adding noise to the "Van Gogh" token does not corrupt semantically related concepts like "Post-Impressionism" or "thick brushstrokes." The paper provides no theoretical or empirical evidence to validate this assumption. If this assumption does not hold, the observed side-effects, e.g., degradation of related concepts, are not minor flaws but the reasonable consequence of the flawed premise: concept entanglement causes the sieve vector $\tau$ to inadvertently target more than just the intended concept.

Third, some core claims are contradicted by the paper’s own results. For example, the claim of “erasure without side-effects” is demonstrably false. Table 3 shows a clear drop in specificity (top-1 acc for other celebrities from 96.6% to 92.2%).

In summary, the paper successfully proves it has built a high-performing tool but fails to substantiate the claims.

**Requested Changes:**

1. The authors should revise the manuscript to accurately frame their contribution. The narrative should shift from “solving the preservation set problem” to “proposing a new method to automate the creation of an implicit preservation set”. This requires major revision to many places including abstract, intro, and conclusion to better reflect the method is a new heuristic, not yet a fundamental solution to the specificity problem.
2. The authors should add a dedicated discussion of the method’s assumption: the linear subtractability and functional disentanglement of concepts in the latent space. They should acknowledge this is a strong assumption and explicitly connect it to the literature and the empirically observed side-effects.
3. To provide evidence for the claim that non-target attributes are preserved, the authors should conduct an analysis comparing the statistical properties of the concept dataset and the concept-negated dataset. For example, the authors might consider to use off-the-shelf classifiers to show that the distribution of key attributes (e.g., pose, background type, secondary objects) are unchanged.
4. The paper is believed to be more robust if i includes a qualitative analysis of failure modes that result from concept entanglement. For instance, does forgetting a concept like "lion" also damage the generation of "tiger" or "cat"? This would be providing insights into whether ther are limitations when we rely on the implicit structure of the CLIP embedding space.

---

> ### Author Response · Authors · 2025-09-10
> **Authors' response to Reviewer Km7W (1/2)**
>
> We sincerely thank the reviewer for their thoughtful review and for recognizing the strengths of our work. We are encouraged that the reviewer found our embedding perturbation strategy for dataset generation and the fine-grained control enabled by our framework to be practical and valuable contributions. We address their questions and concerns below:
>
> - _**Clarification on the concept-negated dataset as an implicit preservation set and the narrative related to it in the paper**_
>
> Many thanks for raising this concern, this gives us the opportunity to clarify our approach better.
>
> **Explicit Preservation set:** The preservation set used by existing related approaches is constructed from examples of classes that need to be retained after forgetting. For instance, if we want to forget Brad Pitt, and want to retain Leonardo DiCaprio and Tom Cruise, these approaches need full disclosure that Leonardo DiCaprio, Tom Cruise should be retained. Such a model is not guaranteed to retain another celebrity, say Tom Hanks. The key takeaway is that the set of concepts to be retained _should be explicitly specified_ in the preservation set of existing approaches. Such a set is indeed _uncountably infinite_.
>
> **Concept-negated (CN) dataset:** In contrast, our approach needs only the concept to be forgotten (Brad Pitt in our example). We generate pairs of concepts and concept-negated dataset, where concept-negated data varies mainly in the concept to be forgotten, while retaining the structure of the concept data. Then this data is used to create a concept vector (a vector in the weight space that contains information of what constitutes the concept to be forgotten, say Brad Pitt). This vector is then subtracted from the base diffusion model to remove the concept. Please note that during this process, we do not need any information of which concepts should be retained (DiCaprio, Tom Cruise, Tom Hanks, etc.). Hence, we do not quantify what all exists in the uncountably infinite set of classes to be retained (either related or unrelated), and instead focus on removing only the class that has been specified.
>
> **Interpretation and understanding of CN dataset:** The concept negated dataset can be treated as an implicit preservation set, but it has been generated from the model without any input regarding which all concepts should be retained. Thus it overcomes the limitations of having an “explicit” preservation set, which suffers from the issues we discussed in the introduction of the paper (brittleness, incompleteness, domain expertise, and not model-specific). Our implicit preservation set, thus, does not serve the “exact same purpose” of the preservation set of existing approaches, because it does not need to explicitly know what all is to be preserved. We will revise the manuscript to further bring out these nuances better.
>
> **CN dataset's effectiveness:** By effectively perturbing out a concept in a given model, our approach helps in improving the model specificity, ensuring minimal impact to other concepts. We provide detailed empirical evidence of the effectiveness of the CN dataset in localizing the concept in our response to reviewer kSZQ (please see response titled "_Empirical validation of the concept-negated dataset construction strategy_"). Along with the crucial refinements of layer localization and column masking, it does indeed provide a good step towards solving the issue of specificity in concept forgetting.
>
> We hope this clarifies the concern of the reviewer. We will revise the manuscript to accurately frame the contribution.
>
> ---
>
> - _**Discussion on the linear subtractability hypothesis used in our method**_
>
> Great question. We provide a discussion on this in our response to reviewer Y7R6 (please see the response titled "_Discussion on the validity of linearly separable weight hypothesis_"), tying it back to relevant past literature on linear subtractability. We’ll add this discussion in our final draft.
>
> ---
>
> - _**Clarification on the claim that non-target attributes are preserved**_
>
> Thanks for the question. We provide a controlled empirical study of the efficacy of the concept-negated dataset in our response to reviewer kSZQ (please see response titled "_Empirical validation of the concept-negated dataset construction strategy_"). We will also add this discussion in the supplementary draft.
>
> ---
>
> - _**Clarification on the claim of "erasure without side-effects"**_
>
> Our intention was not to claim that we perform concept forgetting without _any_ side-effects. Rather, our intended goal was to propose a framework which provides a step towards effectively solving this issue of specificity with _minimal_ side-effects. As mentioned above, we also provide empirical validation of the same. We apologize if the message came out otherwise to the reviewer. We would appropriately update our text to make this clear to the reader.
>
> ---

---

> ### Author Response · Authors · 2025-09-10
> **Continuation of Authors' response to Reviewer Km7W (2/2)**
>
> - _**Qualitative analysis of robustness and failure modes of the method**_
>
> **Additional analysis on concepts like lion:** We performed evaluation of our method on concepts like lion and zebra as per reviewer's suggestion, and checked for preservation of neighboring concepts like tiger, cat and bull. We present the qualitative results in the newly added Fig-31 of the supplementary. Overall, we did not see any major degradation in the generation of neighbouring classes like tiger (while forgetting lion) and horse (while forgetting zebra). There are some minor texture and color differences in the preserved images, but that can be alleviated with some tuning of the hyperparameters.
>
>
> **Qualitative results of failure cases:** One of the failure cases we observed during our experimentation of perturbing in the CLIP latent space is during forgetting a concept like Laptop, where if one tries to forget a laptop like Macbook, because the concept of Macbook is so strongly entangled with the Apple logo, such a forgetting results in only the removal of the Apple logo from Macbook. We show qualitative examples of this in Fig-32 of the supplementary. This highlights the issue of strong concept entanglement in the CLIP latent space, which will need stronger perturbation strategy and additional modifications than only the concept tokens. Future research can dive deeper into these aspects of our framework to improve it further.
>
> We will update the draft with these discussions.
>
> ---
>
> **Summary**
>
> We hope that with the additional qualitative results and clarifications provided, we could address the reviewer's concerns. We look forward to their response.

---

> ### Author Response · Authors · 2025-09-17
> **Gentle reminder to review our rebuttal**
>
> Dear Reviewer Km7W,
>
> We would like to hear from you on whether our rebuttal has addressed your concerns. Please let us know if you have any remaining questions or if any further clarification is needed from our end.
>
> Thanks,
>
> Authors

---

> > ### Comment · Reviewer_LwxK · 2025-09-25
> >
> > Sorry for the late response!
> > We are really pleased with the response of the authors and recommend accepting this paper.
> >
> > Congrats!

---

### Review · Reviewer_LwxK · 2025-09-03

**Summary Of Contributions:**

### Summary

The authors proposed a principled and scalable method for removing concepts from Stable diffusion model. The authors proposed a automated method to curate image text pairs with or without certain concept, specifically, by generating the image twice with target concept token intact or masked out. This paired dataset is used to finetune the model twice, resulting in two DoRA weight vector, then the difference of the weights is used to change the base model, resulting a change in model behavior, focused on removing of the concept.

The authors performed extensive experiment to validate the efficiency and specificity of this method.

**Audience:**

Yes

**Audience Explanation:**

Yes, editting diffusion model is generally interesting question, and pratically relevant to usage of them in reality. The idea in the method (esp. auto data curation) is quite general and should be known by more people.

**Broader Impact Concerns:**

No concern

**Claims And Evidence:**

Yes

**Claims Explanation:**

### Strength

- The general conceptual framework for thinking about diffusion forgetting is interesting! esp. the specificity and generality tradeoff.
- The automated paired dataset creation is very interesting, and this paradigm should be broadly useful for T2I in general.
- Using contrast between finetuning LORA to measure importance of layer and edit model is interesting! It’s like an advanced form of gradient attribution.
- Figures are very well made and illustrative. The experimental validation is also thorough.
- The analysis of cross attention layer and and layer attribution is interesting!

**Requested Changes:**

### Weakness

- The notation and formalism in the method part Sec 3.2 is not clear, and could be improved by a lot.
    - As an example $z^\prime=\phi(x) + \tau \circ \nabla \phi (x)$.
        - in stable diffusion, there is VAE, and UNet, and the UNet has (at least) three inputs, latent $x$, time step $t$ and the prompt conditioning $y$. the notation authors used is a bit unclear, and I recommend making this explicit.
        - Similarly $\nabla$ is not clear which derivative the authors are taking, it’s better to denote it’s specifically gradient or Jacobian towards $w$.
        - also should it be $\phi(x)$ or $\phi(z)$?
    - For Eq. 3 and 4, the authors should mention that this decomposition is applied to matrix like parameters (also mention which matrix parameter do they include e.g. attention?) The equation do not make sense for vector parameters…
    - In Eq. 3, why $\|.\|_c$? what does c denote?
- The loss function seems important, but it’s not a separate equation. It should be written as a oneline equation and note the definition of z’ there.
- Further, for Eq. 3, since it involves gradient to parameter in the loss, does the optimization involves 2nd order gradient? which parameter is trained in this final finetuning? it’s also not clear from the text… is it just the LORA weights or all weights?
- The Figure 8B insight about ft effect on QKVO weights are super interesting! Can you quantify this finding more? I can only see one example, it seems that fine tuning QK is pretty good, kind of removing the style but not the content…

### Questions

- Why Stage 1 corrupted the penultimate layer of CLIP instead of last layer? What’s the rationale or intuition behind it? (I know the authors mentioned that they tuned such parameter and validate it).
- Since the CLIP embedding is contextual and information flows everywhere, is there fear that even if you messed up with the specific tokens correspond to the concept, but other tokens already contains certain information of the ablated concept?
- The training method is very efficient, why? Is it tricky to tune how many gradient steps are used to train the two LORA? Is there risk of under train or over train the LORA?

---

> ### Author Response · Authors · 2025-09-10
> **Authors' response to Reviewer LwxK (1/2)**
>
> We sincerely thank the reviewer for their encouraging feedback. We are glad that the reviewer finds our conceptual framework, automated paired dataset creation, layer importance measures and the analysis of cross-attention layer and layer attribution to be interesting. We are further encouraged by the reviewer's positive remarks on the clarity of our figures, the thoroughness of our experimental validation and the relevance of our work for the TMLR audience. We address the reviewer's questions and concerns below:
>
> - _**Clarifications in notations and formalism**_
>
> Thank you very much for carefully analysing our methodology, and providing valuable feedback.
>
> In the equation $z^′ = {\phi}\_{\theta}(x) + \tau · \nabla{\phi}\_{\theta}(x)$, where ${\phi}\_{\theta}$ refer to the UNet of the Stable Diffusion (we will clarify this further in the explanation towards the start of Section 3.2). $\nabla$ refers to the Jacobian, and ${\theta}(x)$ should be ${\theta}(z)$ (thanks for pointing this out!). We will update the line below Eq-5 to the following:
>
> > The objective function $L$ for this fine-tuning stage is the MSE loss between $z_t$ and ${z^{\prime}}\_{t}$ , where $z_t$ is the encoded latent representation of the image $x_t$ at time step $t$, and  ${z^{\prime}}\_{t}$ is computed as follows: $ {{z^{\prime}}\_{t}} = \phi_{\theta}(z_{t+1}, t, y) + \tau \cdot \nabla\phi_{\theta}(z_{t+1}, t, y)$. The variable $y$ is the text conditioning.
>
> In Eq. 3 and 4, the decomposition is indeed applied to the matrix. ${w^{\prime}}\_{i} \in \mathbb{R}^{m \times n}$ is divided into $\boldsymbol m \in \mathbb{R}^{m}$ and  $\boldsymbol V \in \mathbb{R}^{m \times n}$ .  $\boldsymbol V$ is further decomposed into  $\boldsymbol A \in \mathbb{R}^{m \times r}$ and  $\boldsymbol B \in \mathbb{R}^{r \times n}$ where $r$ is the rank of decomposition.  We train all the linear layers (Attention, time projection etc.) as well as convolution layers of the model.
>
> $\parallel . \parallel_c$, $c$ refers to column-wise norm operation, where ${w^{\prime}}\_{i}$ is broken down into magnitude $m_k$ and unit vector directions $v_k$.
>
> We will ensure that the loss function will be a separate equation. Thanks again for raising these points, it helps us improve the presentation and clarity of the paper.
>
> ---
>
> - _**Clarifications on Eq. 3 and associated text**_
>
> > _Further, for Eq. 3, since it involves gradient to parameter in the loss, does the optimization involve 2nd order gradient? Which parameter is trained in this final finetuning? it’s also not clear from the text… is it just the LORA weights or all weights?_
>
>
> We thank the reviewer for these clarifying questions.
>
> - **On Second-Order Gradients:** Our optimization does not involve second-order gradients. The term $ \tau · \nabla{\phi}\_{\theta}(x)$ in Eq. 3 is used to define the modified output $z'$ via a first-order Taylor approximation, inspired by recent work on task arithmetic in tangent space. During the optimization step, this gradient term
> $\nabla{\phi}\_{\theta}(x)$ is treated as a constant with respect to the parameters being trained. The actual loss function is a standard Mean Squared Error, $L=∥z−z'∥_2^2$​, which is minimized using a standard first-order optimizer like Adam.
>
> - **On Trained Parameters:** We  only train the parameter-efficient DoRA adapter weights. This includes both the magnitude vector $\boldsymbol m​$ and the low-rank directional matrices $\boldsymbol A$ and $\boldsymbol B$​, as shown in Eq. 4 of the paper. All original pre-trained weights of the UNet remain frozen during this process.
>
> We will revise the text to make these points clear.
>
> ---
>
> - _**Quantification of the finding in Fig-8B**_
>
> We thank the reviewer for their interest in our attention-component analysis. To quantify this behavior, we generated 150 images with two prompts (“A painting in the style of Van Gogh” and “Self-portrait of Van Gogh”), identified the two most influential cross-attention layers per prompt, and ran the ablation in Fig. 8B across all images. We selectively trained each component (Key, Query, Value, Out) and measured its effect via LPIPS between the original Stable Diffusion output and the ablated model. To capture these subtle structural changes, we report structured LPIPS instead of the standard LPIPS. Results are shown in the table below:
>
> |**Training configuration**| **Structured LPIPS** |
> |:-|:-:|
> | All Attention layers |0.519 |
> |All but Out|0.516 |
> |All but Value|0.452 |
> |Key and Query|**0.253** |
>
>
> As can be observed, training only the key and query matrices of the selected layers result in the least Structured LPIPS score, suggesting good proximity to the input image compared to other ablations. This quantification corroborates the qualitative results we had presented in Fig-8B. We’ll add this quantification result in the updated draft for completeness.
>
> ---

---

> ### Author Response · Authors · 2025-09-10
> **Continuation of Authors' response to Reviewer LwxK (2/2)**
>
> - _**Rationale behind corrupting penultimate layer of CLIP instead of last layer**_
>
> During our analysis to understand the CLIP latent space for perturbation, we found that adding noise _directly to the final CLIP embeddings_ often results in outputs that fall outside the manifold of realistic images. While a simple alternative is to add noise just before the final normalization layer, this approach _significantly dampens the effect of the perturbation_, likely due to the suppressing effect of the normalization itself. We therefore observed that the latent space prior to the final linear projection is most amenable to smooth interpolation. By adding noise at this intermediate stage, we can effectively negate a concept while ensuring the generated output remains within the desired image manifold. We show some qualitative examples for this in the newly added **Fig-30 of the supplementary material**.
>
> ---
>
> - _**Clarification on token-specific perturbations in CLIP embeddings**_
>
> We thank the reviewer for raising this concern. Our empirical analysis in Table-8 of the supplementary demonstrates that perturbing only the concept tokens in CLIP space is effective and sufficient to perturb out the given concept. Our construction of the concept-negated dataset is through targeted perturbations, and in Table-8, we evaluate it using task classifiers to assess whether the intended concept has been successfully perturbed out or not. We report the table below again for ease of reference (we modify the table a bit to make it read better).
>
> |||||||
> |:-|:-:|:-:|:-:|:-:|:-:|
> |**Task**|**Evaluator**|**Evaluation Metric**|**No Perturbation**|**With Perturbation**|**CN Efficacy**|
> |NSFW|NudeNet classifier|# NSFW images |1504|173|**88.50%**|
> | Celebrity Identity |GCD classifier| % images as Brad Pitt|67.22%|4.81%|**92.90%**|
> |Artistic Style|--|LPIPS|0|0.549|**54.90%**|
> |||||||
>
> The results confirm that the concept-negated dataset behaves as expected, validating the effectiveness and sufficiency of the token-specific perturbation strategy. Combined with the dual-strategies of layer localization and concept masking, it enables effective forgetting of concepts.
>
> ---
>
> - _**Clarification on compute overhead of our method**_
>
> In terms of compute required for forgetting, we perform LORA training for 100 epochs (default setting in the HuggingFace diffusers library), and we found this to be a good default for achieving strong forgetting results. In addition, we ablate the LoRA ranks to ensure a good trade-off between specificity and efficacy (results of this ablation are reported in Sec-G.2 and figure-12 of the supplementary). Notably, 100 epochs of LoRA fine-tuning is computationally lightweight (~5 minutes on a single A6000 GPU) and works out of the box for concept forgetting, demonstrating the usefulness of the technique. For completeness, we also report below the comparative compute overheads of our method relative to existing baselines:
>
> | **Method**| **Data Generation Time** | **Training Time** | **Total time** |
> |-|-|-|-|
> | FMN| 5 images x 7sec| 5 mins| 6 mins|
> | AdvUnlearn|0 | 17 hrs| 17 hrs|
> | Selective Amnesia | 6000 images x 7sec| 80 hrs|92 hrs|
> | Ours| 1000 images x 7sec| 4.5 mins| 2.1 hrs|
> |||||
>
> As can be seen, our method achieves the _most favorable runtime among existing domain-agnostic forgetting approaches_, second only to FMN. However, despite low runtime, _FMN consistently exhibits poor forgetting performance across all three tasks_ as shown in our paper (Table-2 and 3). For instance, in the I2P benchmark, FMN-based forgetting results in 424 NSFW images compared to 16 in ours. Similarly, in celeb identity forgetting, it achieves a forgetting efficacy of 18.75% compared to 4% of ours (lower is better here), with similar results in artistic style as well. This renders FMN's low runtime of limited practical value.
>
> ---
>
> **Summary**
>
> We hope that with the additional results and clarifications provided, we have addressed the reviewer's concerns. We look forward to their response.

---

> ### Author Response · Authors · 2025-09-17
> **Gentle reminder to review our rebuttal**
>
> Dear Reviewer LwxK,
>
> We would like to hear from you on whether our rebuttal has addressed your concerns. Please let us know if you have any remaining questions or if any further clarification is needed from our end.
>
> Thanks,
>
> Authors

---

> > ### Comment · Reviewer_LwxK · 2025-09-19
> > **Thanks for all your responses and clarifications! here are some remaining concerns**
> >
> > Thanks for the effort to clarify the concerns!
> >
> > Here are some small but remaining concerns / requests.
> >
> > **Regarding the notation clarity**
> > In this updated equation,
> > ${{z^{\prime}}_{t}} = \phi_{\theta}(z_{t+1}, t, y) + \tau \cdot \nabla\phi_{\theta}(z_{t+1}, t, y)$
> > I feel it's still ambiguous what the Jacobian $\nabla$ is. I think it's the partial derivative of the model output w.r.t. the parameters? is that right? It will be less ambiguous if we can write out what the Jacobian is differentiating towards e.g. $$\nabla_w$$. since the alternative read might be $$\nabla_z$$ .
> >
> > If it's actually implemented as a first order approximation, maybe it's good to write down what is actually implemented as well? e.g. directly changing the parameters along $\tau$ and then measure the network output.
> >
> > **Computational Efficiency**
> > This table is super informative and will make the paper's claim about efficiency stronger! I'd appreciate the authors for adding them to the final paper appendix.

---

> ### Author Response · Authors · 2025-09-19
> **Authors' response to the remaining concerns**
>
> We sincerely thank the reviewer for their response, and are glad to see that most of their concerns are addressed. We address the remaining concerns below:
>
> - _**Further clarification on the notation**_
>
> We apologize for the confusion surrounding the notation. We make them fully clear below:
>
> Recall the definition of the concept sieve $\tau$:
>
> $\tau = \tau_c - \tau_{cn}$, where
>
> - $\tau_{c}$ is the steering vector for concept dataset $D_c$, defined as $\lbrace w_c^{i} - w_0^i \rbrace_{i=1}^{L}$
> - $\tau_\{cn}$ is the steering vector for concept-negated dataset $D_{cn}$, defined as $\lbrace w_{cn}^{i} - w_0^i \rbrace_{i=1}^{L}$.
> - $w^{i} \in \theta$ is the $i^{th}$ layer weight for the UNet model $\phi_\theta$, and $w_0$ represents the weight of the original Stable Diffusion model.
>
> For notation brevity, let $\tau_k$ be the steering vector for dataset $D_k$: $\tau_k = \theta_k - \theta_0$, $k \in \lbrace c,cn \rbrace$. These steering vectors are trained with the MSE loss. The formulation of the loss for each $k$  is as follows:
>
> $L  = ||z - z^{\prime}||^2_2$,  where
>
> - $z  := z_t$ is the latent encoding of the image $x \in D_k$ at timestep $t$, and
>
> - $z^{\prime} := z_t^{\prime} = \phi_{\theta}(z_{t+1}, t, y ~; \theta_0) + \tau_k ~. \nabla_{\theta} \phi_{\theta}(z_{t+1}, t, y ~ ; \theta_0)$. Here,
>
>   - $z_t^{\prime}$ is the first-order Taylor series approximation of $\phi_{\theta}$ w.r.t. $\theta$ around the point $\theta_0$ (original SD model)
>   - The Jacobian term in $z_t^\prime$ should be read as follows: One computes the Jacobian $\nabla$ of the function $\phi_\theta$ w.r.t $\theta~$, i.e., $\nabla_{\theta} (\phi_{\theta}(.))$, and then evaluates it at the point $\theta_0$. Therefore, the Jacobian term is constant in the above equation w.r.t.  $\theta_k$. Hence, $z^{\prime}$ is linear in $\theta_k$, with $\theta_k$ only occurring in the term $\tau_k$, making the optimization a first-order process.
>
> We hope this clarifies the confusion regarding the notation. To summarize:
>
> > "I think it's the partial derivative of the model output w.r.t. the parameters? is that right?"
>
> Yes, as we clarify above, the partial derivative is w.r.t model parameters $\theta$, evaluated at $\theta_0$.
>
> > "If it's actually implemented as a first order approximation, maybe it's good to write down what is actually implemented as well?"
>
> As noted earlier, $z_t^{\prime}$ is, by definition, the first-order Taylor series approximation of $\phi_{\theta}$. The Jacobian term in this approximation is constant w.r.t the optimization variable $\theta_k$, rendering the optimization process of the MSE loss $L$ first-order. In prior works [1], this linearization is typically implemented through an efficient Jacobian–Vector product [2], and we adopt the same approach here to ensure consistency in implementation.
>
> We’ll update the draft with these notational clarifications.
>
> ---
>
> - _**Regarding the computational efficiency table**_
>
> We are glad the reviewer found the table informative and believe it adds significant value to the paper. We assure the reviewer that all the additional results provided in the rebuttal, including the efficiency comparison table, will be added to the appendix of the final manuscript.
>
> ---
>
> **Summary**
>
> We hope that this helps address the remaining concerns of the reviewer, and we thank you again for your valuable engagement which has strengthened our work.
>
> ---
>
> **Citations**
>
> [1] Ortiz-Jimenez et al. Task arithmetic in the tangent space: Improved editing of pre-trained models. 2023.
>
> [2] https://github.com/gortizji/tangent_task_arithmetic/blob/main/src/linearize.py

---

> > ### Author Response · Authors · 2025-09-24
> > **Gentle reminder to review our response on remaining concerns**
> >
> > Dear Reviewer LwxK,
> >
> > We would like to know if our response was able to resolve the remaining concerns you had on notations. We would be happy to present any further clarifications if needed.
> >
> > Thanks,
> >
> > Authors

---

### Decision · Action_Editor_nxah · 2025-09-29

**Recommendation:** Accept as is

**Additional Comments:**

The revisions should ensure consistency in reporting baselines (e.g., AdvUnlearn results) and provide clear formulation for structured LPIPS in the main text.

The broader impact statement and NSFW disclaimer, promised in rebuttal, shall be included in the final camera-ready draft.

**Audience:**

Yes

**Audience Explanation:**

The problem of concept erasure in generative models is of broad interest to the TMLR readership, spanning communities in AI safety, generative modeling, and responsible AI deployment. The proposed method contributes both a practical tool (Concept Siever) and conceptual insights (paired negated datasets, controllable forgetting strength) that will inform future work on model editing and regulation compliance. Readers interested in diffusion models, concept unlearning, or ethical safeguards for generative systems will find this work relevant.

**Claims And Evidence:**

Yes

**Claims Explanation:**

The submission introduces a novel framework for controllable concept erasure in diffusion models, combining automated paired data generation with localized weight retraining. Across the review cycle, the authors have addressed concerns on notation, methodology, and evaluation robustness. The paper now provides extensive experimental validation: quantitative comparisons on the I2P benchmark, user studies, ablations on attention layers, and runtime analyses. Rebuttals included clarifications on the mathematical formulation (first-order approximation, Jacobian interpretation), additional results across multiple Stable Diffusion backbones (SD-1.4, SD-2.1, SDXL), and evaluations on adversarial prompts.

While some core assumptions (e.g., linear separability of concepts in weight space) remain heuristic, the authors acknowledge these limitations and contextualize them within the literature. The empirical evidence convincingly demonstrates efficacy, specificity, and efficiency of the approach relative to baselines, and the presentation has been strengthened with improved clarity. Taken together, the claims are supported to the degree expected for TMLR acceptance.